# Allosteric activation of an ion channel triggered by modification of mechanosensitive nano-pockets

Charalampos Kapsalis[1], Bolin Wang[1,2,3], Hassane El Mkami[4,5], Samantha J. Pitt [6], Jason R. Schnell [7], Terry K. Smith[8], Jonathan D. Lippiat [3], Bela E. Bode [5,8] & Christos Pliotas [1,2,3,5]*

Lipid availability within transmembrane nano-pockets of ion channels is linked with mechanosensation. However, the effect of hindering lipid-chain penetration into nano-pockets on channel structure has not been demonstrated. Here we identify nano-pockets on the large conductance mechanosensitive channel MscL, the high-pressure threshold channel. We restrict lipid-chain access to the nano-pockets by mutagenesis and sulfhydryl modification, and monitor channel conformation by PELDOR/DEER spectroscopy. For a single site located at the entrance of the nano-pockets and distal to the channel pore we generate an allosteric response in the absence of tension. Single-channel recordings reveal a significant decrease in the pressure activation threshold of the modified channel and a subconducting state in the absence of applied tension. Threshold is restored to wild-type levels upon reduction of the sulfhydryl modification. The modification associated with the conformational change restricts lipid access to the nano-pocket, interrupting the contact between the membrane and the channel that mediates mechanosensitivity.

[1] Biomedical Sciences Research Complex, School of Biology, University of St Andrews, St Andrews KY16 9ST, UK. [2] Astbury Centre for Structural Molecular Biology, University of Leeds, Leeds LS2 9JT, UK. [3] School of Biomedical Sciences, Faculty of Biological Sciences, University of Leeds, Leeds LS2 9JT, UK. [4] School of Physics and Astronomy, University of St Andrews, St Andrews KY16 9SS, UK. [5] Centre of Magnetic Resonance, University of St Andrews, St Andrews KY16 9ST, UK. [6] School of Medicine, University of St Andrews, St Andrews KY16 9TF, UK. [7] Department of Biochemistry, University of Oxford, Oxford OX1 3QU, UK. [8] Biomedical Sciences Research Complex, School of Chemistry, University of St Andrews, St Andrews KY16 9ST, UK. *email: c.pliotas@leeds.ac.uk

Membrane-embedded proteins have evolved along with lipids, and their function is frequently modulated by membrane lipid composition and physical properties[1]. Mechanical force is a fundamental physical property of membranes and a prominent example of mechanical regulation is the mechanosensitive (MS) family of ion channels, which alter their structure and pore conductance in response to changes in membrane tension[2,3]. This ancient mechanism of using forces to regulate ion channel activity[4,5], is not restricted to MS channels, but is observed in many other types of eukaryotic channels[5–8].

Previously, Pulsed ELectron DOuble Resonance (PELDOR) or Double Electron-Electron Resonance (DEER) spectroscopy[9–12] was combined with site-directed spin labeling (SDSL)[13] to identify gating transitions of the homo-heptameric MS channel of small conductance (MscS)[14–17] and other transporters[18–23], as well as to assess folding of membrane proteins[24]. In particular, hydrophobic pockets within transmembrane (TM) helices are an integral structural feature of the MscS channel in solution[14] and lipid bilayers[15]. These pockets are found in MscS structures solved by various laboratories, under different conditions, adopting different conformations, and originating from multiple organisms[25–29]. These nano-pockets (NPs) were thought to be empty (or voids) until a high-resolution MscS X-ray structure revealed the presence of endogenous lipids[30], which led to the proposal of the lipid moves first model[4,30] derived from the force from lipid principle[5]. The model predicts that when lipid-chain access to the NPs is restricted, the MS channel senses this discrepancy and responds by altering its structure. Recent studies on Piezo1, TREK-2, and MscS, favored the lipid moves first model, suggesting a potential common mechanism across life kingdoms[29,31–34].

The MS channel of large conductance (MscL) was the first MS channel to be identified[35]. Its non-selective pore could reach an estimated full opening diameter of ~28 Å[36] and a 3 nS conductance[35], allowing even small proteins to pass through[37]. Crystal structures of the homo-pentameric MscL have been obtained for bacterial (*Mycobacterium tuberculosis*, TbMscL)[38] and archaeal orthologues (*Methanosarcina acetivorans*, MaMscL)[39]. Each subunit consists of an N-terminal short amphipathic helix S1, two TM α-helices (TM1 and TM2) connected through a periplasmic loop, and a C-terminal cytosolic α-helix. A fully open X-ray structure has not been reported to date, and the mechanism by which the expanded MaMscL was achieved is not understood[39]. Among all known mechanically gated channels, MscL presents the highest pressure activation threshold (22 mN/m, TbMscL and 12 mN/m, *Escherichia coli*, EcMscL), significantly higher than MscS (7 mN/m) or Piezo (4 mN/m) and an order of magnitude higher than TRAAK (0.5–4 mN/m)[7,40–42].

Previous attempts to stabilize an activated state relied upon modifications of pore-lining residues[43–45]. These studies proposed modified sites, which were inaccessible to membrane lipids and thus did not provide information on lipid-mediated activation occurring in native channels. Bilayer properties such as lipid composition[1,46–49] or thickness and curvature[50–52] have also been proposed to play a role in MscL function, but the way these activate MscL remains unclear.

MS channels consist of pressure sensitive regions that function by matching the membrane tension profile to achieve energy minimization[32]. MscS NPs are filled with lipid chains[29,30], which may also exist in similar MscL crevices[53–55]. However, a key corollary of the lipid moves first model that has not been tested is whether hindering lipid-chain penetration into NPs alters channel structure and function. We postulated that chemical modifications or mutagenesis that restrict lipid-chain access to NPs should trigger MS channel opening.

To directly test the lipid moves first hypothesis, we couple PELDOR, with cysteine modification on multiple sites targeting the nano-pocket (NP) including the entrance and proximal regions and then interrogate changes in MscL conformation. We hypothesize that this disruption would prevent acyl-chain occupancy in the NPs causing an allosteric structural response resulting in an MS channel partial or full opening, even in the absence of applied tension. We identify a single site located at the entrance of the NPs (distal to channel pore), which disrupts lipid-chain penetration within the NPs and alters the channel's functional behavior.

## Results

**Strategy for targeted NP modification.** TbMscL was selected as a model system to study MS channel gating because (1) it presents the highest pressure activation threshold amongst known ion channels[40,42] and (2) an X-ray model[38] in the closed conformation (PDB 2OAR) is available, allowing accurate in silico distance modeling for PELDOR.

We employed SDSL as a means of chemically modifying NPs to promote mechanical activation and reporting for PELDOR, which offers high accuracy in reporting stoichiometry and conformation of full-length multimeric channels in a native environment[9,14,15].

If the essential feature of MS channel activation is lipid removal from the NPs, then opening should occur either by an increase in lateral tension pulling bilayer lipids away from the NPs or by sterically excluding lipid-chain contacts with NP-forming residues, such as A18, which is adjacent to pore constriction- forming L17 (Fig. 1a, b). Twenty single-cysteine TbMscL mutants were distributed across all protein domains with respect to the NPs: (a) within the NP (I23C, F79C), (b) NP-entrance (F5C, A85, Y87, L89), (c) NP-proximal sites (L2C, F34C, F84C, F88C), (d) distal control sites (L42C, V48C, N70C, L72C, L73C, E102C, V112C), and (e) a charged cluster (R98C, K99C, K100C) known to bind lipid headgroups[56] (Fig. 1a, c).

**Characterization of conformation and oligomericity by PELDOR.** When experiencing a hypo-osmotic shock (Supplementary Fig. 1), mutants displayed similar protection to WT channel. Full-length proteins were solubilized in *n*-Dodecyl-*β*-d-maltopyranoside (DDM), spin-labeled with S-(2,2,5,5-tetramethyl-2,5-dihydro-1H-pyrrol-3-yl)methyl methanesulfonothioate (MTSSL) (thereafter designated as R1) and eluted in single homogeneous peaks consistent with an MscL pentamer (Supplementary Fig. 2a).

Prior to PELDOR, spin labeling efficiency and mobility were assessed by cwEPR (Supplementary Fig. 2b and Table 1)[9,57]. Positions in the pocket itself (I23C, F79C), TM (F34C, F84C, A85C) and N-terminal helix, L2C (NP proximal) and L5C (NP entrance), displayed low spin-labeling efficiency (<20%).

TM (L42C, N70C, L72C, L73C, Y87C, F88C, L89C, R98C, K99C, K100C) and solution exposed (V48C, E102C, V112C) mutants presented high (>60%) spin-labeling efficiency. For distance measurements, low labeling efficiency reduces both EPR signal intensity and modulation depth (dipolar coupling), leading to a quadratic loss (this is only strictly true for systems with two spin labels[17]), which is particularly important for MscL, because the C5 symmetry expected for the pentagon gives rise to two distances ($D_1$ and $D_2$, Fig. 2b) and reliable measurement of $D_2$ requires longer evolution time[58].

Seven positions resulted in broad distance distributions and were excluded from further analysis (Supplementary Fig. 3). Thirteen positions gave PELDOR time traces with clear dipolar oscillations in the raw data, which allowed the extraction of well-defined distance distributions using DeerAnalysis[59] (Fig. 2a). Ratios, $D_2/D_1$, found to be close to 1.6 for all positions, as expected for C5 symmetry (Supplementary Table 2). We note that

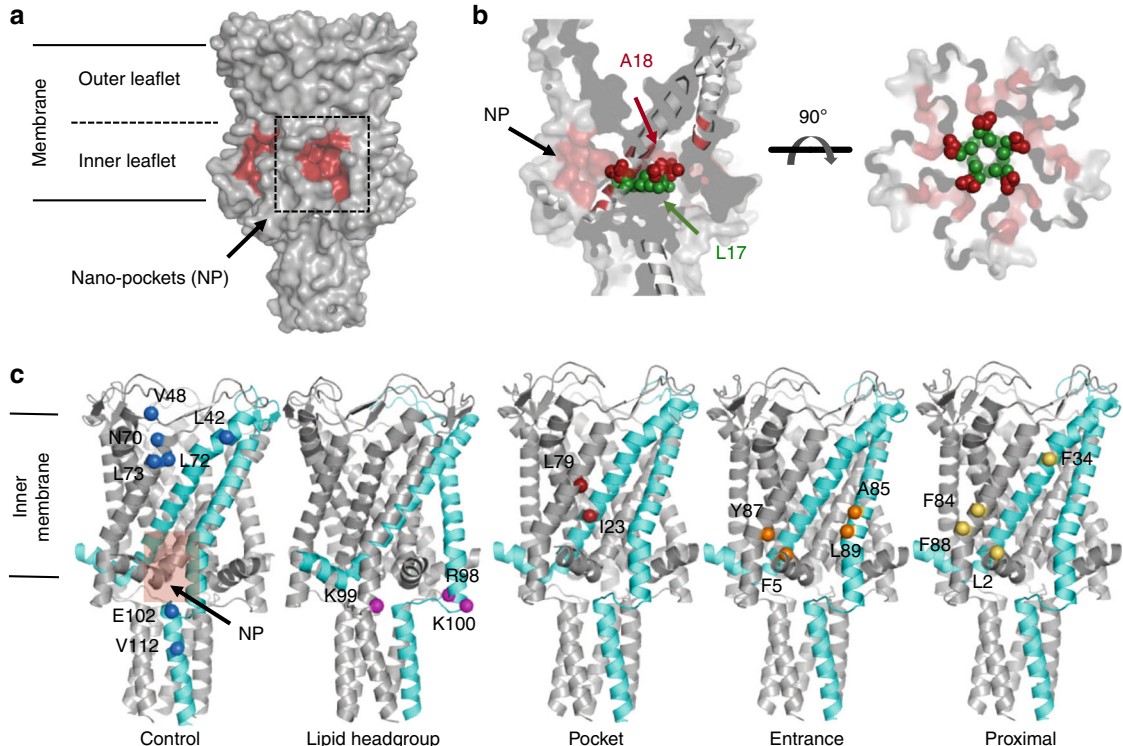

**Fig. 1** Targeting the NP by SDSL. **a** TbMscL NP-forming region residues (red surface view) located within the inner-leaflet (or cytoplasmic) of the membrane. **b** Side and top views of the NPs and residues A18 (red spheres), located at the bottom of the NP, and L17 (green spheres), located at the narrow pore constriction site. **c** Mutation sites on TbMscL (PDB 2OAR). A single subunit is shown in cyan cartoon. From left to right, selected sites in respect to the NPs (transparent red box), Control (blue), Lipid headgroup (magenta), Pocket (dark red), Entrance (orange), and Proximal (yellow) spheres

for a given time window, $D_2$ will be more background correction-dependent than $D_1$, mostly reducing the long-distance contribution. This may shift ratios to smaller numbers and will mainly be relevant in the amber and red regions of the distance distributions (Fig. 2)[9,60]. The experimental modulation depths are consistent with pentamer formation though the multimeric state could not be shown unequivocally (Supplementary Table 3).

Distance ratios for V48R1, L73R1, E102R1, V112R1 (controls), F88R1 (proximal), R98R1, K99R1, K100R1 (lipid headgroup), and Y87R1 (entrance) mutants coincided with the closed pentameric structure (Fig. 2a and Supplementary Fig. 4a) in the absence of bilayer compression (DDM). This is in contrast to the lower MscS pressure threshold[40], previously found to stabilize an open conformation in DDM[14].

**NP-entrance cysteine modification leads to channel opening.** L42R1, N70R1, L72R1 (controls), and L89R1 (entrance) presented significant differences between experimental and modeled distances. L42 lies adjacent to a TbMscL region (i.e., G47 to I56), where the original electron density map has been re-modeled[38,39]. N70R1 and L72R1 presented experimental distances consistent with one or the average of the two predicted R1 side chain populations, respectively, independent of the modeling software used (MMM[61] or MtsslWizard[62]) (Fig. 2a and Supplementary Fig. 4b). Interestingly, both L89R1 distances were significantly longer than expected by ~3 Å ($D_1$) and ~5 Å ($D_2$). L89R1 belongs to the NP-entrance subgroup and its flexible and bulky R1 side chain would be able to sweep penetrating acyl chains away from NPs (Supplementary Fig. 4c). $D_1$ and $D_2$ distance shifts observed for L89R1 correspond to an increase of 5 Å of MscL's inner-diameter (~9 Å total pore diameter), a third of full opening[36,48].

**Monitoring channel structure in lipid membranes.** To investigate the discrepancies between modeled and experimental distances, we reconstituted L42R1 and L89R1, along with Y87R1, F88R1, R98R1, K100R1, and E102R1 into two lipid systems (liposomes and nanodiscs) and performed PELDOR. We hypothesized that flexible MTSSL side chain would clash with acyl chains penetrating the NP. We thus tested whether lateral compression exerted from the lipid bilayer could reverse this effect by supplying sufficient force to push lipid chains into the NP. First, we sought to identify associated endogenous membrane lipids that may be required for protein function and therefore to be included in the reconstitution. The endogenous lipid content of WT protein was investigated by [31]P-NMR and distinct peaks consistent with $sn$-2 $lyso$-phosphatidylethanolamine (2-LPE), phosphatidylethanolamine (PE), and phosphatidylglycerol (PG) were detected (Supplementary Fig. 5a). PE and PG together make up the vast majority of lipid types in *E. coli* and the ratio of intensities for PE and PG [31]P signals observed in the MscL samples is similar to the ratio in which these lipid headgroups are found in *E. coli* membranes (~3:1, PE/PG)[63], suggesting that TbMscL had no preference for lipid headgroup type. In addition, endogenous lipids extracted from purified WT and R1 TbMscL mutants, were subjected to electrospray mass spectrometry (ES–MS). PE and PG lipids, as well as phosphatidic acid (PA), were present in most samples, however, no specificity was detected (Supplementary Table 4 and Fig. 5b). We therefore reconstituted the selected R1 derivatives into *E. coli* lipid extract liposomes and dimyristoyl-phosphatidylcholine (DMPC) containing nanodiscs, with MSP1D1 as the scaffold to allow one MscL pentamer per nanodisc.

No change was observed between DDM and liposomes for L42R1, indicating the channel state remains unchanged (Fig. 3a and Supplementary Fig. 5c). In contrast, a significant distance

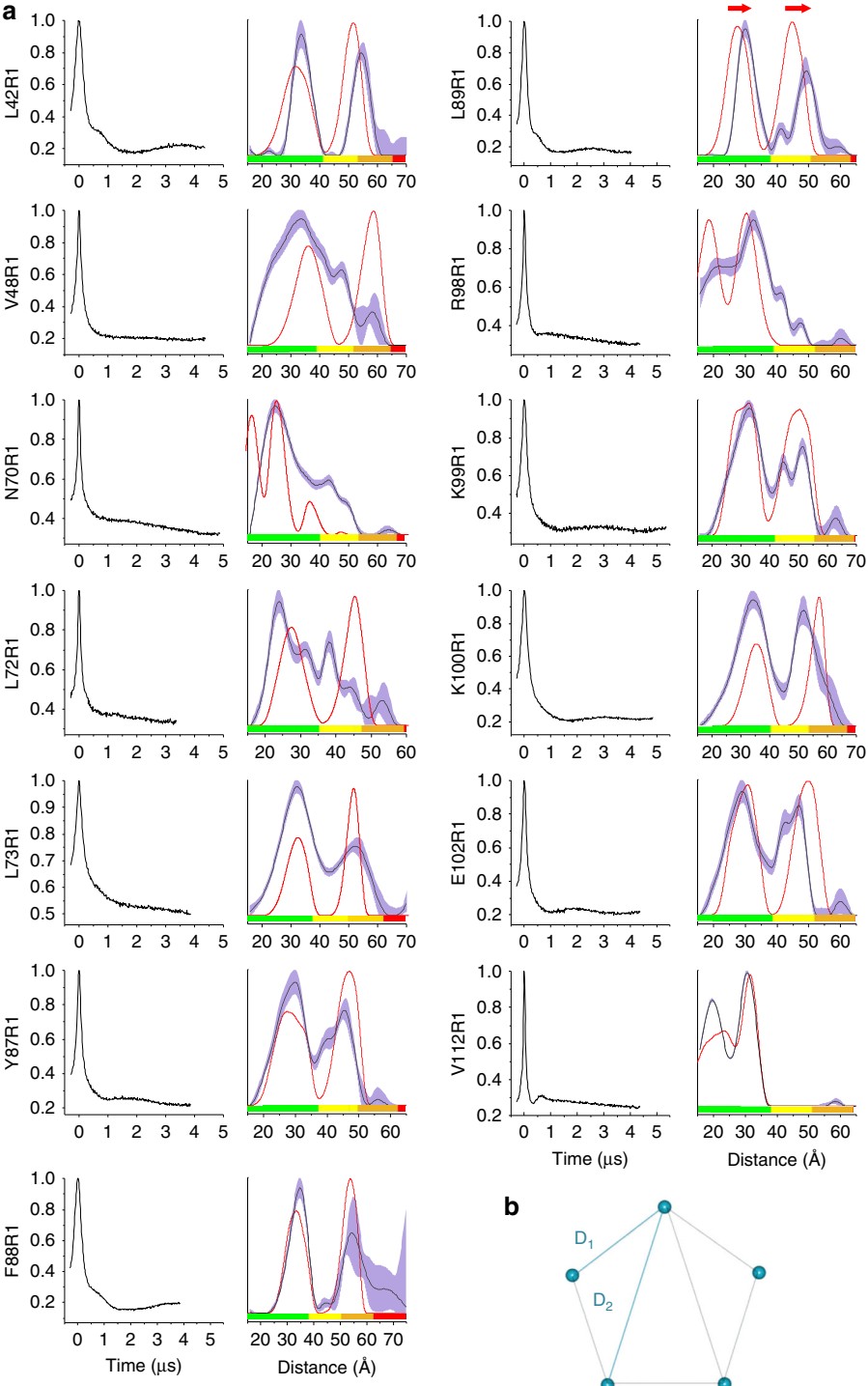

**Fig. 2** TbMscL structural state observed by PELDOR in DDM micelles. **a** Raw (background uncorrected) PELDOR time-domain traces (left columns) and resulting distance distributions (right columns). Red lines correspond to in sillico modeled distances for (PDB 2OAR) TbMscL (closed state). Blue shade areas correspond to mean ± 2σ confidence intervals of measured distributions (calculated by DeerAnalysis validation tool, for details see methods) and rainbow color bars indicate the reliability of the measured distance ranges (calculated by DeerAnalysis: green, shape reliable; yellow, mean and width reliable; orange, mean reliable; red, no quantification possible), corresponding to experimental time windows used. L89R1 distance distribution shift (highlighted by red arrows) between the closed structure TbMscL model and PELDOR distance measurements in solution. **b** Pentameric TbMscL presents C5 symmetry giving rise to two expected distances, D1 and D2. Observation of both distances serves as a control for C5 symmetry

shift was observed for L89R1 in both liposomes and nanodiscs. The distances shorten and coincide with the modeled closed state distances, suggesting a reversible induced change, from the expanded to closed state. Importantly, different lipid compositions used for reconstitution had no effect on distances,

suggesting no specific lipids are required to promote MscL mechanical gating closure. Furthermore, the use of curved (liposomes) and flat (nanodiscs) bilayers allowed us to assess the effect of curvature on MscL conformation. After reconstituting F88R1 and L89R1 and performing PELDOR, distances did

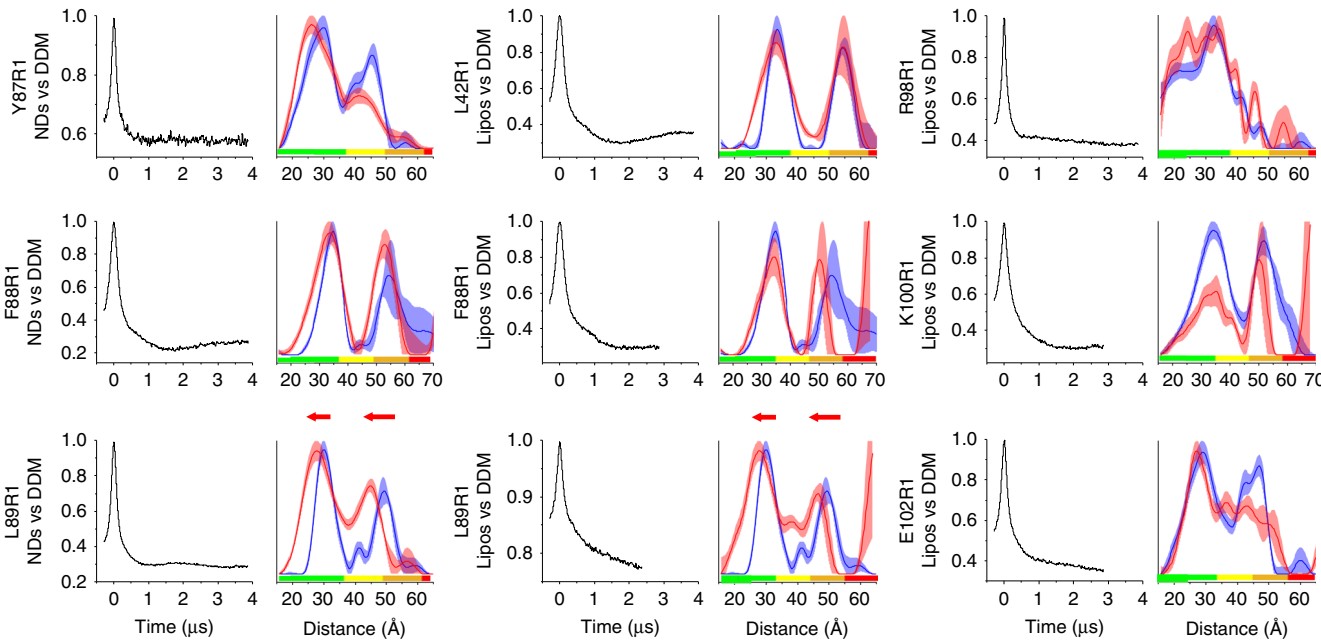

**Fig. 3** TbMscL gating transitions in lipid bilayers. Raw (background uncorrected) PELDOR time-domain traces (left columns) of reconstituted TbMscL in nanodiscs (NDs) or liposomes (Lipos) mutants and distance distributions (right columns). Shaded areas correspond to mean ± $2\sigma$ confidence intervals of lipid-reconstituted (red) and solution (blue) mutant distributions as calculated by the DeerAnalysis validation tool (for details see methods), and rainbow color bars indicate the degree of reliability of the measured distance ranges. Rainbow color bars corresponding to the shortest of the two experimental time windows used for the obtained spectra in each case (see Fig. 2), are included. When these time windows are not equal (liposome samples of F88R1, L89R1, R98R1, K100R1, and E102R1), the shortest is that of the reconstituted sample. Distance shift between L89R1 (solution) and (lipid bilayers) suggesting a reversible conformational change in the presence of lateral compression is highlighted by red arrows

not considerably differ between the two lipid systems, however both equally revert L89R1 back to its original closed conformation (Fig. 3a and Supplementary Fig. 5c).

**NP-entrance W mutation leads to similar MscL expansion.** To exclude the possibility that lipids and/or detergent might affect spin label distances, we tested the effect of other modifications (or mutations) of the same site, on global MscL changes. We thus paired the L89W mutation with four EPR reporters, i.e., N70R1, L72R1, L73R1, K100R1, which previously showed no changes in PELDOR distances between DDM and the modeled closed state. L89W presented functional behavior in vivo similar to WT and all double mutants (introduction of a W and a C) eluted as single homogeneous pentamers (Supplementary Figs. 1, 2a). In L89W, tryptophan's bulky indole side chain is expected to sterically hinder acyl-chain penetration. As a control, K100R1 was paired with F88W, which is adjacent to L89 but not at the NP-entrance (Fig. 1a and Supplementary Fig. 4c).

Distance distributions of N70R1-L89W and K100R1-L89W in DDM did not differ from N70R1 and K100R1, and the same was found with K100R1-F88W (Fig. 4a and Supplementary Fig. 6). However, very short distances were observed for L72R1-L89W and L73R1-L89W in DDM, below PELDOR's detection range (<18 Å) (Fig. 4a). This could arise from TM2 rotation that brings residues facing towards the channel's pore, which occurs during pore expansion, similar to MaMscL (Supplementary Fig. 7 and 8). When MscL was reconstituted into nanodiscs the distributions were within the PELDOR range and had fully reversed to distances in agreement with closed TbMscL. To confirm that short distances arise from a conformational change, low temperature (80 K) cwEPR spectra were recorded. In DDM, both L72R1-L89W and L73R1-L89W displayed the characteristic linewidth broadening associated with dipolar interaction at distances ≤ 15 Å when compared with the lipid-reconstituted

samples (Fig. 4b). To rule out the possibility that short distances arise from protein unfolding, L72R1-L89W and E102R1 were thawed after PELDOR measurements, run over size exclusion chromatography (SEC), and eluted as folded pentamers (Supplementary Fig. 6c).

The observed changes, were supported by probing solvent accessibility[64] with 3-pulse Electron Spin Echo Envelope Modulation (3pESEEM). A remarkable increase in $^2$H accessibility was observed in DDM for L72R1-L89W, over the buried L72R1, followed by a significant decrease in $^2$H accessibility, when L72R1-L89W was reconstituted into ND (Fig. 4c). This indicates that acyl-chain penetration hindrance from the NPs caused by the bulky tryptophan side chain results in a significant solvent exposure of the TM2 outer-leaflet domain that is completely reversible within lipids.

PELDOR, 3pESEEM, and cwEPR data jointly suggest the L89W DDM state exhibits structural similarities to L89R1 DDM and the MaMscL expanded structure (PDB 4Y7K) (Supplementary Fig. 7a)[39]. Despite equivalent MaMscL residues not being resolved in the expanded state, in silico spin labeling of F78, K97, and Q94 located at TM2 edges and adopting similar orientations to respective TbMscL L89W-paired EPR reporters L72, L73, and K100, displayed similar relative distance changes (absolute distances differ owing to different constructs), consistent with anti-clockwise TM2 rotation (Supplementary Figs. 7, 8).

**Monitoring single-channel function in lipids.** To evaluate the functional effects of mutagenesis and the covalent attachment of the label at the entrance of the NPs and TbMscL modification, we performed single-channel recordings from patches excised from collapsed Giant Unilamellar Vesicles (GUVs) containing WT, L89W, or L89R1, and applied negative pressure. EcMscL is known to activate at lytic tensions, thus causing liposome rupture before channel activation, whereas TbMscL is known to activate

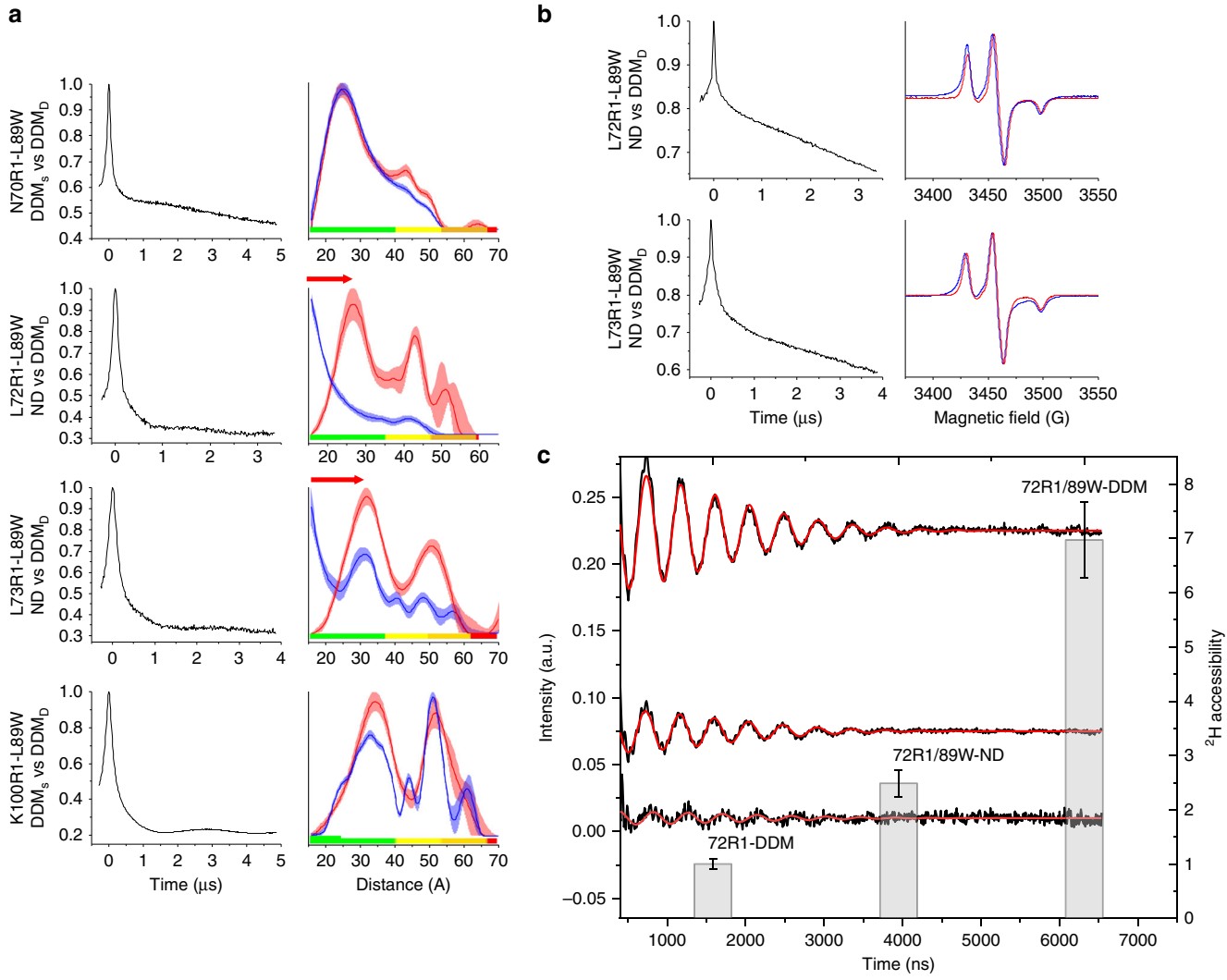

**Fig. 4** Monitoring distal to the NP allosteric site transitions caused by L89W modification. **a** Raw (background uncorrected) PELDOR time-domain traces (left columns) of the double mutants and distance distributions (right columns). Shade colored areas correspond to mean ± 2σ confidence intervals of distributions as calculated by the DeerAnalysis validation tool and rainbow color bars indicate the range of distance reliability corresponding to the experimental time windows used in each PELDOR experiment. Double mutants in DDM (DDM$_D$) are always represented as blue shade areas. For L72R1-L89W and L73R1-L89W red peaks correspond to distance distributions of the Nanodisc (ND) reconstituted sample, whereas for N70R1-L89W and K100R1-L89W they correspond to the distributions of the respective single-cysteine mutants obtained in DDM (DDMs) (see Fig. 2). **b** Raw (background uncorrected) PELDOR time-domain traces of the double mutants in solution (left columns) and cwEPR first derivative spectra of the corresponding double mutant in DDM (blue line) and lipid-reconstituted (red line). Magnified views of the low field peaks are shown for clarity as insets. **c** Time-domain 3pESEEM experimental spectra (black curve) and fitting (red curve) of single mutant L72R1 in DDM and double mutant L72R1-L89W in DDM (~7 × higher $^2$H accessibility than 72R1 in DDM) and in ND (~3 × lower accessibility than same double mutant in DDM). Column bars present the $^2$H accessibility derived from the fitting of 3pESEEM time-domain traces (see Methods). Errors are estimated on the residual of the fitted model over the whole range of the studied data

at a threshold double to EcMscL[42,65]. For WT protein, three out of four patches exhibited large-amplitude single-channel openings with a mean conductance of 2.74 ± 0.10 nS ($n = 3$), but only when pressures of at least −140 mmHg were applied to the patch pipette (Fig. 5 and Supplementary Fig. 9). Single-channel currents were also recorded from patches containing L89W TbMscL with 2.43 ± 0.27 nS ($n = 5$) conductance, but with lower pressure thresholds ranging – 60 to – 80 mmHg (Fig. 5b). These patches also exhibited a high level of background current in the absence of applied pressure, with a conductance of 25.1 ± 4.9 nS ($n = 7$) compared with 1.68 ± 0.89 nS ($n = 4$) from patches containing WT protein. Occasionally, low-amplitude unitary transitions were recorded, which enabled the calculation of a mean unitary sub-conductance of 0.248 ± 0.022 nS ($n = 11$) (Supplementary

Fig. 9c). Channel state transitions of similar conductance were not observed in any of our WT recordings ($n = 4$).

We recorded 2.98 ± 0.07 nS ($n = 4$) conductance openings from 4 out of 13 patches excised from GUVs containing L89R1, but these also activated at threshold pressures lower than WT, ranging between −30 and −120 mmHg (Fig. 5 and Supplementary Fig. 9). Similar to L89W, patches containing L89R1 exhibited background current when no pressure was applied, with a mean macroscopic patch conductance of 15.8 ± 2.3 nS ($n = 13$). Where low-amplitude unitary openings could be resolved from the L89R1 background current (Fig. 5 and Supplementary Fig. 9), these had a mean unitary conductance of 0.273 ± 0.039 nS ($n = 7$). Adding 5 mM DTT to the experimental solutions to remove the label from the modified MscL (L89C) gave rise to

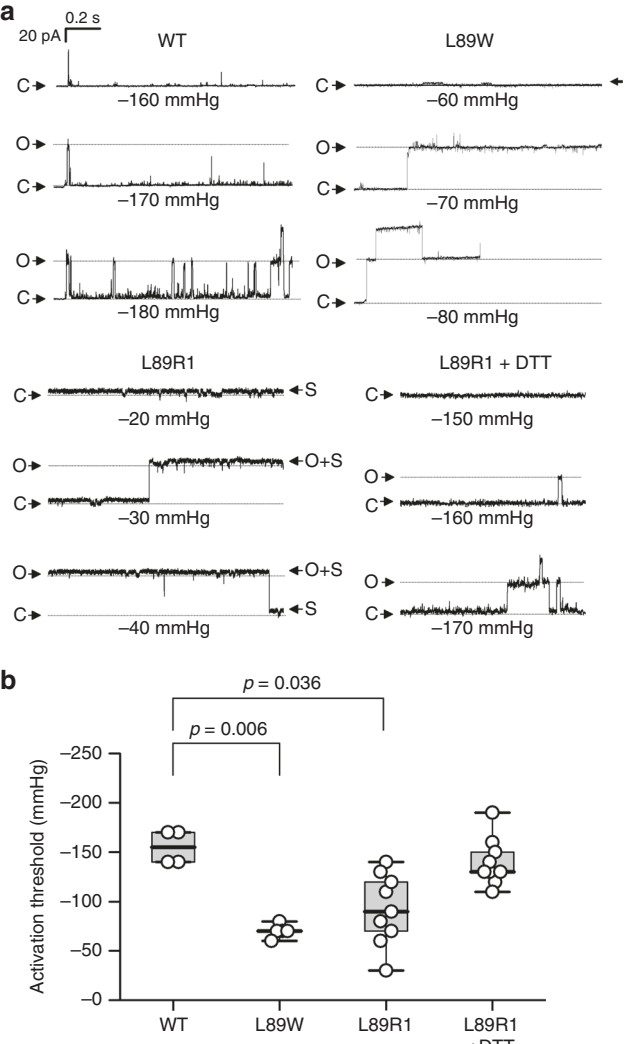

**Fig. 5** Electrophysiological measurements of unitary TbMscL channels. **a** Representative patch clamp recordings at 20 mV from patches excised from GUVs containing WT, L89W, and L89R1 (±5 mM DTT) TbMscL protein. The pressure applied during the recording is indicated below each trace. The current levels representing the closed (C) and fully opened (O, ~60 pA for WT, L89W and L89R1, and 50 pA for L89R1 plus DTT) channels are indicated. Note that channels opening to a subconductance level (S, ~5 pA) were also recorded with L89W and L89R1 (modified channel). **b** Distribution of threshold pressures, applied at 10 mmHg intervals, at which pressure-activated MscL full channel openings were first observed. Data points represent individual excised patches, whiskers show the full range of values, the boxes represent the 25–75 percentile range, and the horizontal line is the median. The p values from independent samples Kruskal–Wallis Test with pairwise comparisons are indicated where these were <0.05, with respect to WT channels

2.30 ± 0.17 nS ($n = 4$) conductance channels that had higher activation thresholds than L89R1, ranging between −120 and −190 mmHg (Fig. 5).

**Lipid headgroup unbinding does not alter MscL state.** To investigate whether obstruction of the acyl chains located within NPs, rather than lipid headgroups in the NP vicinity caused the observed channel opening, a set of double and quadruple mutants were generated to abolish lipid headgroup binding and monitor channel conformation by PELDOR. The R98Q and triple R98Q/K99Q/K100Q (3Q) mutants were generated, to either partially or fully neutralize the positively charged cluster, which is a hot spot for anionic phospholipid headgroups[56]. Each of these mutants was paired with two EPR reporters, one at the upper TM1 (L42R1) and one at the lower TM2 (Y87R1 for 3Q, F88R1 for R98Q). An increase in solvent exposure and spin label mobility was observed for 3Q-Y87R1, compared with 87R1, as expected for a less-hydrophobic environment and consistent with local lipid removal (Supplementary Fig. 10). However, these observations did not correlate with distance changes suggesting MscL state is unaffected by lipid headgroup binding. R98Q-L42R1, R98Q-F88R1, and 3Q-L42R1 did not present any significant distance or $^2$H accessibility change (Supplementary Fig. 10).

**The effect of modification on NP-accessible lipid chains.** Molecular dynamic (MD) simulations were carried out to test whether acyl-chain blocking by L89R1 was physically reasonable. We set up atomistic MD simulations of L89R1 and WT TbMscL within DMPC lipid bilayers and DDM micelles to monitor over time the number of lipid chains residing within MscL's NPs to compare with PELDOR measurements.

Lipid chains penetrated all five WT TbMscL NPs for most of the simulations and make direct contact with A18, which is located at the NP bottom and adjacent to the narrow constriction pore-forming L17 (Fig. 6a and Supplementary Movie 1). In contrast, only a couple of the inner-leaflet lipids were able to access the NPs and make direct contact with A18, for L89R1 (Fig. 6a, b and Supplementary Movie 2). The rest of the acyl chains were bent owing to steric-clash of the R1 side chain with the mid to end part of the DMPC lipid chains. This translated into fewer total DMPC and DDM single acyl chains (Fig. 6c and Supplementary Fig. 11a) residing within the NPs of L89R1, compared with WT, and total fewer contacts for the NP-forming region (Supplementary Fig. 11b). We observed that each NP could fit up to two single acyl chains (10 per pentamer). An average of 5.54 and 2.47 DMPC lipid chains penetrated all five NPs, for WT and L89R1, respectively (Fig. 6c), suggesting that the majority of the WT NPs are continually occupied by lipid chains, in contrast to L89R1. Changes in NP lipid-chain occupancy showed correlation over time with TM1 (pore-lining helix) root-mean-square deviation (RMSD) changes (vs closed TbMscL), in the presence of the modification (L89R1), but not WT (Supplementary Fig. 11c).

To investigate whether there is (1) continuous contact between A18 (NP), residue 89 (native Leu or R1 modification) and lipids (i.e., whether the pressure from lipid tails is still transduced to the gate, but mediated by R1) or (2) increased tension at the polar MscL residues at the interfacial levels or (3) any other significant pattern of tension redistribution, we calculated the pairwise energies between (a) NP and lipidchain, (b) NP and 89 (Leu or R1), (c) lipid chain and 89 (Leu or R1) and (d) TM1 and 89 (Leu or R1) (Supplementary Table 5). The values agree with a model in which L89R1 modification receives most of the load of the bilayer lipid chains attempting to penetrate the NP. We calculated that for the last 20 ns of our simulations an ~110 × larger energy transmission from the acyl chain to the NP is occurring in the absence of the modification (MTSSL), thus forcing the channel to adopt an energetically favorable closed state (WT). No significant relative total energy difference was observed between L89 (or L89R1) and the NP, lipid chains, and TM1. This is consistent with a model in which the modification does not cause structural distortions to the gate and prevents formation of the final contact between the lipid and the NP, thus cutting the link between the bilayer and the channel. Several R1 rotamers (L89R1), efficiently occluded the NP- entrance and restricted access to protruding inner-leaflet lipid chains by opposing steric clashes (Fig. 6 and Supplementary Movie 1, 2).

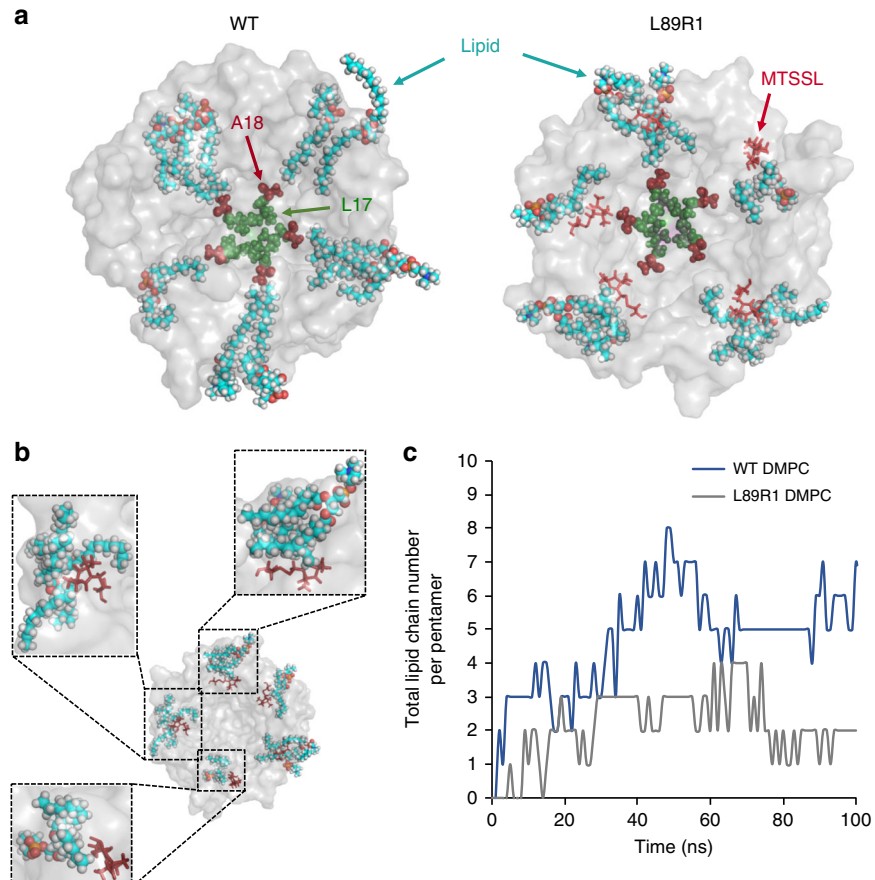

**Fig. 6** L89R1 side chain steric clashes restrict lipid-chain penetration within the NP. **a** Representative MD simulation top pore view snapshots of WT and L89R1 (transparent gray surface view), residues L17 (green spheres, pore constriction site), and A18 (red spheres, NP bottom) are highlighted. NP-associated lipids are shown in cyan sticks. The bulk of the bilayer lipids is hidden for clarity. All five lipid chains make direct contact with A18 in the WT channel and only a couple of them with the same residue in L89R1 channel as MTSSL (red sticks) clashes with lipids obstructing their entrance to the NPs. **b** Close-up views (dashed line boxes) of some different steric-clash interactions between MTSSL and lipids on L89R1 TbMscL. **c** Comparison of the total number of lipid chains residing within the NPs (per single TbMscL pentamer) over time (100 ns atomistic MD simulation), between WT (blue line) and L89R1 (gray line) TbMscL channels

## Discussion

We have demonstrated that disruption of lipid-chain penetration within NPs located at the cytoplasmic leaflet of the bilayer generates structural and functional responses in MS channels. One proposed mechanism for MS channel activation is the use of stored elastic energy[66]. If this was the only mechanism, then gating transitions should be initiated for and observed in more than one of the MscL sites tested in the absence of lateral compression (DDM). This effect however was not construct-specific (TbMscL), as EcMscL I24R1 also adopted a closed state in DDM[50], whereas MscS adopted an open conformation in DDM micelles[9,14,30] and certain crystallization conditions[26,28], and is capable of accommodating two lipids (closed[29]), and one lipid (open[30]), within its NPs.

Despite notable structural differences reflected in their pressure activation threshold, MscL and MscS are both mechanically gated. A valid unified model should be consistent with their membrane tension-sensitivity but also the different extents of their mechanical responses under certain stimuli. According to the lipid moves first model, the protein stored elastic energy is released when sufficient mechanical force is supplied to the system that is capable of reducing lipid availability within the NPs (Fig. 7). In that respect, the energy activation threshold should be different for each individual MS channel and dependent upon its distinct TM structural landscape. Inhibition of Piezo's function[34]

owing to depletion of specific lipids provided an early hint towards the potential applicability of such a model to eukaryotic channels[33] and similarities in their NP-forming regions[4,67–69]. In multimodal K2P MS channels, acyl chains bind to the NPs and block the conduction pathway[6], whereas external tension is still required to promote mechanical activation[31].

In the absence of lateral compression (DDM, PELDOR), the MTS-modification introduces steric clashes that prevent acyl chains belonging either to detergent or endogenous lipids from entering the NP. In doing so, the energy activation barrier is reduced and elastic energy drives MscL opening (Fig. 7). In the presence of compression (nanodiscs, PELDOR), the lipid chains are forced back into the NPs, and the channel adopts a closed conformation. This closed state of NP-entrance modified TbMscL is energetically less stable than the unmodified and WT channels, owing to the reduced probability of NP lipid occupancy, as revealed by our MD simulations. This is reflected in the presence of the subconductance openings at baseline tension (lipid-glass adhesion) and reduced tension required for the fully open state of L89W (or R1) (Figs. 5, 7, and Supplementary Fig. 12).

L89R1, located at the entrance of the NPs, promoted a conformational change, with both $D_1$ and $D_2$ shifting toward longer distances, while maintaining their pentameric state. Instead, when seven distinct MscL modified variants that showed a deviation towards longer distances from the modeled distances (closed state)

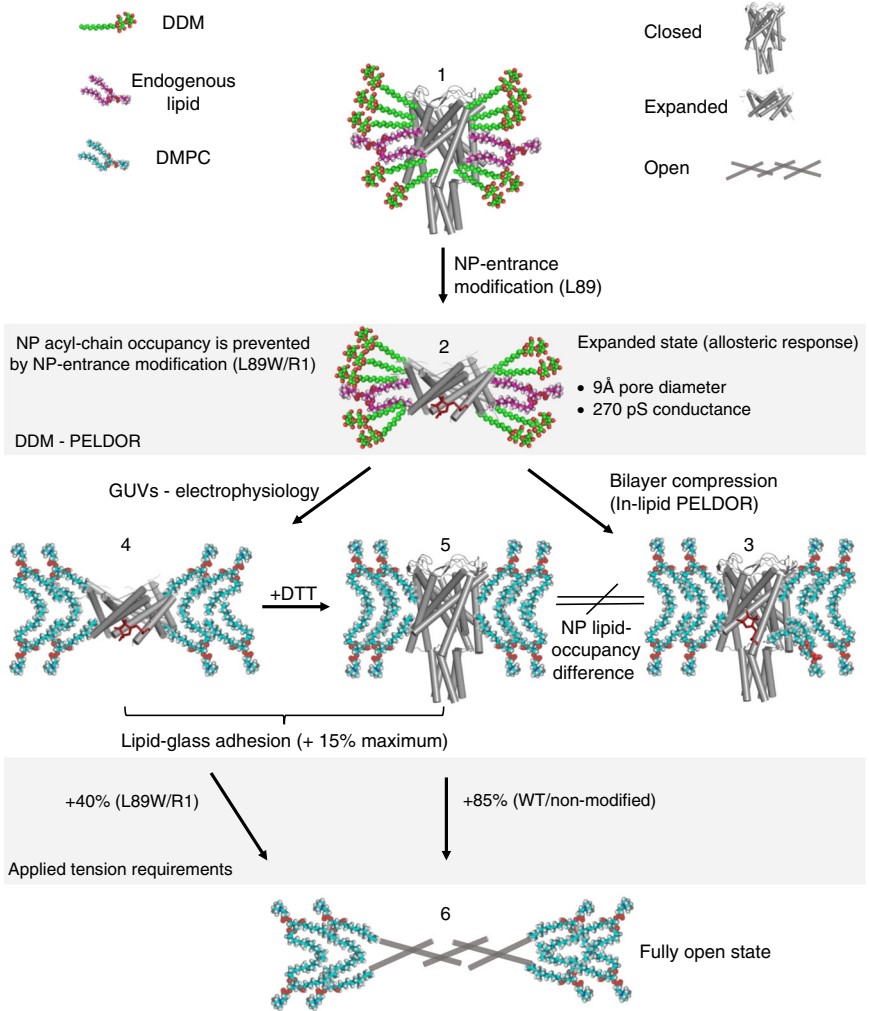

**Fig. 7** Safety-pin acting lipid chains model for mechanical sensing and MscL gating transitions From top to bottom: WT protein or cysteine mutant in detergent solution with NP penetrated by lipids (1), L89R1 (or L89W) mutant in DDM with the MTS-obstructing (non-penetrated NP) conformations. The latter result in an allosteric channel response leading to a structural change (opening), stabilizing an expanded state (2), which is reversible in the presence of lateral compression (closure), during MscL reconstitution in lipid bilayers (nanodiscs or liposomes) (3). Then, upon minimal tension and/or lipid-glass adhesion tension conditions (adding a further up to 15% of tension) the expanded state could be promoted in lipids (4). Therefore, most of the applied tension energy is used for NP lipid removal. The effect could be reversed by the addition of reducing agents and removal of the modification, resulting in channel closure (5), consistent with our electrophysiology recordings. Then only an extra ~40% will be required for NP-entrance modified MscL to transit from the expanded to the fully open state (6), owing to NP lipid removal caused by the modification, in contrast to WT, which requires higher energy tension in order to remove the first lipid(s) from the NPs. L89W (or 89R1) in lipid nanodiscs, although in the closed state, is energetically distinct to WT channel, owing to its reduced NP lipid occupancy probability revealed by our MD studies. The X-ray structure of the expanded MaMscL (PDB 4Y7J) and closed TbMscL (PDB 2OAR) have been used as models for the proposed conformational change. Co-purified endogenous lipids are represented in purple spheres and detergent molecules in green spheres

were reconstituted in liposomes (curved and mixed lipid) and nanodiscs (flat and single lipid), only L89R1 distances shifted significantly, reversing back to the closed state, whereas all other mutants were unchanged (Figs. 2a and 3). Importantly, when L89 was mutated to W, substantial conformational changes distal to the allosteric site occurred, in regions previously unchanged in the absence of NP modifications (Fig. 4). Therefore, the mode of mechanical action seems to be confined within a well-defined space, a pressure sensitive NP, which is a prerequisite for activation. Abolishing lipid headgroups from interacting with the charged cluster[56], had no effect on conformation (Supplementary Fig. 10), whereas specificity to the mediators of lateral tension, the lipids, investigated by $^{31}$P-NMR and ES–MS, was not detected (Supplementary Fig. 5 and Table 4), consistent with a previous study[1].

L89 was identified as a unique site, crucial for gating. To exclude any artefacts from spin label clashes with neighboring residues and/or lipids and investigate the effect on channel structure caused by other molecules at the allosteric site entrance, a tryptophan was introduced (L89W) paired with cysteine mutants on individual sites to report on conformational changes. This method allowed us to disentangle global structural changes from local perturbations and assess whether an allosteric (non-pore residues are triggered) mechanism could drive mechanical activation.

The expanded L89R1 TbMscL state formed an open pore of ~9 Å in diameter at the cytoplasmic leaflet, i.e., a 5 Å diameter increase reported by PELDOR, in addition to ~4 Å closed pore diameter of TbMscL (PDB 2OAR) (Fig. 2a and Supplementary Fig. 12), an opening sufficient for water and ions to pass through. The MaMscL expanded structure, which presents similar pore dimensions to L89R1 TbMscL, is thought to represent a non-conductive state, however this may be owing to

the presence of the fusion protein linked to MscL's C- terminus[39]. PELDOR on the other hand, allows full-length proteins to be structurally investigated under physiological conditions and in lipid bilayers of choice. This diameter of the expanded state is consistent with the ~270 pS sub-conductance state[70] recorded from L89W/R1 (Supplementary Fig. 12). Here, only baseline tension was present in our gigaseal patch clamp recordings, originating from lipid-glass adherence (0.5–4 mN/m)[71], much lower than TbMscL's activation threshold (~22 mN/m), and up to ~15% of tension required for full channel opening. Importantly, when GUV reconstituted L89R1 was incubated with DTT (MTSSL removal), the pressure threshold of the channel was restored to WT levels (Fig. 5). These findings suggest a reversible process within lipids, in which the effects on channel's gating behavior are caused by the presence of the bulky modifying molecule (R1 or W instead of L) and not the mutation (Cys) on this site. When modifications were introduced to multiple sites, spanning all protein domains, the channel remained closed, suggesting high site specificity in mechanical response, consistent with the existence of highly pressure sensitive regions within MscL.

The L89W/R1 expanded and conducting state presents structural similarities to the expanded pentameric MaMscL[39] X-ray model, in which TM2 helix undergoes a rotation accompanied by a simultaneous tilting, leading to pore expansion (Supplementary Fig. 7). However, the mechanism by which the MaMscL expanded state was obtained is not understood[39]. There are differences in the crystallization (1-week vs 5-month crystal growth) between the closed and expanded states, although both structures were obtained from similar samples. Lipid degradation (acyl-cleavage) may have been catalyzed by $(NH_4)_2SO_4$, present only in the crystallization liquor of the expanded form, or hydrolysis over the long crystal growth period[39]. Hence, degradation of endogenous lipids could result in lipid displacement from the NPs similar to L89R1, resulting in channel expansion.

Lipid-chain contacts to pore-forming TM1 residue A18 (NP bottom), which is located adjacent to pore constriction site L17, were consistently higher in WT channel than in L89R1, over the course of the MD experiments (Fig. 6c and Supplementary Fig. 11a), and their changes correlated with TM1 perturbations, not caused by MTSSL's chain presence (Supplementary Fig. 11c and Table 5). In our MD simulations, we did not observe significant conformational changes, as artificially high tension conditions[72] or biasing techniques like steered[55] or targeted MD[73] were not introduced.

Certain spin label L89R1 conformers restricted lipid-chain access to the NPs, but these restrictions did not simultaneously apply to all five NPs (Fig. 6 and Supplementary Fig. 11a). Spin labels are EPR reporters, with intrinsic flexibility, which leads to a distribution of rotamers and are not designed to guard NPs. However, we showed that when modifications are engineered at an appropriate site, i.e., L89 (W or R1), in the absence of lateral compression (detergent) and externally applied tension (lipids, under minimal lipid-glass adhesion tension), they could have a stabilizing effect on an expanded/conducting state (Figs. 2a, 5a, and 7) and a significant overall lowering of MscL's pressure activation threshold, owing to NP-entrance modification. Variability of this effect may arise from the labeling modification efficiency of L89R1 (~60%) (Supplementary Tables 1 and 3) (approximately three labels per pentamer) and the individual ability of MTSSL rotamers to restrict lipid-chain access to NPs (Fig. 6b). L89R1 pressure threshold is on average lower than WT but approaches WT values after DTT incubation. This is supported by the low activation threshold of L89W, which is equipped with a tryptophan on every site and exhibited very limited variability similar to WT (Fig. 5).

The mechanism by which lipid chains act as negative allosteric modulators and penetrate NPs to prevent channel opening is reminiscent of grenade safety pins (Fig. 7). Upon removal of the safety pin (lipid moves first from the NP) the grenade becomes activated (absence of a bilayer compression), but the pressing force prevents the grenade from detonating (bilayer compression is greater than stored protein elastic energy preventing channel opening, nanodiscs). Upon removal of the pressing force (DDM or external tension application in lipids) detonation can occur following a short hysteresis time: stored elastic energy becomes greater than minimum activation energy and released, resulting in channel activation.

For L89W, ~45% of the total tension that is required for WT to achieve full opening (~70 vs 150 mmHg for WT), is sufficient to promote a transition from the closed to the fully open state and ≤ 40% (0–70 mmHg) from the expanded/sub-conducting (~9 Å, 1/3 of full opening and 270 pS, 1/10 of total conductance) to the fully open state (~25 Å and ~3 nS) (Fig. 7). Because this mutation reduces NP acyl-chain occupancy, less lateral membrane stretch is required to remove lipids from the NP, which is consistent with our model that a lipid has to move first and out of the NP (allosteric site). This event constitutes the first essential step in MscL's mechanical response and is consistent with the presence of two lipids and one lipid within the NPs of MscS, in the closed[29] and open[30] states, respectively.

NP lipid-chain removal could be achieved (a) through modification at the entrance of the allosteric site (NP) either in the absence of bilayer compression or under baseline (lipid-glass adhesion) tension for NP-entrance modified channels or (b) upon external tension application for WT or non-NP-entrance modified channels. Therefore, a large portion of the tension energy is required to remove the lipid-chain(s) from the NP and initiate a mechanical response, under native conditions. Independent to the method used, the consequence is the destabilization of the closed state and the increase of the open probability of a substantial conducting state (Fig. 7 and Supplementary Fig. 12).

We have observed that the degree of lipid-chain penetration disruption depends upon the precise location, with respect to the site of allostery (NP), and the type of modification. This suggests that the NP is an allosteric site controlling pore gating (by site-specific and reversible covalent modification of cysteines) and it may thus constitute a potential drug target. De novo design and synthesis of molecules with properties amenable to access and compete for the MS channel NPs could disrupt lipid NP interactions, thus affording allosteric modulation of ion channel pores in lipid bilayers. Such molecules could be tuned to be selective and target specific MS channels, depending on their unique TM structural landscape.

We have shown that hindering lipid acyl-chain access within defined TM hydrophobic crevices leads to a coordinated response in the structure, which is directly linked to the function of an MS ion channel. The lipid acts as a negative allosteric modulator, distal to the pore, and its moving out from the NP as a consequence of increased tension, results in a decrease in channel activation free energy. Upon reversal of membrane stretching, the lipid moves back and occupies the NP in order to increase the free activation energy and keep the channel closed since a leaky channel would be lethal to the cell. Therefore, the lipid, in the case of MS channels, acts as a negative allosteric modulator and conformational changes are not driven by specific ligand-protein interactions. We offer insights into how lateral tension is transmitted from the membrane through the lipid chains to the NP and gates ion channels. We demonstrate that allosteric mechanical gating of ion channels can be modulated by reversible cysteine modification at a site located at the entrance of the NP and distal to the channel pore.

## Methods

**Materials**. DDM anagrade was obtained from Anatrace or Glycon (Germany). Isopropyl-β-D-thiogalactoside (IPTG) was obtained from Formedium, and (Tris(2-carboxyethyl)phosphine (TCEP) was obtained from Thermo Scientific. The MTSSL spin label was obtained from Toronto Research Chemicals. Phospholipids, *E. coli* polar extract and DMPC (14:0/14:0) were purchased from Avanti Polar Lipids and Biobeads from Biorad. All other chemicals unless otherwise stated were obtained from Sigma.

**Cell viability assays**. MJF612[74] *E. coli* cells were grown overnight in LB (non-transformed) or LB with 50 μg/mL (transformed cells). The next morning, over-night cultures were diluted 1:100 in LB and grown at 37 °C with shaking until they reached $OD_{600} = 0.4$–0.5. At this point the cultures were diluted 1:1 in LB supplemented with 2 mM IPTG and 1 M NaCl and grown in the same conditions for 1 h. Cultures were then diluted to 1:1000 in either 0.2 M (osmotic shock) or 0.5 M (mock shock) NaCl and incubated in a static incubator at 37 °C for 20 min. Finally, the shock solutions were diluted to 1:100 in LB with 0.5 M NaCl and plated on agar plates. The following morning the number of colonies was counted for each plate. Assays were independently repeated in quadruplicates.

**Site-directed mutagenesis and protein expression**. TbMscL mutants carrying a C-terminal 6xHis-tag were generated on a pJ411:140126 vector. Primer sequences used in mutagenesis reactions are shown in Supplementary Table 6. Mutant plasmids were transformed into BL21(DE3) (ThermoFisher) *E. coli* cells. Cells were grown in 0.5 L LB medium at 37 °C until they reached $OD_{600} \approx 0.9$ and subsequently cooled down to 25 °C and induced with 1 mM IPTG for 4 h, except for the R98Q/K99Q/K100Q-Y87R1 mutant for which induction lasted 2 h. Cells were harvested by centrifugation at $4000 \times g$ and stored at −80 °C, until further use.

**Protein purification and spin labeling**. Cell pellets were resuspended in phosphate-buffered saline and subjected to lysis using a cell disrupter at 30 kpsi. In order to remove cell debris and intact cells, the suspension was centrifuged at $4000 \times g$ for 20 min and the resulting supernatant was centrifuged again at $100,000 \times g$ for 1 h. The membrane pellet was resuspended and solubilized in buffer containing 50 mM sodium phosphate of pH 7.5, 300 mM NaCl, 10% v/v glycerol, 50 mM imidazole and 1.5% w/v DDM (solubilization buffer), and incubated at 4 °C for 1 h. The sample was then centrifuged at $4000 \times g$ for 10 min and the super-natant was passed through a $Ni^{2+}$-nitrilotriacetic acid ($Ni^{2+}$-NTA) column containing 0.5–0.75 mL Ni-NTA beads. The column was then washed with 10 mL buffer containing 50 mM sodium phosphate of pH 7.5, 300 mM NaCl, 10% v/v glycerol and 0.05% w/v DDM (wash buffer) and then with 5 mL wash buffer supplemented with 3 mM TCEP, for reduction of the cysteines. Afterwards, MTSSL dissolved in wash buffer at a 10-fold excess of the expected protein concentration was added to the column and left to react for 2 h at 4 °C. The protein was then eluted from the column with 5 mL of wash buffer supplemented with 300 mM imidazole. Finally, the protein was subjected to SEC using a Superdex 200 column (GE Healthcare) equilibrated with buffer containing 50 mM sodium phosphate of pH 7.5, 300 mM NaCl and 0.05% w/v DDM. Collected fractions of pure TbMscL were then concentrated to ~800 μM monomer concentration, which is suitable for the EPR samples preparation.

**Reconstitution in lipid bilayers (PELDOR samples)**. Liposomes: 20 mg/mL stocks of *E. coli* polar lipid extract in lipid buffer (containing 50 mM sodium phosphate buffer of pH 7.5 and 300 mM NaCl) were subjected to 10 cycles of flash-freezing in liquid $N_2$ and thawing at 60 °C. After addition of 3–6% v/v Triton X-100, purified protein and lipids were mixed in a 1/100 molar ratio and incubated at 50 °C for 30 min. Lipid buffer was then added to the mixture in a 1/1 v/v ratio and left incubating with 8 mg of Biobeads per μL of added Triton X-100 overnight at 4 °C. After removal of the Biobeads, the mixture was centrifuged at $100,000 \times g$ for 50 min. The resulting proteoliposome pellet was then resuspended in 35 μL of lipid buffer.

Nanodiscs: Purified TbMscL was mixed with the membrane scaffold protein MSP1D1[75] and solubilized DMPC lipids in a 1/2/160 respective molar ratio. The mixture was diluted to 3–4 mL with lipid buffer (as described earlier), supplemented with 1% Triton X-100 and incubated at 25 °C for 30 min. Afterwards, 0.8–1 g of biobeads (Biorad) were added per 1 mL of mixture and left incubating at 25 °C for 4 h. Finally, after removal of the biobeads, nanodisc samples were concentrated to a final volume of 35 μL.

Distance modeling: In silico spin labeling and distance modeling were performed either on the TbMscL (pentameric closed conformation) (PDB 2OAR)[38] or MaMscL (pentameric closed) (PDB 4Y7K) and (pentameric expanded) (PDB 4Y7J) X-ray model structures[39], using MtsslWizard[62] for all mutants in this study and MMM[61] (for TbMscL N70R1). Residues were mutated to cysteines before spin labeling. In *MtsslWizard*, the thorough search option was selected for the MTSSL rotamers and Van der Waals restraints were set to tight option. If this search resulted in < 50 rotamers, the loose option was used instead. In MMM, labeling was performed at ambient temperature (298 K setting).

**PELDOR (DEER) distance measurements and analysis**. Purified TbMscL detergent or reconstituted samples were diluted by 50% with deuterated ethylene glycol as cryoprotectant to a final monomer concentration of ~400 μM and 70 μL of the mixture were loaded in 3 mm (OD) quartz tubes and flash-frozen in liquid $N_2$.

Measurements were performed with an ELEXSYS E580 pulsed Q band (34 GHz) Bruker spectrometer with a TE012 cavity at 50 K. The frequency offset between the detection ($\nu_A$) and pump ($\nu_B$) frequencies was 80 MHz. The pulse sequence used was $(\pi/2)_A - \tau_1 - \pi_A - (\tau_1 + t) - \pi_B - (\tau_2 - t) - \pi_A - \tau_2 - echo$[76]. Detection frequency pulses were 16 ns and 32 ns for $(\pi/2)_A$ and $\pi_A$, respectively, separated by $\tau_1 = 380$ ns, whereas the pump $\pi_B$ pulse was either 12 ns or 14 ns long. Shot repetition time for all measurements was 3 ms. Acquired data were analyzed using the Matlab plugin DeerAnalysis2016[59]. Time-domain spectra were fitted with a mono-exponential decay function, background-corrected and analyzed by Tikhonov-regularization[77]. The validation tool was also employed as previously described[17]. For each trace, the background fitting starting point varied between 5% and 80% of the total time length of the trace in 16 steps, whereas 50% random noise was added with 50 trials per step, resulting in 800 trials in total. Finally, data sets that where > 15% above the best (lowest) RMSD were excluded. In our data, cases where the background fitting function of the best RMSD was continuously rising (theoretically corresponding to negative spin and, therefore, sample concentration) were associated with real but incomplete oscillations of the trace, rather than artefacts. To overcome these truncation effects, either the measurement time window needs to be increased, which is not possible due to the low signal intensity and intrinsic dephasing, or the portion of the trace that displays the incomplete oscillation needs to be cut. The latter would result in substantial shortening of the time window, which would greatly compromise the accuracy of the resulting distance distributions. Therefore, the best fit was consistently used for the full traces of all data sets. This was found in all cases not to lead to significant changes to the distance distribution with respect to fitting the background to the latter two thirds of the trace. The modulation depth $\lambda$ was estimated to 0.46 by fitting the equation $\Delta = 1 - (1 - \lambda f)^{n-1}$[17] using $n = 5$ and the experimental modulation depths ($\Delta$) and labeling degrees ($f$) (Supplementary Table 1 and 3). Calculated labeling degrees were obtained by resubstituting the estimated $\lambda$ and solving for $f$ (Supplementary Table 3).

**3pESEEM measurements and analysis**. Sample preparation is identical to the one described earlier for PELDOR measurements. X-band three pulse ESEEM measurements were performed on a Bruker ELEXSYS E580 spectrometer with a 4 mm dielectric resonator (MD4). Measurements were conducted at 80 K. The 3-pulse sequence used for the experiments is $\pi/2 - \tau - \pi/2 - T - \pi/2 - \tau - echo$ with $\pi/2 = 16$ ns and inter-pulse delay $\tau$ set at 140 ns and 450 ns corresponding, respectively, to blind spots of the proton and deuterium. The delay $\tau$ was incremented in 12 ns steps. All the measurements were performed at the maximum of the filed sweep spectrum of the nitroxide. For solvent accessibility determination, the three pulse ESEEM experiments were set up with $\tau$ that corresponds to the proton blind spot. The obtained time-domain traces were background-corrected, apodized with a hamming window and zero-filled prior to Fourier transformation. The solvent accessibility can be determined either from the modulation depth in time domain or the corresponding amplitude of the deuterium peak in frequency domain. In the present work, the modulation depth for each mutant was determined by fitting the corresponding time-domain trace by a damped harmonic oscillation function and the outcome from the fitting was used to estimate the solvent accessibility.

**cwEPR spectroscopy**. Room temperature: 30 μL of purified TbMscL samples at monomer concentrations of ~450 μM were loaded into glass capillary tubes. Spectra were recorded on a Bruker EMX 10/12 spectrometer operating at ~9 GHz with 100 kHz modulation frequency using an ELEXSYS Super High Sensitivity Probehead (Bruker ER4122SHQE). Spectra were averaged for 20 scans with a modulation amplitude of 0.2 mT, magnetic field sweep width of 16 mT and magnetic field center at 350.5 mT. X-axis resolution was set to 512, microwave bridge power to 1 mW and both time constant and conversion time were set to 40.96 ms.

Low temperature (i.e., 80 K). Sample preparation was identical to the one described for PELDOR. Measurements were carried out using a Bruker ELEXSYS E580 X-band spectrometer with an Oxford Instruments CF935 helium flow cryostat operating with liquid nitrogen and Flexline probehead housing 5 mm dielectric ring resonator (MD5) that was critically coupled. Spectra were taken as single scans with a modulation amplitude of 0.2 mT, magnetic field sweep width of 20 mT and magnetic field center at 346.0 mT. X axis resolution was set to 1024, microwave bridge power to 1.5 μW, time constant to 40.96 ms and conversion time was set to 346 ms.

**Atomistic MD simulations**. The crystal structure of TbMscL (PDB:2OAR)[38] sourced from the Orientations of Proteins in Membranes (OPM) database[78] was used as a starting structure to build simulation systems, whereas all the hetero atoms from the structure were removed. The TbMscL89R1 mutation was made by substituting LEU with CYS for residue 89 and conjugating R1 spin label (MTSSL) to the mutated CYS for all the five chains of TbMscL using CHARMM-GUI PDB Manipulation Options. The *Membrane Builder* module in CHARMM-GUI was used to insert the TbMscL structure into a pre-equilibrated patch of DMPC bilayer

containing approximately 385 lipids and occupying an area of $120 \times 120$ Å$^2$. The protein and membrane bilayer were solvated with TIP3P water and 150 mM NaCl.

All calculations were performed in an NPT (constant particle number, pressure, and temperature) ensemble at 303.15 K using GROMACS_2016.4[79] with CHARMM36 force field[80]. The particle mesh Ewald (PME)[10] method was applied to calculate electrostatic forces, and the van der Waals interactions were smoothly switched off at 10–12 Å by the force-switch manner. The time step was set to 2 fs in conjunction with the LINCS algorithm[11]. After the standard minimization and equilibration steps using the GROMACS input scripts generated by CHARMM-GUI, 100 ns dynamic simulation was calculated. Identical set up was followed for TbMscL WT and L89R1 in DDM micelles. For the latter, 150 DDM molecules were used for single-channel incorporation. At least three independent 100 ns atomistic MD simulations experiments were set up and performed for each of the L89R1 and WT TbMscL channels. The number of single acyl chains occupying the NPs, e.g., ≤ 5 Å distance from A18, was calculated for each 1 ns of the respected MD simulations in VMD. Pairwise energy, RMSD and lipid acyl-chain atom contacts with TbMscL amino acids were calculated using Gromacs command *gmx energy*, *gmx rms*, and *gmx mindist*, respectively. The videos were made through the VMD movie-maker module.

**GUV formation and reconstitution**. The method used was similar to the EPR liposome sample preparation, with the difference that the protein monomer: lipid molar ratio was 1:1000 or 1:5000. Further, after proteoliposomes were formed, they were dried overnight in a desiccator at 4 °C and then rehydrated for 2 h at room temperature in rehydration buffer (50 mM sodium phosphate buffer pH 7.5, 300 mM NaCl, and 400 mM sucrose). GUVs were subsequently collapsed in working solution (5 mM HEPES pH 7.2, adjusted with KOH, 200 mM KCl, and 40 mM MgCl$_2$). For MTSSL removal, TbMscL L89R1 containing liposomes were rehydrated in rehydration buffer (supplemented with 5–10 mM DTT) and were then collapsed for 30 min in working solution containing 5 mM DTT.

**Patch clamp electrophysiology with applied tension**. Patch pipettes were pulled from thick-walled borosilicate glass capillaries (Harvard Apparatus), which when filled with working solution had resistances of 3–6 MΩ. Currents were recorded, lowpass filtered at 2.9 kHz, and digitized at 20 kHz using an EPC10 patch clamp amplifier and Patchmaster software (HEKA Elektronik). Following seal formation and patch excision, a pipette potential of 20 mV, relative to the bath, was applied. Pressure pulses lasting 1.5 s were applied at 10 mmHg intervals every 3 s, using a high-speed pressure clamp (HSPC-1, ALA Scientific Instruments) connected to the pipette holder and controlled through the patch clamp amplifier and software. In all cases tension was applied within a few seconds (1 min maximum) and immediately after excision, to avoid any accidental patch slippage. Recordings were digitally filtered at 1 kHz for analysis and data presentation.

**$^{31}$P-NMR endogenous lipid analysis**. $^{31}$P-NMR spectra were recorded at 243 MHz ($^{31}$P) and 303 K on a Bruker Avance spectrometer equipped with a BBO Prodigy probe. TbMscL protein samples for NMR contained 50 mM Tris, pH 7.5, 150 mM NaCl and 0.04% DDM. Buffer and salt conditions for lipid standards were identical but contained 1% DDM to increase lipid solubility. Spectra were indirectly referenced to the protons of 4,4-dimethyl-4-silapentane-1-sulfonic acid.

**ES–MS of endogenous lipids**. Lipid extractions from purified mutants and proteins were achieved by three successive vigorous extractions with ethanol (90% v/v), after the initial sample was spike with dimystritoyl PE and dimystritoyl PG as internal standards to allow relative quantification. The pooled extracts were dried by nitrogen gas in a glass vial and re-extracted.

Extracts were dissolved in 15 μl of choloroform/methanol (1:2) and 15 μl of acetonitrile/propan-2-ol/water (6/7/2) and analyzed with an Absceix 4000 QTrap, a triple quadrupole mass spectrometer equipped with a nano-electrospray source. Samples were delivered using a Nanomate interface in direct infusion mode (~125 nl/min). Lipid extracts were analyzed in both positive and negative ion modes using a capillary voltage of 1.25 kV. MS/MS scanning (daughter, precursor, and neutral loss scans) were performed using nitrogen as the collision gas with collision energies between 35 and 90 V.

**Reporting summary**. Further information on research design is available in the Nature Research Reporting Summary linked to this article.

## Data availability

Data supporting the findings of this manuscript are available from the corresponding author upon reasonable request. A reporting summary for this Article is available as a Supplementary Information file. The source data underlying Figs. 2, 3, 4a–c, 6 and Supplementary Figures 3, 4a, 5c, 6a–b, 10b–e, 11 are provided as a Source Data file.

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

## Acknowledgements

C.P. is supported by the Royal Society of Edinburgh. C.P. acknowledges support by Tenovus (T15/41) and Carnegie Trust (OS000256) and the University of St Andrews, the University of Leeds and the Chinese Scholarship Council for the C.K. and B.W. studentships. S.J.P. is supported by the Royal Society of Edinburgh. J.S. acknowledges funding from the MRC (M019152 and L018578). B.E.B. and C.P. acknowledge support by the Leverhulme Trust (RPG-2018-397). This work was supported by Wellcome Trust [099149/Z/12/Z] and BBSRC equipment grants (BB/R013780/1). The authors thank Robert Nairn and Benjamin J Lane for help with computational modeling and mutagenesis, Nader Amin with help in acquiring NMR data, professor Ian R Booth for provision of the MJF612 cells and professor James H. Naismith for the WT TbMscL plasmid and useful discussions during the initial stages of the project.

## Author contributions

C.K. performed sample preparation, EPR experiments and analyzed data. B.W. set up and carried out MD simulations. H.E.M. performed ESEEM measurements and analyzed data. S. J.P. and J.D.L. supervised and performed single-molecule measurements and analyzed data. J.S. carried out NMR and analyzed data. T.K.S. carried out lipid ES–MS and analyzed data. B.E.B. supervised EPR experiments and analyzed data. C.P. conceived the project, designed

the experiments, performed research, analyzed data, and supervised the study. C.P. wrote the manuscript with input from all authors. All authors reviewed and approved the paper.

## Competing interests

The authors declare no competing interests.
