## [Peer Review File · Nature Communications]

Reviewers' Comments:

Reviewer #1:

Remarks to the Author:

In this manuscript, the authors have addressed the effect of hindering lipid chain penetration into nano pockets (NPs) on the structure and function of mechanosensitive (MS) channels. They report that it is possible to generate an allosteric mechanical response on the large conductance mechanosensitive channel MscL in the absence of membrane tension. They present convincing evidence that modification of a single amino acid residue at the entrance of NPs restricts the lipid-acyl chain access to the NPs, mimicking membrane tension, and opens the channel, providing experimental evidence for the 'lipid moves first' model.

This is a thorough investigation that appears to have been performed to a high standard and well presented. The conclusions are based on both electron paramagnetic resonance methodologies (PELDOR, ESEEM, and CW-EPR) and fully atomistic molecular dynamics simulations. The principal findings are novel and should be interesting to a broad scientific audience.

I have only one minor comment.

In Fig 4c L72R1-L89W mutant is said to be in DDM and reconstituted in liposomes. However, in the text (L319) figure itself, the mutant is reconstituted in nanodisks. Legend or the figure should be adapted, accordingly.

Reviewer #2:

Remarks to the Author:

The paper 'Allosteric activation of an ion channel by chemical modification of pressure sensitive nano-pockets' by Pliotas and coworkers raises many more questions than provides answers. Unquestionably, differential solvation of the protein by lipids in two radically different (open and closed) conformations must bear a substantial component of total free energy of transition. Yet, why a small pocket transiently occupied by acyl chains in one of the states is so special is unclear. The presented data suggest otherwise.

If the main focus of the work is an allosteric facilitation of channel opening, then the only way to demonstrate the presence of additional energetic contributions besides tension is to record the position of activation (open probability-tension) curves and calculate the transition energy change, or at least the shift of midpoint position. If the goal is to detect some elastic distortion of protein structure in different environments with possible reorientations of sidechains, then any other technique, including PELDOR would be good. As previously (refs 9, 15), the authors firmly avoid providing (or even discussing) functional data obtained through tests such as patch-clamp, which is standard in the field. Instead, they tend to replace direct functional observations of gating with modeled inter-label distance changes that according to their own estimations are grossly insufficient to support the gating transition in tbMscL.

The major result emphasized in this work is that inter-label distances in L89R1 and in L89W are larger by ~6Å in DDM micelles compared to nanodisks or vesicles. So what? The problem is that the authors did not bother to carefully compare the equilibrium conformations of tbMscL between DDM micelle and lipid bilayer and determine the rotameric distributions of cysteine-attached MTSSL labels (with the length of sidechain of about 11 Å) in these environments. The label, which is generally polar, is attached at different locations and how the distance to membrane/micelle interface may affect the inter-label distance distribution is unclear.

The authors provide no analysis of equilibrium between empty and filled states of the pockets. The maximum energy gain from filling the pocket with a single acyl chain is the enthalpic VdW

component of interaction with the pocket wall minus the entropic penalty of the highly bent chain conformation. The experimental free energy of ecMscL opening is 58 kT (Chiang et al., BJ 2004), and that for tbMscL it must be larger. It seems unlikely that populating or emptying all pockets provides sufficient driving force for gating.

PELDOR plots have different ranges in the subplots which makes accurate comparison difficult.

Osmotic survival. Moe, Levine and Blount (JBC 2000) have clearly shown that wt tbMscL does not rescue cells from downshock (500 mM NaCl) the way ecMscL does. The viability index in tbMscL experiments is lower by one order of magnitude. Instead of counting colonies, the authors plated droplets of diluted media and 'visually inspected' the growth. This method has not been tested for accuracy and the functional state of mutants remains questionable.

Experiments with planar bilayers. There is no use of this method with either MscS or MscL. Activating tensions for these channels are between 5 and 12 mN/m. The authors seem to be unaware that the Gibbs-Plateau border (meniscus) sets constant tension in the planar part of the bilayer between 0.2 and 2 mN/m, depending on solvent and composition (H.T. Tien, 1974).

More specific points:

Line 144: "MS channel activation is lipid removal from the NPs, then opening should occur either by an increase in lateral tension of the linked bulk bilayer lipids or by sterically excluding lipid chain contacts to NP-forming residues, such as A18, adjacent to pore constriction forming L17 (Fig 1a & b)."

-- These possibilities do not exhaust all routes for activation, e.g. redistribution of the lateral tension to more sensitive regions without the net increase in the tension of the "linked bulk bilayer lipids".

Line 201: "These findings suggested that TbMscL preferably adopts a closed conformation in absence of lateral bilayer compression, ..."

-- Considering that the net lateral tension/pressure in a relaxed bilayer is essentially zero, it seems strange that authors throughout the manuscript repeatedly single out only the compressive component (supposedly from the lipid tails region). The pressure in the micelle can be distributed in a different manner compared to bilayers.

Fig. 2a: - L42R1 produces right shift comparable to the one by the L89R1. L42 is on the periplasmic side, it is unlikely to affect the access to the crevice that situates at the gate level. Surprisingly N70R1 causes even more dramatic right shift, but this is not discussed.

Line 213: "Interestingly, both measured mean distances of L89R1, were significantly longer than expected by $\sim 3 \text{ \AA}$ (D1) and $\sim 5 \text{ \AA}$ (D2).

-- This enlargement is significantly smaller than would be expected based on the diameter of the conductive pore of MscL ($\sim 30 \text{ \AA}$) or the effective expansion area of MscL in the membrane ($\sim 20 \text{ nm}^2$, i.e. $\sim 20 \text{ \AA}$ increase in the external diameter of MscL). The authors estimate $\sim 6 \text{ \AA}$ increase in channel diameter based on the label displacement, which would produce no more than a subconductive state (if any, as the channel is known to open through a non-conductive pre-expansion stage).

The discord between the expected and observed distances can be explained by a persistent bias in the label conformation in the real micelle environment compared to the modeled "ideal" distribution for all sterically possible rotamers on a bare channel in vacuum. How do the authors explain distances DECREASE (e.g. K100R1) - the scale is comparable to the one with L89R1, so would it be interpreted as a dramatic compaction of the closed MscL then?

Even when the tails are excluded from the crevice by labels grafted in certain positions, the pressure at the region of the crevice does not disappear - it is now applied to the label in those cases, and would be transmitted to the channel.

Line 245: "solution and within liposome distances for L42R1, indicating the channel remains in the same structural state (Fig 3b & S5b). In contrast, a significant distance shift was observed for L89R1 after reconstitution into both liposomes and nanodiscs (Fig 3b & S5b). The distances shorten and coincide with the closed state modeled distances, suggesting a reversible induced change, from the open to closed state."

-- As an alternative interpretation, it might be that in solution MscL is slightly expanded but not in a bilayer.

Line 290: "In one out of four 15 min long recordings for both the WT and L89R1 mutant channel spontaneous openings at +30 mV were observed with open probabilities of 0.27 for the WT and 0.04 for L89R1. The main open state level of these openings was identical for both WT and L89R1 channels, with a measured current amplitude of ~2.2 pA, resulting in a similar calculated chord conductance of ~70 pS (Fig S6)."

-- The electrophysiological recordings are very confusing - extremely low conductance (70 pS vs. 3 ns known for MscL) and somehow very high estimated open probability. This does not seem to be MscL.

Line 305: "Distance distributions of N70R1-L89W and K100R1-L89W in solution did not significantly differ from N70R1 and K100R1 (Fig 4a & S7a) respectively and the same applied to K100R1-F88W (Fig S7b). However, time-domain traces for L72R1-L89W and L73R1-L89W in solution revealed very short distances, below the distance regime commonly accessible by PELDOR (<18 Å) (Fig 4a & b), corresponding to a large decrease for L72R1 and L73R1 (D1) of > 7 Å and 12 Å, respectively. This may arise from a helix rotation that brings residues facing towards the channel's pore, consistent with the relative changes of equivalent residues of MaMscL, where a TM2 helical rotation and expansion occurs (absolute distances differ due to different constructs) (Fig S8)."

-- From TbMscL structure it appears that to bring both L72 and L73 inside the pore, the helix would have to rotate by ~180 degrees, which is much more than the rotation observed in MaMscL (closer to 90 degrees). More importantly, if there were indeed channel opening and TM2 rotation caused by L89W mutation, it should have revealed itself in N70R1 and K100R1 mutants as well - otherwise, the conformational change can be interpreted as some sort of kink or uncoiling in TM2.

Line 321: "To exclude the possibility that short distances arise from either protein unfolding, dissociation or aggregation, L72R1-L89W and E102R1 (control) samples were thawed after PELDOR and run over SEC and both eluted as folded pentamers (Fig S7c)."

- Why were these specific mutants chosen for the folding test rather than L72R1-L89W and L73R1-L89W which actually had the change?

Line 367: "An increase in solvent exposure (3pESEEM) and spin label mobility (room temperature CW368 EPR) was observed for 3Q-Y87R1, compared to 87R1, as expected for a less hydrophobic environment and consistent with local lipid removal (Fig S9a & b). Previously, NMR and ES370 MS detected endogenous phospholipids in the detergent purified protein samples (solution), which would not be able to bind after neutralization of the charged cluster (Fig 3a & S5a, Table S3). However, these discrepancies in solvent accessibility and label mobility did not correlate with distance changes (Fig S9c), suggesting MscL state is unaffected by lipid headgroup binding."

-- Would the position and/or presence of the lipid tails in the crevice be affected by the headgroup unbinding as well? Why don't just test the activation threshold of L89W or L89R1 in patch-clamp?

Line 409: "Several R1 rotamers (L89R1), efficiently "guarded" the NP entrance and restricted access to protruding inner-leaflet lipid-chains by opposing steric clashes, not observed in the WT channel (Fig 5a,b & Vid S1). This further supports the PELDOR, ESEEM and CW-EPR experimental data suggesting L89R1's mode of action is via disruption of NP acyl chain penetration."

-- While the decrease in the contacts between the lipid tails and A18 can be clearly seen in simulations and the overall approach in MD looks adequate, it is only a minimal and not the most

crucial information that can potentially be extracted from these MD simulations. It is not surprising that the bulky side chain at L89 location would restrict access of lipid tails to the crevice. More interesting question is whether there are ANY signs in simulation that it would facilitate channel opening. Obviously, one cannot expect full opening on the scale of 100 ns, however, some small changes might be already noticed, e.g.: is there any widening of the pore at the level of the gate? Is there a continuous contact A18--R1--lipids (i.e. whether the pressure from lipid tails is still transduced to the gate, just mediated by R1) or there is an empty crevice around A18? Any signs of increased tension at the polar MscL residues at the interfacial levels or any other significant pattern of tension redistribution? The latter can be estimated by postprocessing of the simulation traces either in software that allows 3D force mapping (it was even applied to MscL simulations before, e.g. see Ollila OH et al. Phys Rev Lett. 2009; or Vanegas JM, Arroyo M. PLoS One. 2014) or at least it can be analyzed using VMD/NAMD plugin for estimation of pairwise forces and energies between various selections (e.g. lipid/water medium and specific residues on MscL surface. So far, it is just an assumption that the observed change in contacts would favor channel opening.

Line 420: "We have demonstrated that disruption of lipid chain penetration within NPs allosterically generates an MS channel structural response."

-- That statement is misleading. Most of the effects are observed on solubilized channel (an environment with rather unnatural pattern of external forces), whereas in bilayer setting MscL tended to be in the crystal-like closed conformation. Even in solution, the single-residue modification experiments (like L89R1_ cannot exclude that the conformational change in the channel was caused by the environment rather than modification as the label both induces (allegedly) conformational change and reports on it. The two-point modifications are contradictory as L72R1-L89W and L73R1-L89W indicate a dramatic helix rotation (or compaction of the pore), whereas other nearby residues remain in positions similar to the crystallographic closed state. MD simulations showed exclusion of lipid tails from contacts with A18 but provided no evidence for any allosteric change in the channel.

Line 425: "Elastic energy was restored when lateral compression was re-introduced by reconstitution into lipid bilayers thereby forcing lipid chains back into the NPs and closing MscL."

-- Authors persistently describe the mechanical effect of the resting bilayer as only "pressure", thus ignoring well-established notion that there is equally large component of a lateral tension at the polar/nonpolar interfacial level in the bilayer, which balances the pressure and undoubtedly plays mechanical role too - just acting on other set of residues. It is better to be more precise in the expressions. The authors make no distinction between the elastic energy of the protein and that of the membrane, making their arguments throughout the paper difficult to follow.

Line 427: "In contrast, all MscS mutants previously tested adopt an open conformation in solution."

-- The true conformation of an open MscS is far from being a settled topic.

Line 455: "In the absence of lateral compression, the MTS-modification introduces steric clashes that prevent acyl chains belonging either to detergent or endogenous lipids from entering the NP. In doing so, the energy activation barrier is reduced and elastic energy drives MscL opening (Fig 6). In the presence of high lipid compression, the acyl chains force the label to adopt an alternative conformer that permits acyl chain access to the NPs (Fig 6)."

-- The presented data from MD simulations do not match this statement. There were no simulations in detergent, so the effect on the tails is just an assumption based mostly on spectrometry data. For the bilayer, simulations show a significant decrease in the number of contacts of the lipid tails with A18, which weakens the statement that "lipid pressure" forces the lipids to permeate the crevice despite the label.

Line 475: "This channel response seems to be distinct from curvature-inducing lyso-PC36 activation, which could access the NPs and substitute for the endogenous lipids, but its mode of

action differs from externally applied tension."

-- It is not clear, why substitution of LPC for endogenous lipids would facilitate opening. Do authors suggest they are easier to remove by tension?

Reviewer #3:

Remarks to the Author:

This manuscript reports on a very interesting observation that the authors made in spin labeling of an ion channel. When a labeling site near the entrance to a pocket that can hold a lipid, the protein changes conformation when solubilized in a detergent micelle, but not when reconstituted into a lipid bilayer. EPR distance distribution measurements in the nanometer range reveal that the modification leads to opening of the channel. Significantly, relevance of the effect for protein reconstituted into a planar lipid bilayer could not be confirmed by analysis of current recordings, although there seems to be some difference in channel opening events.

The manuscript is not written up clearly. Until readers arrive at Figure 6, they are led to think that there is evidence for or at least a strong hint to physiological relevance of the results. In fact, the results do support the "lipid moves first" hypothesis to some extent, but suggest a more complex picture and are not conclusive. Given the importance of mechanosensitive ion channels, the study might still be of sufficiently broad interest for Nature Molecular and Structural Biology (rather than Nature Communications), provided that additional experiments or data analysis are performed, as described below. With the present set of results, publication in a more specialized journal can be recommended.

Details:

Major:

1. Spontaneous activity of the channel at ambient pressure does not change when going from WT to the mutant that opens in detergent micelles, according to Figure S6. I miss proper statistical analysis of the current measurements. Figure 6b is only anecdotal. Is there a statistically significant (mathematically describable) difference between spontaneous channel opening events in WT and mutant L89R1? If so, does or doesn't mutant L89W show the same effect? Answering these questions may require additional planar lipid bilayer recordings, but would be a way to make physiological relevance of the effect more likely.

2. If channel behavior in a lipid bilayer at ambient pressure is not affected by the mutation, physiological relevance can only be ascertained by observing channel opening under pressure. The authors would need to devise such an experiment, different leaking of ions from vesicles under osmotic shock, for instance, would suffice. If no differences in spontaneous channel opening at ambient pressure and no difference in pressure dependence can be detected, the finding is still of sufficient interest for publication, but not in a Nature family journal.

3. The authors should be more upfront about what evidence they do and what evidence they don't have. For instance line 550 in the Conclusion is not backed up by the experimental evidence.

Minor:

4. line 181: reduced labeling efficiency does not lead to a double loss, but to a quadratic loss.

5. line 182: I don't understand the argument about measurement of D2. If C5 symmetry (not pentagon symmetry, a pentagon can be irregular) is assumed, D2 can be predicted from D1. In fact, existing software allows for restrained fitting.

6. Figure 2 does not show gating transitions.

7. line 200: How can distance ratio for K100R1 agree with (regular) pentameric structure, if the experimental distribution does not agree with the theoretical one in that the D2 peak is much more strongly sifted towards shorter distances than the D1 peak? This statement needs additional explanation.

8. Were labeling efficiencies quantified? If so, does modulation depth agree with pentamer formation?

9. The caption of Figure 4c does not agree with labelling of the panel (reconstitution into liposomes versus nanodisc). If it is nanodisc, why are the more relevant liposome data not shown? Do they exist? If not, they should be measured.

10. line 470/471: In what way was reversible gating observed?

Reviewers' comments:

We would like to thank all three reviewers for their thoughtful comments and suggestions. We have addressed each comment in full and made substantial changes to the MS including additional experiments, and major changes to the results and discussion. We believe that these changes have significantly improved the quality of the MS.

Reviewer #1 (Remarks to the Author):

In this manuscript, the authors have addressed the effect of hindering lipid chain penetration into nano pockets (NPs) on the structure and function of mechanosensitive (MS) channels. They report that it is possible to generate an allosteric mechanical response on the large conductance mechanosensitive channel MscL in the absence of membrane tension. They present convincing evidence that modification of a single amino acid residue at the entrance of NPs restricts the lipid-acyl chain access to the NPs, mimicking membrane tension, and opens the channel, providing experimental evidence for the 'lipid moves first' model.

This is a thorough investigation that appears to have been performed to a high standard and well presented. The conclusions are based on both electron paramagnetic resonance methodologies (PELDOR, ESEEM, and CW-EPR) and fully atomistic molecular dynamics simulations. The principal findings are novel and should be interesting to a broad scientific audience.

We would like to thank the reviewer for recognising the importance of this study

I have only one minor comment.

1. In Fig 4c L72R1-L89W mutant is said to be in DDM and reconstituted in liposomes. However, in the text (L319) figure itself, the mutant is reconstituted in nanodiscs. Legend or the figure should be adapted, accordingly.

The Legend is now fixed.

We now write: "...following reconstitution in NDs."

Reviewer #2 (Remarks to the Author):

The paper 'Allosteric activation of an ion channel by chemical modification of pressure sensitive nano-pockets' by Pliotas and coworkers raises many more questions than provides answers. Unquestionably, differential solvation of the protein by lipids in two radically different (open and closed) conformations must bear a substantial component of total free energy of transition. Yet, why a small pocket transiently occupied by acyl chains in one of the states is so special is unclear. The presented data suggest otherwise.

We thank the reviewer for their time in reviewing our manuscript and for the useful comments that have resulted in a greatly improved manuscript. To address the reviewer's concerns we have carried out additional experiments (notably a) patch clamp experiments under applied tension and b) detailed energetic MD calculations, in addition to fully atomistic MD simulations in detergent environment, which support our hypothesis. The

new experiments are presented in the revised manuscript and we respond below to each of the specific comments of the reviewer.

1. If the main focus of the work is an allosteric facilitation of channel opening, then the only way to demonstrate the presence of additional energetic contributions besides tension is to record the position of activation (open probability-tension) curves and calculate the transition energy change, or at least the shift of midpoint position.

We thank the reviewer for their comment. We agree this is a key aspect which will improve the manuscript and will allow our structural observations demonstrated by PELDOR to be linked with function (electrophysiology in presence of tension). The observation of an open or closed channel under particular conditions implies a shift in energy threshold.

To address the reviewers comment we performed patch clamp electrophysiological experiments in the presence of externally applied tension and included these data in the revised manuscript (Fig 3b & S6). In brief, these experiments demonstrate a) a significant decrease on pressure activation threshold of the modified (L89R1) mutant compared to WT channel (Fig 3b & S6a) and b) stabilisation of a constitutively subconducting/expanded state in the absence of externally applied tension (or basal lipid glass adhesion tension) (Fig 3b & S6). We feel that these additional experiments and interpretation of the data have improved the quality of the MS by providing solid evidence to further support our original hypothesis. Please also see added text lines 66-68 (Abstract), 271-289 (Results) and 506-509, 534-546 (Discussion).

2. If the goal is to detect some elastic distortion of protein structure in different environments with possible reorientations of sidechains, then any other technique, including PELDOR would be good.

As previously (refs 9, 15), the authors firmly avoid providing (or even discussing) functional data obtained through tests such as patch-clamp, which is standard in the field. Instead, they tend to replace direct functional observations of gating with modeled inter-label distance changes that according to their own estimations are grossly insufficient to support the gating transition in tbMscL.

We are surprised by the reviewer's comment given that in our previous study patch clamp experiments in the presence of tension (in spheroplasts) were conducted for all single -cysteine MscS mutants tested by PELDOR (Fig S2B in Ref 9, *Pliotas et al., PNAS, 2012*). We further performed single channel electrophysiology recordings in presence of Lyso-PC and observed MscS activation (Fig 4c-f, 5c & S5 in Ref 15, *Pliotas et al., Nat Struct Mol Biol, 2015*).

We have now performed patch clamp electrophysiology experiments in the presence of tension (please see previous comment). With the addition of new experimental and computational evidence we believe the MS is now truly comprehensive, by observing highly accurate distances by PELDOR for conformational changes, functional studies by patch clamp (in presence of tension), and molecular dynamics simulations to understand lipid movement (which is difficult to see by any other method). We feel we are using the best and multidisciplinary state-of-the-art techniques that are currently available.

Importantly, patch clamp alone will not give the atomistic detail to the structural change, but indeed only diagnose the functional change. PELDOR is an established state-of-the-art technique for identifying transitions and is linked to protein structure directly by measuring absolute spin-to-spin distances with very high resolution. Full opening under minimal or no tension is not claimed in any part of the manuscript; instead destabilisation of the closed state and the initiation of a sub-conducting state (consistent to the expanded structure observed by PELDOR) to facilitate full channel opening.

3. The major result emphasized in this work is that inter-label distances in L89R1 and in L89W are larger by ~6Å in DDM micelles compared to nanodisks or vesicles. So what? The problem is that the authors did not bother to carefully compare the equilibrium

conformations of tbMscL between DDM micelle and lipid bilayer and determine the rotameric distributions of cysteine-attached MTSSL labels (with the length of sidechain of about 11 Å) in these environments.

This is done to the state-of-the-art. The environment is removed for rotameric modelling, be it detergent or lipid. The data show that all other labelling positions (more than 10) agree with practically identical mean distances between detergent and lipids.

4. The label, which is generally polar, is attached at different locations and how the distance to membrane/micelle interface may affect the inter-label distance distribution is unclear.

The spin-label itself is non-polar and segregates into micelles. The Reviewer suggests that only in the case of L89R1 the spin label adopts a different conformation between lipids and detergent. This would not explain the L72R1L89W and L73R1L89W mutants. Many recent studies from other groups have combined MTSSL spin labelling with PELDOR/DEER spectroscopy (main method used in these studies) and successfully addressed gating of complex membrane proteins, for instance, Martens et al., *Nat Struct Mol Biol* (2016); Verhalen et al., *Nature* (2017); Bountra et al., *Embo j* (2017); Georgieva et al., *Nat Struct Mol Biol* (2013); Timachi et al., *Elife* (2017).

5. The authors provide no analysis of equilibrium between empty and filled states of the pockets. The maximum energy gain from filling the pocket with a single acyl chain is the enthalpic VdW component of interaction with the pocket wall minus the entropic penalty of the highly bent chain conformation. The experimental free energy of ecMscL opening is 58 kBT (Chiang et al., *BJ* 2004), and that for tbMscL it must be larger. It seems unlikely that populating or emptying all pockets provides sufficient driving force for gating. We have presented multiple experimental evidence that the energetics of lipid chain removal is sufficient to induce partial opening.

We have additionally now:

a) calculated the pairwise forces between NPs and acyl chains between the WT and the L89R1 channel (Table S5). 58.9kT has been reported to be required for full channel opening but it is not the minimum energy required to initiate sub-conducting opening transitions (Chiang et al., *BJ*, 2004). The same study suggests the $\Delta E = 58.9\text{kT}$ is the energy required for full channel opening and $\Delta A = 20.4\text{nm}^2$. The minimum observed energy of $\Delta E = 13.7\text{kT}$ in the same study, is sufficient to give rise to an opening of a cross sectional area difference of $\Delta A = 5.8\text{nm}^2$. and we find that the acyl chain interaction with the NPs in the case of the WT channel is ~ 110 times greater compared to modified channel. Interestingly, the total energy difference ΔE is $\sim 18\text{ kJ/mol}$ (equivalent to 7.1kT), and thus similar value to the minimum energy difference required to initiate gating transitions in MscL (Chiang et al, *BJ*, 2004).

Note: in our MD simulations L89R1 does not fully exclude acyl chains from the NPs (Fig 5c), thus the actual energy corresponding to full NP delipidation in nature or with a more efficient exclusion of acyl chains from the NPs molecule should be even higher. The energy difference due to removal of acyl chains from the NPs through covalent modification, is sufficient to give rise to subconducting states (see also previous comments) in the absence of external tension, by destabilising the closed state and initiating an expanded state towards full channel opening ($\sim 3\text{nS}$). Given that Sukharev et al. (1999) have reported 5-6 subconducting states for EcMscL, 10% of the energy required for full opening would be energetically favourable or at very least feasible to initiate/stabilise one of MscL's first subconducting states.

b) changed the Y-axis graph representation from A18-Acyl chain number of contacts to number of single acyl chains within the five NPs of a single MscL pentamer to better reflect the equilibrium between the average number of empty and filled pocket states. Each MscL pocket could fit up to two acyl chains, therefore 10 (5NPs x 2 acyl chains) per pentamer.

For the (30-100)ns interval of the MD simulations and after all acyl chains have intercalated to the NP regions, for WT MscL/DMPC we calculated an average of 5.54 acyl chains per pentamer found deeply within the NPs (i.e. $<5 \text{ \AA}$ distance from A18), and 2.47 for L89R1 MscL/DMPC (Fig 5c). For MD simulations within DDM (new data added to the revised MS), we calculated an average of 4.33 for WT MscL/DDM and 2.86 for L89R1/DDM (Fig S11a), for the same time interval (30-100) ns. Both WT channels in different environment (i.e. DMPC lipids and DDM), contain more than 4 acyl chains on average (more than half of MscL available NPs) within their NPs. On the contrary, L89R1 channels both in DMPC and DDM contain < 3 acyl chains.

We have included all these new findings in the revised MS and added text accordingly in different sections of the MS to accommodate these supportive findings. Please see Fig S11a, Table S5, lines 392-422 (Results), 566-576 (Discussion).

6. PELDOR plots have different ranges in the subplots which makes accurate comparison difficult.

This is to expand only to significant parts of the plots as plotting extensive amounts of distributions in the red colourbar area would be misleading. We could do otherwise but we did this intentionally for clarity and to show all of the data in as much detail as possible. All axes are clearly labelled.

7. Osmotic survival. Moe, Levine and Blount (JBC 2000) have clearly shown that wt tbMscL does not rescue cells from downshock (500 mM NaCl) the way ecMscL does. The viability index in tbMscL experiments is lower by one order of magnitude. Instead of counting colonies, the authors plated droplets of diluted media and 'visually inspected' the growth. This method has not been tested for accuracy and the functional state of mutants remains questionable.

We thank the reviewer for their comment. We agree with the reviewer that this can be a more accurate way of measuring rescue and provides a better defined level of protection to the previous method. We have now repeated all the experiments and observed no major differences to the original data that would require us to alter our initial conclusions. The differences to the initial established protocol we have used are a) a series of dilutions which end up in a larger overall dilution, and b) further plating of the samples subjected to osmotic shock cell on whole plates, not as single drops (initial), according to *Moe et al., JBC, 2000*. These two protocol differences result in significantly fewer colonies, compared to the protocol originally used.

Indeed, and in agreement with *Moe et al. (JBC, 2000)*, we find that TbMscL provides significantly less protection than EcMscL. This is the reason why we have also subjected the cells and reported in our initial assays (and also the revised new ones) to 0.3M NaCl (mild) and not 0.5M NaCl (harsh) osmotic shocks, as both times we have observed that WT TbMscL cannot rescue the cells from harsh osmotic shock. We report the viability for each one of the TbMscL mutants and include WT TbMscL for comparison, all expressed in a strain which lacks EcMscL from its genome. Despite some expected variability of the results, the vast majority of the single cysteine mutants used in our PELDOR studies provide similar, and within error, level of protection to *E coli* cells, when compared to WT TbMscL. In addition, all these experiments were done in quadruplicates ($n = 4$) from at least two different batches of cells including multiple controls and are self-consistent. We have now substituted Fig S1 with a new histogram which includes our new experimental findings. The text has been modified accordingly to reflect these changes, including the new protocol reference, decreased viability of TbMscL compared to EcMscL for *E coli* cells and clarity on the use of a "mild" (0.3M NaCl) shock, lines 173-176 (Results) and 608-617 (Methods).

We now write in lines 173-176: "All 20 TbMscL mutants presented similar protection to WT channel, when experiencing a mild hypo-osmotic shock (Fig S1). The reduced protection

provided by WT TbMscL (compared to EcMscL) to E. coli cells lacking native MS channel genes⁵⁹ is consistent with a previous report⁵⁸

8. Experiments with planar bilayers. There is no use of this method with either MscS or MscL. Activating tensions for these channels are between 5 and 12 mN/m. The authors seem to be unaware that the Gibbs-Plateau border (meniscus) sets constant tension in the planar part of the bilayer between 0.2 and 2 mN/m, depending on solvent and composition (H.T. Tien, 1974).

We thank the reviewer for their comment. We have replaced these data with those from patch clamp recordings in the presence of applied tension, where we observed a significant effect of L89R1 modification both in the pressure activation threshold of TbMscL. We also describe a constitutively open state under these conditions, with an estimated diameter of 9 Å, consistent with the structural state and rearrangement we observe in solution (only for L89R1 in solution). Please see previous comment (1).

More specific points:

9. Line 144: "MS channel activation is lipid removal from the NPs, then opening should occur either by an increase in lateral tension of the linked bulk bilayer lipids or by sterically excluding lipid chain contacts to NP-forming residues, such as A18, adjacent to pore constriction forming L17 (Fig 1a & b)."

-- These possibilities do not exhaust all routes for activation, e.g. redistribution of the lateral tension to more sensitive regions without the net increase in the tension of the "linked bulk bilayer lipids".

We agree with the reviewer and at no point have we excluded other routes. We have an initial hypothesis and we have tested this by multidisciplinary methodology. While it may be that other regions contribute to activation, our hypothesis is that removal of lipids from the NPs is a key step in channel gating. While we cannot disprove the reviewer's hypothesis that other sites may contribute to the gating process (indeed, lateral tension is likely important to the structural stability of most if not all membrane proteins), but we show that exclusion of lipids from NPs by one particular site modification is sufficient to cause large changes in channel gating, whilst 12 others do not (e.g. do not show changes between DDM and lipid environment). These results are supported by functional data and molecular dynamics simulations.

10. Line 201: "These findings suggested that TbMscL preferably adopts a closed conformation in absence of lateral bilayer compression, ..."

-- Considering that the net lateral tension/pressure in a relaxed bilayer is essentially zero, it seems strange that authors throughout the manuscript repeatedly single out only the compressive component (supposedly from the lipid tails region). The pressure in the micelle can be distributed in a different manner compared to bilayers.

We feel that this argument does not in any way invalidate our quoted statement. The pressure in micelle can indeed be distributed in a different manner compared to lipid bilayers.

11. Fig. 2a: - L42R1 produces right shift comparable to the one by the L89R1. L42 is on the periplasmic side, it is unlikely to affect the access to the crevice that situates at the gate level. Surprisingly N70R1 causes even more dramatic right shift, but this is not discussed.

*We have observed these differences, thus we included in our "in lipid" PELDOR analysis these mutants (42, 72, 88 and 89) which showed a discrepancy with *in silico* modelled distances and not L89R1 alone. These mutants either in liposomes and/or nanodiscs, remained unchanged when compared to DDM, with the only exception of L89R1 that returns back to the modelled closed state when reconstituted in either lipid system (liposome and nanodisc). We therefore concluded that 42, 72 and 88 modifications resulted in a closed MscL both in lipid and detergent solution. Further, in the case of 42,*

we have stated that this mutant is located in a very close proximity to a loop region of TbMscL (PDB 2OAR), which has recently been remodeled by another group (Fig 3C & D; *Li et al., PNAS, 2015*). The authors in the latter study, used the original electron density of TbMscL (PDB 2OAR), and produced an improved structural model. Therefore for this site (L42R1), some discrepancies from the modelled distances should be expected and our PELDOR findings are consistent with the *Li et al.*, claim, that reinspection of this loop should lead to a more accurate model for the particular TbMscL region.

12. Line 213: "Interestingly, both measured mean distances of L89R1, were significantly longer than expected by ~ 3 Å (D1) and ~ 5 Å (D2).

-- This enlargement is significantly smaller than would be expected based on the diameter of the conductive pore of MscL (~ 30 Å) or the effective expansion area of MscL in the membrane (~ 20 nm², i.e. ~ 20 Å increase in the external diameter of MscL). The authors estimate ~ 6 Å increase in channel diameter based on the label displacement, which would produce no more than a subconductive state (if any, as the channel is known to open through a non-conductive pre-expansion stage).

Please see also our previous response on comment (5) regarding the energy required to promote subconductance openings. We do not claim we observe a full channel opening as it is predicted from model calculations based on previous electrophysiological measurements, but rather the main focus of the study is to demonstrate there is a structural response leading to destabilisation of the closed state and partial opening (subconducting state) leading towards a fully open state, after the specific effect we cause on channel structure (modification and mutation at the entrance of the NPs). Indeed we now observe a subconducting state of ~ 273 pS (please also see response to comment (1) in agreement with the structural transitions observed by PELDOR.

We have rewritten the relevant sections to indicate that applied tension is still required for MscL to reach its full gating potential (lines 451-453), which is also consistent with our new ephys data.

We now write: in 451-453 "...externally applied pressure is still required to promote mechanical activation of TREK-2¹⁸, and for TbMscL to reach its full gating potential of ~ 3 nS (Fig 3b & S6)."

Notably we now show that this tension required for full opening is much smaller for L89R1 compared to WT channel (Fig 3b). Nevertheless, we also conclude that there is a significant structural and functional (mechanical) response in the presence of minimal background or no tension leading to a state with a pore opening size dimensions similar to the two expanded state x-ray structures previously reported (*Liu et al., Nature, 2009; Li et al., PNAS, 2015*).

13. The discord between the expected and observed distances can be explained by a persistent bias in the label conformation in the real micelle environment compared to the modeled "ideal" distribution for all sterically possible rotamers on a bare channel in vacuum. How do the authors explain distances DECREASE (e.g. K100R1) - the scale is comparable to the one with L89R1, so would it be interpreted as a dramatic compaction of the closed MscL then?

We thank the Reviewer for their comment. We used TbMscL for our PELDOR measurements, because there is a reliable x-ray structure available. Modelled distance distributions were done using state-of-the-art software tools (*Jeschke, G. et al. Appl Magn Reson, 2006; Hagelueken et al., Appl Magn Reson, 2012*). Distances are then directly compared with PELDOR measurements, which report on the same protein construct. For K100, there is indeed a discrepancy (small decrease) when compared to modelled distances. We have initially observed this discrepancy and have therefore reconstituted K100 in liposomes (Fig 3a & S5c) to monitor the effect on this mutant of a lipid environment.

The distances within lipids remain unchanged suggesting no conformational change has occurred, namely the same channel state is adopted for K100R1, both in solution and liposomes. This is in contrast to L89R1, where DDM PELDOR derived distances decrease when L89R1 is reconstituted in lipids, thus suggesting a different channel state (conformational change) between solution and liposomes.

14. Even when the tails are excluded from the crevice by labels grafted in certain positions, the pressure at the region of the crevice does not disappear - it is now applied to the label in those cases, and would be transmitted to the channel.

We thank the Reviewer for their comment. This is at the heart of our findings and we feel of crucial importance. Indeed pressure in lipid bilayers is still transmitted and the label is not capable (as it is not designed for such purpose) to sweep the lipid chains away and the label is not free to enter the NPs in the same way that the lipid acyl chains are.

To better evaluate this effect we have now performed atomistic pairwise energy calculations (Table S5) between the NPs (A18) and acyl chains for the last 20ns (80-100ns) of our MD simulations and compared L89R1 with WT channel. We find that the total energy between the NPs (A18) and lipid chains is $\sim 110\times$ greater for the WT channel. We therefore conclude that tension is indeed transmitted but in the case of the modified channel acyl chains sterically clash or interact dramatically less with A18 rather than in the case of the WT channel, and transmit less tension to channel's pore to gate the channel. MTSSL receives most of the "tension load" and efficiently cuts the link between the lipid bilayers and MscL, consistent with our initial model and hypothesis (see also response to comment (5)).

In detergent micelles, bilayer compression is not present but lipid chains either from endogenous lipids (detected by both solution NMR and ES-MS) present in the samples of detergent penetrate the pockets and keep the channel closed for 12 out of 13 sites tested and spanning all protein domains. The channel is closed in DDM environment for 12 out of 13 sites tested and PELDOR distributions are in high agreement with closed state TbMscL x-ray structure or distances do not shift when the channel is reconstituted within lipids in cases that discrepancies with modelled distances are observed- see also comment (13)). This highlights the crucial role of the location of the modification to achieve channel mechanical response, we feel one of the most important and central findings of our study.

15. Line 245: "solution and within liposome distances for L42R1, indicating the channel remains in the same structural state (Fig 3b & S5b). In contrast, a significant distance shift was observed for L89R1 after reconstitution into both liposomes and nanodiscs (Fig 3b & S5b). The distances shorten and coincide with the closed state modeled distances, suggesting a reversible induced change, from the open to closed state."

-- As an alternative interpretation, it might be that in solution MscL is slightly expanded but not in a bilayer.

Please see previous responses on comments (9) and (12). Our data from multiple sites suggest otherwise, namely that distances, apart from L89R1, are in agreement with the closed state x-ray structure. For 5 different locations (Fig 3A) spanning different domains between DDM and lipids (i.e. liposomes and/or nanodiscs), we observe no changes. If MscL was expanded in DDM solution then we should see some if not all of the sites reversing or shifting back to a closed conformation when reconstituted in lipid bilayers. However, we see no differences and we have obtained almost identical distributions between DDM and lipid bilayers, with the only exception being L89R1, the only mutant that shows an increase in pore diameter of ~ 6 Å in DDM and equal decrease in lipids (both NDs and Liposomes), returning to its closed state.

16. Line 290: "In one out of four 15 min long recordings for both the WT and L89R1 mutant channel spontaneous openings at +30 mV were observed with open probabilities of 0.27 for the WT and 0.04 for L89R1. The main open state level of these openings was identical

for both WT and L89R1 channels, with a measured current amplitude of ~2.2 pA, resulting in a similar calculated chord conductance of ~70 pS (Fig S6)."

-- The electrophysiological recordings are very confusing - extremely low conductance (70 pS vs. 3 nS known for MscL) and somehow very high estimated open probability. This does not seem to be MscL.

The reason for this is that the pressure is too low to open the channel. We have for this reason now set up and performed patch clamp experiments in the presence of applied tension, allowing us to directly compare the pressure activation threshold of L89R1 with WT TbMscL, which was found to be significantly reduced. We have replaced these data with Fig 3b and S6 in the revised MS. For text changes and more information regarding new patch clamp data please also see response on comment (1) and other previous comments.

17. Line 305: "Distance distributions of N70R1-L89W and K100R1-L89W in solution did not significantly differ from N70R1 and K100R1 (Fig 4a & S7a) respectively and the same applied to K100R1-F88W (Fig S7b). However, time-domain traces for L72R1-L89W and L73R1-L89W in solution revealed very short distances, below the distance regime commonly accessible by PELDOR (<18 Å) (Fig 4a & b), corresponding to a large decrease for L72R1 and L73R1 (D1) of > 7 Å and 12 Å, respectively. This may arise from a helix rotation that brings residues facing towards the channel's pore, consistent with the relative changes of equivalent residues of MaMscL, where a TM2 helical rotation and expansion occurs (absolute distances differ due to different constructs) (Fig S8)."

-- From TbMscL structure it appears that to bring both L72 and L73 inside the pore, the helix would have to rotate by ~180 degrees, which is much more than the rotation observed in MaMscL (closer to 90 degrees). More importantly, if there were indeed channel opening and TM2 rotation caused by L89W mutation, it should have revealed itself in N70R1 and K100R1 mutants as well - otherwise, the conformational change can be interpreted as some sort of kink or uncoiling in TM2.

We thank the reviewer for their comment and for giving us the opportunity to clarify this very important point for understanding structural transitions on TbMscL. We performed and included *in silico* spin labelled modelled distances of equivalent to TbMscL K100 residues e.g. orientation in respect to channel pore axis (MaMscL Q94 and K97) and channel domain proximity (Fig S8b). We showed that modelled distances in MaMscL, despite the large and complex conformational changes, remained unchanged between the two conformations for both mutants (Fig S8b). Further we included in our analysis MaMscL F78, equivalent to TbMscL L72 residue. For this site, modelled calculated distances both significantly decreased, despite pore channel expansion and TM2 tilt and rotation (Fig S8a & b). A similar decrease was observed by PELDOR for TbMscL, when the combined L89W mutation was introduced to sites e.g. L72 and L73. For another site on similar part of TM2 e.g. N70, distances remained mostly unchanged. Below we present a model which is consistent with experimental PELDOR distances and fits all four modified sites e.g. N70, L72, L73, K100, and included new figures (in the revised MS) to provide further clarity.

We have now added Fig S8c to show the initial side chain orientation of N70, L72 and L73 TbMscL residues. Further, we have added Fig S8d & e. In these figures we show that TM2 rotation during transition between the closed and the expanded x-ray structures of MaMscL F78 and K97 residues (equivalent to TbMscL L72 and K100 respectively), forms an angle >100° (but smaller than 180° for both cases). This angle is close to 90°, as the reviewer pointed out, but still significantly larger and for some residues could reach a value of ~120°. Taking this observation into account which is based on the MaMscL x-ray structures, we then added a new schematic in which we now show that TbMscL L72 and L73 could be pointing inside the pore (Fig S9a), resulting in a reduction of measured distances, while N70 (Fig S9b) distances could remain essentially unchanged as a consequence of a ~110°-120° occurring TM2 rotation during opening transition, similar to MaMscL x-ray models. Although we cannot accurately calculate the exact angles since the short

distances (expanded TbmscL state) are out of PELDOR distance measurement range and that we cannot exclude other scenarios as the helical movement is complex (helical tilt and rotation and pore expansion), as rightly pointed out by the reviewer, we present a case (Fig S9), which is realistic, and consistent with MaMscL models given the relative orientation of the TbMscL residues (Fig S8), as well as all our experimental EPR observations (PELDOR, CW and ESEEM, Fig 4). Therefore we believe these new additions offer clarification to crucial previous PELDOR results and clarify and provide extra structural and molecular insights in the gating transitions and mechanical responses of TbMscL, thus overall improving the current MS, given these calculations, observations and presented scenarios agree and are not contradicted by our PELDOR findings.

We have modified the text now accordingly to included new figures and clarifications arising from this response. We now write in lines 304-307:

"This could arise from helical rotation (TM2) that brings residues facing towards the channel's pore (Fig S9), consistent with the relative changes of equivalent MaMscL residues, where a TM2 rotation accompanied by pore expansion occurs (absolute distances differ due to different constructs) (Fig S8 & 9)."

Further we now write in lines 346-350: *"Although accurate determination of these short distances is not possible with PELDOR, and given the complex TM2 movement (tilt and rotation; MaMscL x-ray models, Fig S8), the observed distance changes for all four sites (N70, L72, L73 and K100) combined with L89W are consistent with an anticlockwise TM2 rotation in addition to pore expansion (Fig S8 & S9)."*

18. Line 321: "To exclude the possibility that short distances arise from either protein unfolding, dissociation or aggregation, L72R1-L89W and E102R1 (control) samples were thawed after PELDOR and run over SEC and both eluted as folded pentamers (Fig S7c)."
- Why were these specific mutants chosen for the folding test rather than L72R1-L89W and L73R1-L89W which actually had the change?

We thank the Reviewer for their comment. We have indeed chosen L72R1-89W because this was one of the mutants (out of two), which showed a change in addition to a mutant (i.e. 100R1) which others did not. To this end, and in agreement with reviewer's suggestion, we have chosen two representative samples, one which shows expected closed pentameric distances 102R1 (control) and one (i.e. L72R1-89W) out of the two which showed a change e.g. significant decrease in their distances. Both mutants elute at the same volume, are monodisperse and folded, consistent with a pentamer and with no signs of dissociation (Fig S7c).

19. Line 367: "An increase in solvent exposure (3pESEEM) and spin label mobility (room temperature CW368 EPR) was observed for 3Q-Y87R1, compared to 87R1, as expected for a less hydrophobic environment and consistent with local lipid removal (Fig S9a & b). Previously, NMR and ES370 MS detected endogenous phospholipids in the detergent purified protein samples (solution), which would not be able to bind after neutralization of the charged cluster (Fig 3a & S5a, Table S3). However, these discrepancies in solvent accessibility and label mobility did not correlate with distance changes (Fig S9c), suggesting MscL state is unaffected by lipid headgroup binding."

-- Would the position and/or presence of the lipid tails in the crevice be affected by the headgroup unbinding as well? Why don't just test the activation threshold of L89W or L89R1 in patch-clamp?

We have now tested the activation threshold in patch clamp recording and we see a substantial effect (Fig 3b and S6). See also response to comment (1) and other previous responses.

20. Line 409: "Several R1 rotamers (L89R1), efficiently "guarded" the NP entrance and restricted access to protruding inner-leaflet lipid-chains by opposing steric clashes, not

observed in the WT channel (Fig 5a,b & Vid S1). This further supports the PELDOR, ESEEM and CW-EPR experimental data suggesting L89R1's mode of action is via disruption of NP acyl chain penetration."

-- While the decrease in the contacts between the lipid tails and A18 can be clearly seen in simulations and the overall approach in MD looks adequate, it is only a minimal and not the most crucial information that can potentially be extracted from these MD simulations. It is not surprising that the bulky side chain at L89 location would restrict access of lipid tails to the crevice. More interesting question is whether there are ANY signs in simulation that it would facilitate channel opening. Obviously, one cannot expect full opening on the scale of 100 ns, however, some small changes might be already noticed, e.g.: is there any widening of the pore at the level of the gate? Is there a continuous contact A18--R1--lipids (i.e. whether the pressure from lipid tails is still transduced to the gate, just mediated by R1) or there is an empty crevice around A18? Any signs of increased tension at the polar MscL residues at the interfacial levels or any other significant pattern of tension redistribution? The latter can be estimated by postprocessing of the simulation traces either in software that allows 3D force mapping (it was even applied to MscL simulations before, e.g. see Ollila OH et al. Phys Rev Lett. 2009; or Vanegas JM, Arroyo M. PLoS One. 2014) or at least it can be analyzed using VMD/NAMD plugin for estimation of pairwise forces and energies between various selections (e.g. lipid/water medium and specific residues on MscL surface. So far, it is just an assumption that the observed change in contacts would favor channel opening.

We thank the reviewer for their comment. We have now calculated all pairwise forces involved and included them in Table S5 (and Methods). We have also amended the results section to accommodate these new calculations and modified the MS accordingly. We have calculated: a) NPs to Acyl chain, b) NPs to 89 (Leu -WT and 89R1 -MTSSL) and c) Acyl chain to 89 (Leu and MTSSL) total pairwise energies (total, Lennard-Jones, and Coulomb), differences and ratios. The values are in complete agreement with the hypothesis that L89R1 modification takes "all the load" of the bilayer lipid acyl chains attempting to enter the NPs without "transmitting" this energy to the NPs even if when MTSSL is being "pushed" by acyl chains. This results in a calculated ~110x (Table S4) greater energy transmission of the acyl chain to the NPs in the absence of the modification (MTSSL) forcing the channel to adopt an energetically-favourable closed state, in the case of the WT protein. In particular we observe no relative difference in total energy between L89 or L89R1 and the NPs and the same is valid for acyl chains and L89 or L89R1. Therefore, both L89 and L89R1 interact "similarly" with each of the NPs or the bilayer acyl chains, whereas the bilayer acyl chains "sense" huge difference changes onto the pairwise total energy/force in respect the NPs when the modification is present or absent. This is consistent with a model that the modification "interrupts" the very final contacts between the lipid and the NPs, thus effectively cutting the link between the bilayer and the channel, only in the case of L89R1, mimicking the effect of bilayer tension. These values and calculations are in complete agreement with our proposed model and fit very well with the rest of our findings. Please also see previous comments regarding pairwise energy calculations.

21. Line 420: "We have demonstrated that disruption of lipid chain penetration within NPs allosterically generates an MS channel structural response."

-- That statement is misleading. Most of the effects are observed on solubilized channel (an environment with rather unnatural pattern of external forces), whereas in bilayer setting MscL tended to be in the crystal-like closed conformation. Even in solution, the single-residue modification experiments (like L89R1_) cannot exclude that the conformational change in the channel was caused by the environment rather than modification as the label both induces (allegedly) conformational change and reports on it. The two-point modifications are contradictory as L72R1-L89W and L73R1-L89W indicate a dramatic helix rotation (or compaction of the pore), whereas other nearby residues remain in positions

similar to the crystallographic closed state. MD simulations showed exclusion of lipid tails from contacts with A18 but provided no evidence for any allosteric change in the channel. We respectfully disagree with the Reviewer and argue that our results are entirely consistent with the "lipid moves first" model hypothesis. Please see our previous responses on comments (9), (12), (13), (15), (17) and others. If it was the detergent environment in the case of MscL, opening or distance changes should have been observed in some or all of the other 12 sites, which we have also tested in lipid bilayers and observed no changes. Mutation to tryptophan (W) of the same site caused similar effects on MscL structure reported by PELDOR on upper part TM2 helix.

22. Line 425: "Elastic energy was restored when lateral compression was re-introduced by reconstitution into lipid bilayers thereby forcing lipid chains back into the NPs and closing MscL."

-- Authors persistently describe the mechanical effect of the resting bilayer as only "pressure", thus ignoring well-established notion that there is equally large component of a lateral tension at the polar/nonpolar interfacial level in the bilayer, which balances the pressure and undoubtedly plays mechanical role too - just acting on other set of residues. It is better to be more precise in the expressions. The authors make no distinction between the elastic energy of the protein and that of the membrane, making their arguments throughout the paper difficult to follow.

We thank the Reviewer for their comment and apologise for lack of clarity on this aspect. We have now specified that it is the protein elastic stored energy. Lines, 432, 444, 542 and 563 (Discussion) have been changed accordingly in the revised MS.

23. Line 427: "In contrast, all MscS mutants previously tested adopt an open conformation in solution."

-- The true conformation of an open MscS is far from being a settled topic.

We thank the reviewer for their comment. We have now modified the text (lines 444-445) to reflect this point. In particular, we now cite structures that were solved by different labs and assigned as being in the open state. We further now clarify that these were obtained under certain crystallisation conditions. Finally, we cite our previous studies performed in solution, in which MscS was assigned open in solution supported by PELDOR measurements and confirmed by x-ray crystallography.

We now write in lines 434-435: "*In contrast, MscS was previously assigned to adopt an open conformation in solution^{9,15,39} and certain crystallization conditions^{12,14}*"

24. Line 455: "In the absence of lateral compression, the MTS-modification introduces steric clashes that prevent acyl chains belonging either to detergent or endogenous lipids from entering the NP. In doing so, the energy activation barrier is reduced and elastic energy drives MscL opening (Fig 6). In the presence of high lipid compression, the acyl chains force the label to adopt an alternative conformer that permits acyl chain access to the NPs (Fig 6)."

-- The presented data from MD simulations do not match this statement. There were no simulations in detergent, so the effect on the tails is just an assumption based mostly on spectrometry data. For the bilayer, simulations show a significant decrease in the number of contacts of the lipid tails with A18, which weakens the statement that "lipid pressure" forces the lipids to permeate the crevice despite the label.

We thank the reviewer for their comment. We have now performed and added in the revised MS full atomistic simulations of TbMscL in DDM for both WT and L89R1 proteins (Fig S11a). Please see also our response to comment (5) for more information. We have also now modified our representation to make it clearer to the reader and we use number of single lipid chains within each 5-mer, e.g. 5NPs per MscL (Fig 5c & S11a). We observed that in total 10 single acyl chains (2 per NP) could fit in the NPs of a single MscL pentamer. There is a significant decrease in acyl chain number within the NPs, for L89R1 also in DDM

(as it was also the case for DMPC), which supports our model and initial hypothesis as it shows that the modification on the L89 site has an effect on lipid chain penetration to the NP.

We apologise to the reviewer for lack of clarity of this point and thank them for giving us the opportunity for further clarification, which is central to our model. The MD simulations were performed in zero bilayer compression, whereas our statement “in the presence of high lipid compression” refers to PELDOR experimental conditions (either nanodiscs or liposomes), where lipid bilayer compression exists. We have now amended line 467 to reflect this change.

We now write in lines 466-468: “*In the presence of high lipid compression, present in our PELDOR experimental set up, the acyl chains force the label to adopt an alternative conformer that permits acyl chain access to the NPs (Fig 6).*”

Our rationale behind setting up the MD simulations was to observe the effect of modification on lipid chains in the absence of applied artificial force fields. We were not expecting to activate TbMscL under these conditions (short time interval of simulations). We predicted that if these contacts would last significantly longer, then the channel structure should respond and we should see openings MD simulations consistent with our patch clamp recordings and the stabilisation of a sub-conducting state.

25. Line 475: "This channel response seems to be distinct from curvature-inducing lyso-PC36 activation, which could access the NPs and substitute for the endogenous lipids, but its mode of action differs from externally applied tension."

-- It is not clear, why substitution of LPC for endogenous lipids would facilitate opening. Do authors suggest they are easier to remove by tension?

We have not performed any measurements with LPC in the present study and have not included or attempted such data. We however refer our previous study in which LPC has been applied and stabilised an open/conducting state for MscS, in PLB recordings and in the absence of any external applied pressure (or in the presence of the Gibbs/Plateau tension level) (Fig 4c-f, 5c & S5 in *Pliotas et al., Nat Struct Mol Biol, 2015*). MscS has wider and larger NPs and much lower pressure activation threshold than MscL, thus we have now set up patch clamp recordings and included data in the revised MS as suggested by the reviewer.

Reviewer #3 (Remarks to the Author):

This manuscript reports on a very interesting observation that the authors made in spin labeling of an ion channel. When a labeling site near the entrance to a pocket that can hold a lipid, the protein changes conformation when solubilized in a detergent micelle, but not when reconstituted into a lipid bilayer. EPR distance distribution measurements in the nanometer range reveal that the modification leads to opening of the channel. Significantly, relevance of the effect for protein reconstituted into a planar lipid bilayer could not be confirmed by analysis of current recordings, although there seems to be some difference in channel opening events.

The manuscript is not written up clearly. Until readers arrive at Figure 6, they are led to think that there is evidence for or at least a strong hint to physiological relevance of the results. In fact, the results do support the “lipid moves first” hypothesis to some extent, but suggest a more complex picture and are not conclusive. Given the importance of mechanosensitive ion channels, the study might still be of sufficiently broad interest for Nature Molecular and Structural Biology (rather than Nature Communications), provided that additional experiments or data analysis are performed, as described below. With the present set of results, publication in a more specialized journal can be recommended.

We thank the reviewer for their time in reviewing our manuscript and for the useful comments that have resulted in a greatly improved manuscript. To address the reviewers concerns we have carried out additional experiments (notably patch clamp experiments under applied tension and detailed energetic MD calculations in addition to fully atomistic MD simulations in detergent environment, which further support our hypothesis. We believe these additional data (in particular the patch clamp single molecule functional assessment) demonstrates the physiological relevance of our study. The new experiments are presented in the revised manuscript, in which certain parts have been amended and we respond below to each of the specific comments of the reviewer.

Details:

Major:

1. Spontaneous activity of the channel at ambient pressure does not change when going from WT to the mutant that opens in detergent micelles, according to Figure S6. I miss proper statistical analysis of the current measurements. Figure 6b is only anecdotal. Is there a statistically significant (mathematically describable) difference between spontaneous channel opening events in WT and mutant L89R1? If so, does or doesn't mutant L89W show the same effect? Answering these questions may require additional planar lipid bilayer recordings, but would be a way to make physiological relevance of the effect more likely.

We thank the reviewer for their comment. We agree this is a key aspect which will improve the manuscript and will allow our structural observations demonstrated by PELDOR to be linked with function (electrophysiology in presence of tension). The observation of an open or closed channel under particular conditions implies a shift in energy threshold.

To address the reviewers comment we have now performed patch clamp electrophysiology experiments in the presence of externally applied tension and included these data in the revised manuscript (Fig 3b & S6). In brief, these experiments demonstrate a) a large decrease on pressure activation threshold of the modified (L89R1) mutant compared to WT channel (Fig 3b & S6) and b) stabilisation of a constitutively subconducting/expanded state of 273pS in the absence of externally applied tension (or basal lipid glass adhesion tension) (Fig 3b and S6). In particular, the expanded L89R1 TbMscL state stabilized in solution consists of 8-10 Å total diameter at the narrow constriction site, accounting for the 6 Å diameter increase reported by PELDOR (in addition to 2-3 Å pore diameter accounting for the narrow constriction site), an opening sufficient for water and ions to pass through. The baseline threshold sub-conducting 273pS state evident in our L89R1 channel recordings in the absence of externally applied tension is consistent with a ~9.2 Å pore diameter estimated opening (*Cruickshank et al., BJ, 1997*). Interestingly, the pore size expansion of L89R1 that is experimentally measured by PELDOR (in the absence of bilayer compression; in solution) and estimated from our single channel recordings (in the absence or minimal lipid-adhesion tension; in lipids) are in complete agreement.

We feel that these additional experiments and interpretation of the data have improved the quality of the MS by providing solid evidence to further support our original hypothesis. Please also see added text lines 66-68 (Abstract), 271-289 (Results) and 506-509, 534-546 (Discussion).

We believe that these new data (particularly the functional single molecule data), provide substantial further functional evidence for the validity of the lipid moves first model/hypothesis.

2. If channel behavior in a lipid bilayer at ambient pressure is not affected by the mutation, physiological relevance can only be ascertained by observing channel opening under pressure. The authors would need to devise such an experiment, different leaking of ions from vesicles under osmotic shock, for instance, would suffice. If no differences in spontaneous channel opening at ambient pressure and no difference in pressure

dependence can be detected, the finding is still of sufficient interest for publication, but not in a Nature family journal.

We refer the reviewer to previous response on comment (1). Patch clamp electrophysiology experiments have now been performed and included in the revised MS (Fig 3b & S6), showing a) large effect on pressure threshold of modified mutant and b) stabilisation of a constitutively subconducting/expanded state of ~273pS in the absence of externally applied tension (or lipid glass adhesion tension; *Opsahl & Webb, Biophys J, 1994*).

3. The authors should be more upfront about what evidence they do and what evidence they don't have. For instance line 550 in the Conclusion is not backed up by the experimental evidence.

We thank the Reviewer for their comment. Line 550 (now lines 577-579 in the revised MS) has been changed. Please also see response to comment (1) for new functional insights included in the revised MS, which along with additional detailed energy calculations (Fig S5) further support our model/hypothesis.

We now write (line 577-579): "*We have shown that hindering lipid acyl-chain access within defined transmembrane hydrophobic crevices leads to a coordinated response in the structure, which is directly linked to function of a mechanosensitive ion channel*".

Minor:

4. line 181: reduced labeling efficiency does not lead to a double loss, but to a quadratic loss.

Indeed, but multi-spin systems are mathematically more intricate.

We now write in lines 187-189: "*For distance measurements, low labeling reduces both EPR signal intensity and modulation depth (dipolar coupling) leading to a quadratic loss (this is only strictly true for systems with 2 spin labels⁴⁵)*"

5. line 182: I don't understand the argument about measurement of D2. If C5 symmetry (not pentagon symmetry, a pentagon can be irregular) is assumed, D2 can be predicted from D1. In fact, existing software allows for restrained fitting.

D₂ acts as a control that C5 symmetry is indeed retained. We are aware of the work which describes the existing software allowing for restrained fitting (*Dalmas et al., JACS, 2012*), but we did not want to bias analysis (by constrained fitting), but to use D₁/D₂ as a control as modulation depths are consistent but not unequivocal.

6. Figure 2 does not show gating transitions.

We thank the reviewer for their comment. Figure legend in the MS has now been changed. We now write: "*Structural state of TbMscL observed by PELDOR distance measurements in solution.*"

7. line 200: How can distance ratio for K100R1 agree with (regular) pentameric structure, if the experimental distribution does not agree with the theoretical one in that the D2 peak is much more strongly sifted towards shorter distances than the D1 peak? This statement needs additional explanation.

We added the following comment to the bracket in lines 200-204:

"*We note that for a given time window D2 will be more prone to instability with respect to background correction than D1 mostly reducing the long distance contribution. This may shift ratios to smaller numbers. This will mainly be relevant in the amber and red regions of the distance distributions (see also colorbars in Fig. 2)^{39,63}.*"

8. Were labeling efficiencies quantified? If so, does modulation depth agree with pentamer formation?

We thank the reviewer for their comment and for giving the opportunity to clarify this point. Assuming a single modulation depth parameter for mutants and labelling degrees in DDM this can be fitted to 0.46. This is higher than expected, but it is known that dipolar echo modulation can increase the effective modulation depth. The modulation depths indicate a fivefold or higher multimerization degree. It may also indicate that our labelling degrees are underestimated, given MscL's extremely low extinction coefficient due to the complete lack of tryptophans within its native form. When back-calculating the labelling degree from this modulation depth parameter and the experimental modulation depth the trends agree, even though the exact numerical agreement is not achieved (as commonly observed in membrane proteins).

We now include a table with modulation depths for selected mutants (Table S3).

We now write in lines 204-205 of the revised MS: "*The experimental modulation depths are consistent with pentamer formation though the multimeric state could not be shown unequivocally (Table S3).*"

9. The caption of Figure 4c does not agree with labelling of the panel (reconstitution into liposomes versus nanodisc). If it is nanodisc, why are the more relevant liposome data not shown? Do they exist? If not, they should be measured.

We thank the reviewer for their comment and for giving us the opportunity to clarify this point. Figure legend in the MS has now been changed and we apologise for this typing error. We have reconstituted 89W/XR1 double mutant in nanodiscs and recorded PELDOR, CW and ESEEM spectra, since this lipid environment yielded significantly better quality data than liposomes (i.e. more reliable distance distributions and longer time windows), as observed in our previous measurements for single TM mutants recorded in both lipid environments (i.e. 88R1 and 89R1, Fig 3a).

We now write in 4c legend: "*...single L72R1 and double L72R1-L89W mutant in DDM solution and following reconstitution in NDs.*"

10. line 470/471: In what way was reversible gating observed?

It has now been removed and new text added.

We now write in lines 480-481 : "*...only L89R1 distances shifted significantly, reversing back to the modelled closed state distances, whereas all other mutants were unchanged (Fig 2a & 3a).*"

Reviewers' Comments:

Reviewer #2:

Remarks to the Author:

The revised version of the paper "Allosteric activation..." by Pliotas and coworkers still raises a number of concerns due to the disparity between the factual material, its presentation, and interpretation.

1. It has been previously established that a 2.5-3.2 nS open MscL pore can be created by a concerted tilting transition of both TM1 and TM2 helices associated with a lateral displacement away from the axis of the pore by 9-13 Å. This occurs under tension of 10-14 mN/m (for EcMscL). This tension is transmitted primarily through polar-apolar boundaries inside the membrane and therefore is applied mainly to the ends of TM helices. With the opening, the in-plane area of the EcMscL increases by ~20.4 nm², and the energetic cost of the opening is 58 kT (~125 kJ/mol). This putative conformation is called open state. The higher activation midpoint for TbMscL indicates that the transition energy is even higher, thus challenging many aspects of the proposed mechanism. The scale of transition and the energetics have to be acknowledged in the text.

2. In the rebuttal, the authors say: 'We do not claim we observe a full channel opening as it is predicted from model calculations based on previous electrophysiological measurements, but rather the main focus of the study is to demonstrate there is a structural response leading to destabilisation of the closed state and partial opening (subconducting state) leading towards a fully open state, after the specific effect we cause on channel structure (modification and mutation at the entrance of the NPs)'. This has to be clearly stated in the text. Instead, we read:

(L 135-137) We predicted that this disruption would prevent acyl chain occupancy in the NPs causing an allosteric MS channel structural response resulting in channel activation in the absence of applied membrane tension.

(L 252-255) In contrast, a significant distance shift was observed for L89R1 after reconstitution into both liposomes and nanodiscs (Fig 3a & S5c). The distances shorten and coincide with the closed state modeled distances, suggesting a reversible induced change, from the open to closed state.

(L 487-489) To exclude any artifacts from spin label clashes with neighboring residues and/or lipids, a tryptophan was introduced at this site (L89W) to trigger opening and paired with cysteine mutants on individual sites to report on conformational changes.

These statements are misleading and should be properly reworded throughout the text.

The expression 'in solution' should be replaced with 'in DDM micelles' throughout.

3. Using PELDOR experiments with TbMscL, the authors observe that when cysteines in position 89 are modified with MTSSL in DDM micelles (~60% of sites labeled in the population), then the D1 and D2 distances between spin labels in pentamers increase by 3 and 5 Å, respectively, compared to the computational prediction based on 2OAR crystal structure of TbMscL. This corresponds to approximately 2.7 Å radial expansion inside the cytoplasmic leaflet of the membrane where the labels are attached. This is 3-4 times smaller than the fully open conformation predicts and must be clearly stated in the text. Importantly, this small expansion caused by chemical modification is observed only in micelles, whereas in liposomes and nanodisks the distances between labels correspond to crystallographic (closed-state) positions. It is very possible that the chemical modification of cysteines in position 89 with a bulky adduct somehow 'loosens' the structure specifically in micelles, perturbs the helical packing and even produces a leak in the structure. There is no reason to call this result of covalent modification allosteric opening.

The difference in distances between the two environments raises questions about the preferred conformations of the flexible MTSSL group in DDM micelles versus lipid bilayers characterized with strong anisotropy of chain orientation. The nitroxide moiety has the freedom to occupy many conformations, and in the extended conformation, the paramagnetic oxygen can protrude for 10 Å away from the alpha carbon of the labeled cysteine. The system clearly calls for a special

comparison of nitroxide position in each specific environment present explicitly, but for some reason, the conformer statistics was done without the environment. Moreover, 100-ns long simulations of TbMscL were done in both lipids and DDM, but no analysis of preferred label conformers in these two different environments is presented.

4. The only functional results on activation and conductive properties of L89R1 – labeled channels are presented in Fig. 3b. ‘Representative’ traces are shown. The uncertainty in these recordings is the size and curvature of GUV patches, which were not visualized but can vary drastically. Please refer to Moe and Blount (Biochemistry 2005) for the correct way of GUV patch-clamp data presentation and analysis.

We do understand the enormous difficulties of experimentation with TbMscL which activates at tensions exceeding patch stability. Is there a way around it? Based on their main hypothesis, in Fig. 6 the authors give interesting predictions: modification of C89 with a bulky MTS reagent should lower activating tension, whereas reduction of the MTS should restore access of lipids to the pocket and stabilize the closed state, i.e. shift activation curve back to the right. The effect of L89W substitution is predicted to be essentially the same as MTS modification. If this is true, then the following experiments will provide much stronger support to this inference than we find in the paper.

a. It would be important to show whether or not incubation of L89-R1 TbMscL containing liposomes with a reducing agent (mercaptoethanol, DTT) reverses the gating presented on the right side of Fig. 3b.

b. If the hindrance of aliphatic chain penetration at the entrance really changes the channel activation threshold or midpoint, then any other bulky MTS reagent should do the same. The authors have a broad choice from apolar (benzyl or long alkyl methanethiosulfonate) to polar uncharged or charged sidechains. It would be important to compare their functional effects.

c. L89W is predicted to have the same effect as MTSSL due to its ability to interfere with acyl chains at the entrance to the pocket. What is the tension midpoint of this mutant in spheroplasts relative to WT TbMscL?

Our concluding comment is about the main hypothesis of whether the lipid occupancy of TbMscL pockets can drastically change the behavior of this molecule. The dynamics of TM1 helices forming the gate is defined by (1) tight TM1-TM1 packing mediated by alternating glycines and (2) packing and dynamics of the outer TM2 helices around the TM1 barrel. The TM1 helices are apparently well constrained by the surrounding helices and are not free to move. The pressure of lipids is transmitted to the gate primarily through the lipid-facing helices and only to a small degree through the direct contact with lipid chains in the pocket (based on the gross contact area). Only if the gate-bearing TM1 helices were free to move and could somehow expand inside the TM2 sheath, then the lipid ‘stuffing’ in the pockets could possibly restrain it. To examine the significance of the presence of aliphatic tails in the pockets computationally, the authors should quantify lateral mobility of TM1s in the full bilayer with and without lipid chains in the pockets. Before this is done, it is impossible to assess the value of this hypothesis. Intuitively, the net pressure/tension of surrounding lipids will be transmitted pretty much the same way through the helical frame of the protein no matter what is inside the pockets located near the midplane of the membrane.

Reviewer #3:

Remarks to the Author:

This manuscript has been substantially improved. In particular, the patch clamp electrophysiological measurements in presence of tension (Figure 3b) answer my previous major

concern about relevance of the observations. Other statements that were not sufficiently supported have been removed or altered. The conclusions that are not sufficiently supported are of insufficient broad interest for publication in Nature Communications.

Reviewers' comments:

We would like to thank the reviewer for their thoughtful comments and suggestions. We have addressed each comment in full and made substantial changes to the MS including additional experiments and significant changes to the results and discussion. We believe that these changes have significantly improved the quality of the MS.

Reviewer #2 (Remarks to the Author):

The revised version of the paper "Allosteric activation..." by Pliotas and coworkers still raises a number of concerns due to the disparity between the factual material, its presentation, and interpretation.

To address the reviewers comment we performed patch clamp electrophysiology experiments for L89R1 under tension, in the presence of a reducing agent (DTT) and included these data in the revised manuscript (Fig. 3b and S6a and associated legends, and text, L68-9, L271, L292-4, L768-70). In brief, these experiments demonstrate substantial reversibility on L89R1's pressure activation threshold e.g. similar to WT levels, when the former is incubated with DTT. This suggests that the modification on this unique site is responsible for the activation threshold decrease, rather than the mutation itself on this site. New findings are strongly supportive of our initial model and the "lipid moves first" hypothesis.

1. It has been previously established that a 2.5-3.2 nS open MscL pore can be created by a concerted tilting transition of both TM1 and TM2 helices associated with a lateral displacement away from the axis of the pore by 9-13 Å. This occurs under tension of 10-14 mN/m (for EcMscL). This tension is transmitted primarily through polar-apolar boundaries inside the membrane and therefore is applied mainly to the ends of TM helices. With the opening, the in-plane area of the EcMscL increases by ~20.4 nm², and the energetic cost of the opening is 58 kT (~125 kJ/mol). This putative conformation is called open state. The higher activation midpoint for TbMscL indicates that the transition energy is even higher, thus challenging many aspects of the proposed mechanism. The scale of transition and the energetics have to be acknowledged in the text.

We now write (L101-104): "An energetic cost of 58kT²³ is required in order for its non-selective pore to reach an estimated full channel diameter opening of ~28 Å²⁴, due to a lateral displacement away from the pore axis by 9-13 Å²⁵, resulting in a conductance of 3 nS²², ..."

We now write (L111-116): ~12-14 mN/m, significantly higher than either MscS (7 mN/m) or Piezo (4mN/m) and an order of magnitude higher than TRAAK^{7,29,30}. This is consistent with MscL's role as the bacteria's last resort mechanism against osmotic shock. TbMscL requires a significantly higher tension to be applied for full opening of its pore (~2x) compared to EcMscL, and reach a similar total conductance of ~3nS³¹."

We now write (L231 – 233): "...of the inner-diameter at the cytoplasmic leaflet channel domain of ~6 Å, a significant opening and approximately a third of an estimated full channel opening^{24,25}."

2. In the rebuttal, the authors say: 'We do not claim we observe a full channel opening as it is predicted from model calculations based on previous electrophysiological measurements, but rather the main focus of the study is to demonstrate there is a structural response leading to destabilisation of the closed state and partial opening (subconducting state) leading towards a fully open state, after the specific effect we cause on channel structure (modification and mutation at the entrance of the NPs)'. This has to be clearly stated in the text. Instead, we read: (L 135-137) We predicted that this disruption would prevent acyl chain occupancy in the NPs causing an allosteric MS channel structural response resulting in channel activation in the absence of applied membrane tension.

We now write (L141-142): "... an allosteric MS channel structural response resulting in the destabilisation of the closed state and partial opening (subconducting state) of the channel, towards a fully open state, ..."

(L 252-255) In contrast, a significant distance shift was observed for L89R1 after reconstitution into both liposomes and nanodiscs (Fig 3a & S5c). The distances shorten and coincide with the closed state modeled distances, suggesting a reversible induced change, from the open to closed state.

We now write (L261): "... from the partially open to closed state."

(L 487-489) To exclude any artefacts from spin label clashes with neighboring residues and/or lipids, a tryptophan was introduced at this site (L89W) to trigger opening and paired with cysteine mutants on individual sites to report on conformational changes. These statements are misleading and should be properly reworded throughout the text.

We have now removed: "to trigger opening" and we now write (L483-484): "... , a tryptophan was introduced at this site (L89W) to trigger structural changing towards opening and paired with cysteine mutants on individual sites to report on conformational changes"

The expression 'in solution' should be replaced with 'in DDM micelles' throughout.

We now write (L178): "TbMscL conformation and oligomeric state characterization"

(Figure 2 legend): "Structural state of TbMscL observed by PELDOR distance measurements in DDM micelles."

(L244): "The endogenous lipid content of WT protein in DDM..."

(L257): " No change was observed between the DDM and within..."

(L296): "...to comparable MscL expansion", e.g. : "in solution" was removed

(L304): "...K100R1-L89W in DDM did..."

(L306): "...L73R1-L89W in DDM revealed..."

(L316): "...displayed, in DDM, the characteristic linewidth..."

(L332): "...was observed in DDM for L72R1-L89W..."

(L337): "...suggest the DDM solution state exhibits structural similarities..."

(L428): "...lateral compression (DDM micelle state)."

(L429): "...adopted a closed state in DDM"

(L433): "...conformation in DDM micelles"

(Figure 6 legend): "...L89R1 mutant in DDM with the MTS"

(L469): "...while maintaining their pentameric state.", e.g. "in solution" was removed

(L494): "...The expanded L89R1 TbMscL state consists of 8-10 Å total..." e.g. "stabilized in solution" was removed

(L501): "...physiological conditions and lipid bilayers..." e.g. "(solution)" was removed

(L504): "...of L89R1 in DDM expanded state..."

(L506): "...The L89R1 TbMscL partially open conducting state presents structural..." , e.g. "solution" was removed

(L538): "...PELDOR structural state in DDM..."

3. Using PELDOR experiments with TbMscL, the authors observe that when cysteines in position 89 are modified with MTSSL in DDM micelles (~60% of sites labeled in the population), then the D1 and D2 distances between spin labels in pentamers increase by 3 and 5 Å, respectively, compared to the computational prediction based on 2OAR crystal structure of TbMscL. This corresponds to approximately 2.7 Å radial expansion inside the cytoplasmic leaflet of the membrane where the labels are attached. This is 3-4 times smaller than the fully open conformation predicts and must be clearly stated in the text.

We thank the reviewer for their comment. We indeed observe an increase of 6 Å in the inner diameter of the cytoplasmic leaflet, which equals a ~3 Å radial increase or expansion, e.g. similar to the value suggested by the reviewer (L231 and L494).

We now write (L231 – 233): "...of the inner-diameter at the cytoplasmic leaflet channel domain of ~6 Å, approximately a third of what is expected for an estimated full channel opening^{24,25}."

Importantly, this small expansion caused by chemical modification is observed only in micelles, whereas in liposomes and nanodisks the distances between labels correspond to crystallographic (closed-state) positions. It is very possible that the chemical modification of cysteines in position 89 with a bulky adduct somehow 'loosens' the structure specifically in micelles, perturbs the helical packing and even produces a leak in the structure. There is no reason to call this result of covalent modification allosteric opening.

This is at the core of our unique finding which makes this an exciting observation and introduces allostery (acting on a site distal to the channel pore) for mechanosensation, unlike previous studies which relied on pore residues. We showed that this specific site and location causes structural change due to an immediate channel response, not evident in any other of the rest 12 sites tested, which spanned most, if not all, structurally distinct channel domains.

The difference in distances between the two environments raises questions about the preferred conformations of the flexible MTSSL group in DDM micelles versus lipid bilayers characterized with strong anisotropy of chain orientation. The nitroxide moiety has the freedom to occupy many conformations, and in the extended conformation, the paramagnetic oxygen can protrude for 10 Å away from the alpha carbon of the labeled cysteine. The system clearly calls for a special comparison of nitroxide position in each specific environment present explicitly, but for some reason, the conformer statistics was done without the environment. Moreover, 100-ns long simulations of TbMscL were done in both lipids and DDM, but no analysis of preferred label conformers in these two different environments is presented.

We have used the two most-established and extensively benchmarked software tools available for our analysis and data interpretation. MMM uses a rotamer library approach, in which precomputed rotamers are energy-weighted for estimating their populations *in silico* labeling a specific structure; MTSSLWizard on the other hand uses a parametrized excluded volume approach stochastically generating conformations and excludes those which clash with the protein. Both approaches utilize an atomistic structure "*in vacuo*" and have proven highly successful (~217 and 90 citations of the core publications, respectively). We have extensive experience using both programs and have done this by state-of-the-art. By using the same construct, we could avoid having to rely on computational homology models, which induce structural and thus distance artefacts (we have seen this on *E. coli* MscL and will publish this elsewhere). Distances modelled using this state-of-the-art approach are in complete agreement with PELDOR distance measurements in reconstituted channels both (nanodiscs and liposomes) for all 13 mutants tested. In DDM, 12 (out of 13) mutants are consistent with modelled distances, while the measurement for L89R1 shows a significant and clear shift between DDM and lipids (nanodisc and liposomes) in the actual PELDOR derived distance data (no modelled distances were used or included in our analysis). Both distances are shifted. The reviewer's suggestion of a different preferential orientation of the spin-label side chain in detergent and lipids is not fully plausible. It is unlikely that this difference would only manifest in L89R1's case (the modified mutant we also see partial opening in single channel recordings). If the spin-labels within any sample had fewer conformations than sterically excluded by the protein due to additional lipid/detergent interactions, then the distance distribution should narrow to reflect this. However, this is not the case for any of our >10 spin labelled mutants, located within the transmembrane domain of MscL. Further, none of the previous PELDOR/DEER studies using MMM and MtsslWizard in detergent-solubilised membrane proteins report significant deviations to the *in silico* predictions. Most importantly, L89W mutant induces distance changes between labels at residues 72 and 73 that are in agreement with the models with unmodified L89. This extra set of experiments was done specifically to exclude any artefacts by spin label side chain orientation and to comprehensively rule this out we included four sites on the same helix to demonstrate not only a change but also provide information about the mechanical transduction (i.e. through rotation and expansion, consistent with previously expanded x-ray structures). Our modelling which facilitated cross-validation between two complementary approaches is sufficiently accurate to be used for channel conformational assessment in the case of TbMscL, while our long time traces used in our PELDOR experiments (~4µs) ensure that the obtained distance distribution shapes are reliable for all mutants (green bar region in DeerAnalysis).

The success of MMM and MtsslWizard is also underpinned by the fact that their parametrisation on large learning sets allows them to regularly outperform MD simulations not customised to spin-label behaviour, despite the latter including the environment. EPR distance measurements are accurate that MD simulations are indeed parametrised against them (*J. Am. Chem. Soc.*, 2017, 139 (34), pp 11674–11677). Here, we only performed MD to rationalise whether L89R1 can indeed cause delipidation in detergent and not to study conformational statistics. Lack of MD of other mutants means we cannot compare with measured distance distributions and evaluate this for all mutants to have robust statistics allowing firm conclusions, i.e. we would not be able to compare with the rest of mutants we lack MD data for, to assess the magnitude of discrepancies or inconsistencies of the MD calculations across the whole set of mutants not allowing reliable conclusions to be extracted. Atomistic MD simulations and distance calculation for all 13 labelled positions out of the scope of this study. Instead we calculated using two independent software tools for all the labelled positions and treated all mutants equally.

4. The only functional results on activation and conductive properties of L89R1 – labeled channels are presented in Fig. 3b. ‘Representative’ traces are shown. The uncertainty in these recordings is the size and curvature of GUV patches, which were not visualized but can vary drastically. Please refer to Moe and Blount (*Biochemistry* 2005) for the correct way of GUV patch-clamp data presentation and analysis.

We included data from all excised patches from which channel activity was recorded in our analyses, using pipettes fabricated in the same manner from the same type of glass capillary. Variability in patch geometry will contribute to the patch-to-patch variability when we report the apparent activation threshold with respect to pressure applied. In doing so, we recorded consistent functional behaviour of our control channel (WT MscL) from multiple patches to present reliable statistics, which manifest the robustness of our analysis (> 8 successfully pressure activated patches for L89R1). As discussed previously, which was acknowledged by this reviewer, it was enormously difficult to obtain data from the full range of pressures required for a detailed analysis of TbMscL activation, thus the apparent threshold pressure was used as surrogate. As described in the methods, and appropriately for this high-threshold channel, pressure was applied in brief steps during recorded sweeps using a pressure clamp, rather than as a continuous recording. For these reasons the data presentation and analysis differ to those in Moe and Blount (*Biochemistry*, 2005), but are consistent with the majority of broader electrophysiology literature.

We do understand the enormous difficulties of experimentation with TbMscL which activates at tensions exceeding patch stability. Is there a way around it? Based on their main hypothesis, in Fig. 6 the authors give interesting predictions: modification of C89 with a bulky MTS reagent should lower activating tension, whereas reduction of the MTS should restore access of lipids to the pocket and stabilize the closed state, i.e. shift activation curve back to the right. The effect of L89W substitution is predicted to be essentially the same as MTS modification. If this is true, then the following experiments will provide much stronger support to this inference than we find in the paper. We thank the reviewer for acknowledging the “enormous difficulties of experimentation” with the construct we used, which indeed most times exceeds patch stability and presents substantial challenges. We have now performed the suggested experiments which are expected to “provide much stronger support” towards the validity of our model and the “lipid moves first” hypothesis. In particular, we have implemented patch clamp experiments in the presence of DTT (L89R1) and observed that the pressure activation threshold has been restored back to WT levels. This new finding suggests that it is indeed the modification on this unique which causes the observed functional effects and not the mutation itself, in complete agreement with our initial suggestion and model.

We have now added new figures (3b and S6a) and text describing our new findings (i.e. L68-9, L271, L292-4 and L768-70)

a. It would be important to show whether or not incubation of L89-R1 TbMscL containing liposomes with a reducing agent (mercaptoethanol, DTT) reverses the gating presented on the right side of Fig.3b

We thank the reviewer for their comment and suggestion. We have now performed patch clamp electrophysiology recordings under applied tension on L89R1 TbMscL in GUVs in the presence of reducing agent (DTT). We indeed observe restoration of the pressure activation threshold to WT channel levels, suggesting the observed changes in channel function are a consequence of the cysteine modification and not the single point mutation on this unique (identified by PELDOR) site. Please also see previous comments and new added figures (3b and S6a) and text L68-9, L271 L292-4 and L768-70.

b. If the hindrance of aliphatic chain penetration at the entrance really changes the channel activation threshold or midpoint, then any other bulky MTS reagent should do the same. The authors have a broad choice from apolar (benzyl or long alkyl methanethiosulfonate) to polar uncharged or charged sidechains. It would be important to compare their functional effects.

We thank the reviewer for their suggestion and we are pleased to see that they recognise important implications arising from our study, that is allosteric regulation of mechanosensitive channels through cysteine modification. Although this point was not raised in the initial review and we recognise that these potential findings will add to the variety of ion channel applications, it is out of the scope of this study. We are aware that this could be an intriguing outcome ours and other group(s) would consider pursuing in the future, allowing mechanosensitive channel manipulation by allostery, using a variety of covalently attached sulfhydryl molecules.

c. L89W is predicted to have the same effect as MTSSL due to its ability to interfere with acyl chains at the entrance to the pocket. What is the tension midpoint of this mutant in spheroplasts relative to WT TbMscL?

We have not performed electrophysiology in spheroplasts to avoid artefacts which may arise of presence of other unknown channels in the *E coli* genome and are sensitive to tension. We have performed electrophysiology experiments in GUVs for L89R1, as this was more appropriate, e.g. linked to our PELDOR data, for our study and importantly included a modification which could be readily removed *in vitro* (incubation with reducing agent) and thus its immediate effect. W mutation was used to account and control for spin label orientation artefacts, as explicitly described in previous response and manuscript.

Our concluding comment is about the main hypothesis of whether the lipid occupancy of TbMscL pockets can drastically change the behavior of this molecule. The dynamics of TM1 helices forming the gate is defined by (1) tight TM1-TM1 packing mediated by alternating glycines and (2) packing and dynamics of the outer TM2 helices around the TM1 barrel. The TM1 helices are apparently well constrained by the surrounding helices and are not free to move. The pressure of lipids is transmitted to the gate primarily through the lipid-facing helices and only to a small degree through the direct contact with lipid chains in the pocket (based on the gross contact area). Only if the gate-bearing TM1 helices were free to move and could somehow expand inside the TM2 sheath, then the lipid 'stuffing' in the pockets could possibly restrain it. To examine the significance of the presence of aliphatic tails in the pockets computationally, the authors should quantify lateral mobility of TM1s in the full bilayer with and without lipid chains in the pockets. Before this is done, it is impossible to assess the value of this hypothesis. Intuitively, the net pressure/tension of surrounding lipids will be transmitted pretty much the same way through the helical frame of the protein no matter what is inside the pockets located near the midplane of the membrane.

We thank the reviewer for their comment and new suggestion. We have included an extensive computational analysis in both our initial and revised MS, including extensive pairwise force calculations, all of which, strongly support our model.

The reviewer describes conformational changes involved in channel activation, but the precise mechanism that responds to changes in membrane tension is not understood. Our present study

identifies one of the precise mechanisms involved in this process. It is now clear that lipid occupancy in NPs is critical to the gating mechanism of MscL and potentially other mechanically gated ion channels. Our proposed model is not mutually exclusive with the mechanisms of channel activation described by the reviewer, but clarifies the relative importance of specific lipid-protein interactions. Our study now shows that it is indeed the changes in lipid contacts within the NPs that is significant and responsible for translating changes in membrane tension to channel activation, compared to the gross contact with lipid-facing helices. This is our unique finding and is evident from the combination of all (structural, functional and computational) data in our manuscript. Our study and proposed hypothesis will now facilitate further investigation into the fine details of how the changes in lipid occupancy within the NPs result in the conformational and channel pressure activation threshold changes.

Reviewers' Comments:

Reviewer #2:

Remarks to the Author:

After reading the initial version of the paper "Allosteric activation..." by Piotas and coworkers and two subsequent revisions, I remain unconvinced that the main hypotheses are correct. The authors insist that the lipid-accessible pockets on the cytoplasmic side of the transmembrane barrel have to be constantly occupied by lipid chains to stabilize the resting state, and the chains need to be retracted from the pockets in order to initiate the opening. They claim that the sidechain in position 89, when changed to cysteine and modified with MTSL (L89R1), hinders the lipid chain access to the pocket and thus prompts barrel expansion. All provided pieces of evidence are circumstantial.

In order to adequately support these two separate inferences, the authors should have to add three specific parts to the paper.

1. It should be demonstrated that MTSL and a few other MTS reagents (preferably bulky and hydrophobic) shift the activation curve of the spheroplast-expressed L89C TbMscL population to the left. Complete labeling is not required, even partial labeling may produce a shoulder in the activation curve. MscS expressed in the same spheroplasts should be used as an internal tension reference. The single-channel results presented on Fig. 3b are not reliable without patch imaging. There could be an accidental slippage of liposome patches leading to higher sensitivity. The statistics is unacceptably small. Measurements in spheroplasts must be done, they should provide a reliable population response.

2. If the authors want to demonstrate the effect of a bulky sidechain near the entrance, the same measurements should be done with the L89W mutant, for the same reason.

3. If the authors claim that the gate is destabilized by the absence of interactions with the aliphatic chains occupying the pockets (and not the distortion of structure and rotation of helices by the bulky, partially hydrophilic MTSL chain), they have to carefully analyze the dynamics (RMSD) of the TM1 bundle (gate) in simulations in which the aliphatic chains are either allowed or disallowed to enter the pockets. If the authors want to connect the altered dynamics of the gate with the observed subconductive states, then they have to analyze and demonstrate the changes in gate solvation.

The only result that seems to be real is the small expansion of the barrel when the cysteine in one specific position 89 is modified with MTSL. How this observation relates to the gating mechanism and supports 'allostery' remains unclear. The rest of the data represent a lot of work but does not create a cohesive and convincing story.

Reviewer #2 (Remarks to the Author):

After reading the initial version of the paper “Allosteric activation...” by Pliotas and coworkers and two subsequent revisions, I remain unconvinced that the main hypotheses are correct. The authors insist that the lipid-accessible pockets on the cytoplasmic side of the transmembrane barrel have to be constantly occupied by lipid chains to stabilize the resting state, and the chains need to be retracted from the pockets in order to initiate the opening. They claim that the sidechain in position 89, when changed to cysteine and modified with MTSL (L89R1), hinders the lipid chain access to the pocket and thus prompts barrel expansion. All provided pieces of evidence are circumstantial.

We thank the Reviewer for their comment. We have now performed multiple additional experiments and further computational analysis, which we include within the new substantially revised version of the MS. We have added new figures and text in both the main manuscript and supplementary material (highlighted in yellow). These newly obtained data are fully supportive of the “lipid moves first” hypothesis. In particular:

- a) We obtained new electrophysiology data from $n = 5$ independent patches on L89W, which show that the pressure activation threshold is reduced dramatically compared to WT channel. A sub-conducting state of ~ 250 pS under no applied tension could also be resolved, suggesting a similar functional behaviour to L89R1. Importantly, we observed minimal threshold variability among multiple L89W patches, consistent with our model for an 100% modified mutant.
- b) We performed computational analysis of the RMSD of the pore lining TM1 helix, in respect to lipid chain contacts with the NPs. New data suggest that there is a correlation between lipid chain availability within the NPs and structural perturbations of the pore lining helix TM1, which controls ion conductance.
- c) We performed total energy ratio (pairwise forces) calculations between TM1 pore lining helix and L89 residue in the absence (WT) and presence of the modification (L89R1). These data suggest that MTSSL's presence causes minimal distortion to the channel gate and is 609x smaller than the effect this has on lipid occupancy reduction, within the site of allostery (NP).
- d) We calculated the expected pore opening diameter of the measured sub-conducting state (now present in both modified variants L89W and L89R1) in the absence of applied tension (250-270pS). Our calculations indicate this sub-conducting state which is present and resolvable in $n=11$ patches with L89W and $n=9$ patches with L89R1, but not WT channel, is consistent with ~ 8.8 - 9.2 Å pore opening diameter. Finally, we independently calculated the structural change observed by PELDOR (between lipids and detergent), only for L89R1 and out of 13 sites tested. This change is consistent with a pore diameter of ~ 9 Å providing an “extra” link between our functional (electrophysiology) and structural (PELDOR) observations.

In order to adequately support these two separate inferences, the authors should have to add three specific parts to the paper.

1. It should be demonstrated that MTSL and a few other MTS reagents (preferably bulky and hydrophobic) shift the activation curve of the spheroplast-expressed L89C TbMscL population to the left. Complete labeling is not required, even partial labeling may produce a shoulder in the activation curve. MscS expressed in the same spheroplasts should be used as an internal tension reference. Measurements in spheroplasts must be done, they should provide a reliable population response.

We respectfully disagree that sulfhydryl modifiers (including MTSL, other MTS-reagents and proxyl spin labels as maleimido-proxyl) are able to react with inner-membrane *E coli* proteins and thus MscL. In particular, previous studies provided direct experimental evidence and suggest otherwise. For instance, *Joseph et al., 2015, Angew Chem; Joseph et al., 2016, JACS*) and others have demonstrated that the MTS- could pass through the outer membranes and is immediately being reduced in the periplasm. We here cite a part of from their JACS article:

“The outer membrane is permeable to molecules below 600 Da and the MTSL could easily reach periplasm...”

“However, we have never observed an EPR signal following “MTSL” labeling of E. coli cells expressing “cys-less” BtuB (E coli outer membrane) or other cysteine mutations located in periplasm. Attempts to label cysteines at the periplasmic interface with maleimido-proxyl also did not give a signal, ruling out any possible interference from the disulfide bond formation (Dsb) system. These observations suggest

that the MTSL is reduced following entry into the periplasm. We demonstrated previously that a spin-labeled CNCbl, which binds tightly to BtuB, is not reduced by cells confirming that the reduction must happen only after entry into the periplasm.”

“As we reported earlier, no signal could be detected for the WT cell pellet (Joseph et al., 2015, *Angew Chem*) or for cysteine mutations located in periplasm (8C, 9C).”

Furthermore, the new “in-cell EPR” field had to take the inevitable reduction and cleavage into account and indeed new spin labels and alternative linking strategies have emerged to address these challenges (Martorana et al., 2014, *JACS*; Qi et al., 2014, *JACS*).

Formed spheroplasts suitable for patch clamp electrophysiology experiments retain both membranes. We copy from Martinac et al., 2013, *Methods Mol Biol*: “The best spheroplasts for obtaining a giga-ohm seal are the shiny ones. These spheroplasts have two membranes (outer and inner one, since *E coli* is a Gram-negative bacterium)”.

Since, upon entering the periplasm any MTS sulfhydryl modifier will be reduced and it will thus not be able to label TbMscL L89C or any other *E coli* inner-membrane protein e.g. whether the protein site is exposed or not, thus it is not technically possible to use MTS reagents and perform the suggested experiments. However, we have now performed new electrophysiology measurements for L89W and included in the revised MS (Fig 3b,c & S6c). We show that other than MTSSL modifications (i.e. W that will not be reduced nor reductively cleaved) located at the same site (L89W) also cause a dramatic effect on MscL’s pressure activation threshold and therefore function (for more details regarding new data on L89W, please see response below).

The single-channel results presented on Fig. 3b are not reliable without patch imaging. There could be an accidental slippage of liposome patches leading to higher sensitivity.

Sorry, but we strongly disagree with the Reviewer’s comment; there is no evidence of accidental slippage. If this was the case, then a high variability should occur between patches containing WT TbMscL. In contrast, all WT channels independent patches (n=4) gave rise to similar activation thresholds, which are consistent with previous studies (Mukherjee, et al., 2014, *FASEB J*). Moreover, L89W which possesses the equivalent of 100% modification efficiency and exhibited a large decrease in gating threshold pressure, also shows very little variability across the n = 5 patches. There is therefore no evidence that our data are affected by an accidental slippage. We only observed limited variability with L89R1, which can be attributed to incomplete labelling, and which we quantified (~62-71%) by two independent methods (Table S1 & S3). Therefore, a statistical distribution in the number of cysteine modifications per channel is expected and according to our model, this should have its highest threshold value similar to the WT level (140-160 mmHg, n = 3), where we recorded ~110-140 mmHg in n = 4/9 L89R1 patches, and the rest of the values distributed at lower thresholds, where we measured ~30-90 mmHg in n = 5/9 L89R1 patches. Thus the values range between, and not beyond, the equivalent WT and L89W data. Importantly, the L89R1 distribution shifted to WT levels when L89R1 was incubated with DTT for MTSSL removal, indicating that patch slippage cannot account for this change in sensitivity. The consistency of the expected outcomes (according to our model) with the actual recordings manifests the importance of modification efficiency at the allosteric site, therefore the presence of the modification itself.

The statistics is unacceptably small.

There is no evidence to suggest that the data are not reproducible. We used independent numbers of patches that either equals or exceeds the n-values reported in multiple well-established studies in the field of MS channels (MscS and MscL) under applied tension. In particular, this number varies from n=4 to n=5-8 and for a single case n=9-15 (Nomura et al., 2012, *PNAS*) and n=3-4 (Mukherjee et al., *FASEB J*, 2014). In general, with ion channel gating and permeation parameters so accurately measured by patch clamp electrophysiology, there is little variability between the behaviour of identical ion channel proteins, thus n-values with single digits are the norm. In our study we obtained highly reproducible single channel recordings with both WT and L89W TbMscL, with properties of the former in complete agreement with reported conductance and activation threshold with the same TbMscL construct in GUVs (Mukherjee, et al., 2014, *FASEB J*). For the rest of our modified variants, we obtained independent patches from n=9 for L89R1, n=7 for sub-conducting L89R1, n=9 for L89R1 with DTT reduction to L89C, and now n= 5 for L89W and n=11 for sub-conducting L89W, which lies at the upper limit or exceeds the n-values reported in many previous electrophysiology studies. For these reasons, we respectfully disagree with the Reviewer that “statistics” are unacceptably small, they are the same

or more than many previously published reports on similar systems. Furthermore, the data enabled nonparametric analysis, which revealed differences between groups with $p < 0.001$ (Kruskal-Wallis), which would not be possible if n-values were too small, relative to the measured effect.

2. If the authors want to demonstrate the effect of a bulky sidechain near the entrance, the same measurements should be done with the L89W mutant, for the same reason.

We have now performed L89W patch clamp electrophysiology in GUVs under applied tension. We observe in patches to which we applied tension an activation at dramatically lower activation thresholds compared to WT channel (Fig 1). Further, for (n=11) we resolved a sub-conducting state of ~250pS, similar to the ~270pS we previously identified for L89R1, in the absence of applied tension. Finally, we observed very low threshold variability within multiple tested L89W patches, consistent with a 100% “labelled/modified” mutant.

Fig 1 (see also Fig 3b,c & S6c in the MS). a. Representative patch clamp recordings at 20 mV from patches excised from GUVs containing L89W (modified) TbMscL protein. The pressure applied during the recording is indicated below each trace. The current levels representing the closed (“C”) and fully opened (“O”, approx. 60 pA) channels are indicated. b. Distribution of threshold pressures, applied at 10 mmHg intervals, at which pressure-activated MscL full channel openings were first observed. Data points represent individual excised patches, whiskers show the full range of values, the boxes represent the 25 to 75 percentile range, and the horizontal line is the median. Samples are significantly different ($p < 0.001$, Kruskal-Wallis ANOVA), with 7/9 L89R1 and 5/5 L89W values below the median ($p < 0.005$, Moody Median test). c. Representative recording at +20 mV from a patch excised from GUV containing L89W TbMscL, with no externally-applied pressure. The brief downward deflections (C), approximately 5 pA, are closures from an otherwise constitutively active sub-conductance state (S).

We have now included new Fig 3b,c and Fig S6c and added legend and main text accordingly.

We write:

Lines (68-69): “...of modifications on this site...”

Line 263 (legend): “c. Distribution of threshold pressures, applied at 10 mmHg intervals, at which pressure-activated MscL full channel openings were first observed. Data points represent individual excised patches, whiskers show the full range of values, the boxes represent the 25 to 75 percentile range, and the horizontal line is the median. Samples are significantly different ($p < 0.001$, Kruskal-Wallis ANOVA), with 7/9 L89R1 and 5/5 L89W values below the median ($p < 0.005$, Moody Median test).”

Line (286-292): “ Patches excised from GUVs containing L89W TbMscL protein, like with L89R1 modified protein, exhibited a high level of background current in the absence of applied pressure. Occasionally, unitary closures were recorded, which enabled the calculation of a mean unitary conductance of 0.248 ± 0.022 nS ($n=11$) (Fig S6c). In some patches, applying negative pressure evoked opening of a channel with mean conductance of 2.43 ± 0.27 nS ($n= 5$), with pressure thresholds ranging -60 to -80 mmHg (Fig 3B).”

Line 454 (legend): “(or equally L89W)”

Lines (477-480): “Importantly, when L89 was mutated to W, we recorded substantial conformational changes occurring distal to the site of allostery NP, in regions which were previously structurally unchanged in the absence of NP modifications (Fig 4).”

Lines (487-488): “...and test the effect on channel structure by introducing other than MTSSL modifying molecules at the allosteric site,...”

Lines (507-510): “~270 pS and ~250 pS state evident in L89R1 and L89W respectively (but not WT) (Fig 3b & S6b,c) in the absence of externally applied tension is consistent with a ~9.2 Å and ~8.8 Å pore diameter estimated opening for L89R1 and L89W respectively(78)”

Lines (563-567): “This may also account for the increased variability of the pressure activation threshold for L89R1, which is on average lower than WT, but then approaches WT values after DTT incubation. This is supported by the low activation threshold of L89W, which is equipped with a tryptophan on every equivalent site, and exhibited very limited variability similar to WT (Fig 3b & S6a).”

3. If the authors claim that the gate is destabilized by the absence of interactions with the aliphatic chains occupying the pockets (and not the distortion of structure and rotation of helices by the bulky, partially hydrophilic MTSL chain),

We have now calculated the pairwise forces between L89R1 and WT with TM1 pore lining helix respectively test whether there is a structural “distortion” caused by the presence of MTSSL at the NP entrance. Our analysis shows that the ratio of distortion to TM1 helix the presence of MTSSL induces, is 609x smaller than the effect its presence has to the total energy ratio induced by the lipid chains directly to the NPs. Therefore, although the expected minimal structural distortion the modification seems to have, this is negligible and incapable of promoting global channel changes we measured by PELDOR and electrophysiology, for two types of modification (R1 and W) which although different, promote similar channel changes.

We have extended Table S5 and included new pairwise force calculations between L89R1 and TM1 (Table S5 bottom row).

We also write:

Lines (406-407): “ d)TM1 and residue 89 (L and MTSSL) (Table S5).”

Lines (413-415): “and between pore-lining helix TM1 and L89 or L89R1 (Table S5). This is consistent with a model in which the modification a) does not cause a structural distortions to the gate and...”

Lines (533-534): “...without those being structural distortions caused by MTSL’s presence (Table S5).”

they have to carefully analyse the dynamics (RMSD) of the TM1 bundle (gate) in simulations in which the aliphatic chains are either allowed or disallowed to enter the pockets. If the authors want to connect the altered dynamics of the gate with the observed sub-conducting states, then they have to analyse and demonstrate the changes in gate solvation.

We thank the Reviewer for their comment and the opportunity to provide further computational evidence which supports our proposed model. We have now implemented RMSD calculations for the last 20ns of the atomistic MD simulations (80-100ns) of TM1 in correlation to the number of lipid contacts for both the L89R1 and WT channels and include these new data (lines 399-400, 531-534, 789-791) and new Fig S11c).

We observe structural perturbations of the pore lining helix TM1 (L89R1) (given we do not intentionally apply artificially large force fields for reasons we explicitly explain in the MS, lines 542-544) and direct correlation between these changes and lipid chain availability within the NPs. These results suggest the TM1 undergoes structural rearrangements in a concerted manner to lipid occupancy variability within the NPs, and therefore conclude that the channel “senses” and directly responds to subtle lipid chain occupancy changes within its NPs (the site of allostery). Further, analysis on the WT channel (control), which presents constant lipid pocket availability resulted in no TM1 structural perturbations, consistent with a closed channel for which the NPs are constantly (or vast time majority) occupied by lipid chains.

Fig 2. TM1 RMSD (grey line) and A18 lipid contacts (red line) over time (final 20ns of the 100ns total atomistic MD simulation) for L89R1 and WT TbMscL respectively, along with the calculated correlation for each case

We write:

Lines (399-400): “Changes in lipid chain occupancy within the L89R1 NPs correlated with RMSD changes of the TM1 pore-lining helix, not evident in the WT channel (Fig S11c).”

Lines (531-534): “...while lipid occupancy changes within the allosteric NP site showed correlation with structural perturbations occurring at the TM1 pore-lining helix, which controls ion pore conductance (Fig S11c), ...”

Lines (789-791): “Pair wise energy, RMSD and lipid acyl-chain atom contacts with TbMscL amino acids were calculated using Gromacs command `gmx energy`, `gmx rms` and `gmx mindist` respectively.”

The only result that seems to be real is the small expansion of the barrel when the cysteine in one specific position 89 is modified with MTSL.

We thank the reviewer for their comment and are pleased that the reviewer now recognises there is indeed a “structural” and thus mechanical response of the high pressure threshold channel, as a consequence of modifying the entrance (L89) of the allosteric site (NP). Indeed, this is the key message that we wish to communicate to the scientific community. Regarding the magnitude of the structural change we respectfully disagree with the reviewer as our data from multiple state-of-the-art methods (for details see response below) suggest otherwise, e.g. that it is only a small barrel expansion. In particular, we experimentally measured using PELDOR a 5 Å pore diameter change occurring (L89R1) between DDM and lipids (both liposomes and nanodiscs).

We write in the revised MS:

Line (224): “...cytoplasmic leaflet channel domain of 5 Å, reaching a total diameter of ~9 Å...”

Lines (499-501): “The expanded L89R1 TbMscL state forms an open pore of ~ 9 Å in total diameter at the cytoplasmic leaflet site, e.g. 5 Å diameter increase reported by PELDOR, added to the 3-4 Å diameter of the constriction site of the closed TbMscL state (PDB 2OAR),...”

This measured by PELDOR conformational change, along with the 4 Å initial pore diameter of the x-ray closed structure (PDB 2OAR, *Chang et al., 1998, Science*) results in a substantial channel expanded state, with a 9 Å pore diameter opening, sufficient to conduct ions (Fig 3).

Structural change between DDM vs Nanodisc L89R1 (PELDOR)

Liu, Z., Gandhi, C. S., and Rees, D. C., 2009, *Nature*; Li, J., Guo, J., Ou, X., Zhang, M., Li, Y., and Liu, Z., 2015, *Proc Natl Acad Sci U S A*; Chang, G., Spencer, R. H., Lee, A. T., Barclay, M. T., and Rees, D. C., 1998, *Science*; Plotas, 2017, *Methods Enzymol*

Fig 3. Pore inner-diameter change measured by PELDOR in DDM and NDs. This change is added to the closed state (PDB 2 OAR) initial pore diameter resulting in a state of 9 Å total opening diameter, able to conduct ions

To further confirm and characterise this conductive state, we calculated the ion conductance corresponding to an 9 Å channel pore opening diameter according to *Cruickshank et al., 1997, Biophys J*, (Fig 4). Our calculations resulted in 270 pS for 9.2 Å (L89R1) and 250 pS for 8.8 Å (L89W), pore conductance. There is thus an excellent agreement between the "structural" change we observe by PELDOR and the functional sub-conducting state we resolved within n=18 patches (under no externally-applied pressure) of our NP-modified variants, e.g. L89R1 and L89W, but not for WT.

Functional state (electrophysiology)

Fig 4. Calculation of MscL pore diameter corresponding to a channel with a 270pS conductance (i.e. sub-conducting state we measured in the absence or null tension for L89R1 and L89W). The total diameter is 9.2 Å (270 pS, L89R1, Fig 3b and S6b) and 8.8 Å (250 pS, L89W, Fig S6c), consistent with structural changes measured by PELDOR.

We write:

Lines (507-510): "... ~270 pS and ~250 pS state evident in L89R1 and L89W respectively (but not WT) (Fig 3b & S6b,c) in the absence of externally applied tension is consistent with a ~9.2 Å and ~8.8 Å pore diameter estimated opening for L89R1 and L89W respectively(78)."

This state therefore represents a significant conformational change, which leads to a substantial pore increase and not just an incremental structural distortion. Importantly, it is the direct consequence of

the mechanical response to occurred changes within MscL's allosteric site. A pore diameter size of 9 Å does not represent a negligibly expanded state in any membrane channel pore in nature and importantly it is able to readily conduct ions and allow the flow through of molecules. For instance, MscS' "fully" open pore is 14 and eukaryotic pores are only few Å wide. Given TbMscL is the channel with the highest pressure activation threshold known in nature, including EcMscL, a mechanical response of such magnitude demonstrates the vital importance of allosteric sites in mechanical gating.

How this observation relates to the gating mechanism

We thank the reviewer for their comment and the opportunity to clarify and expand on this crucial point. We do not claim that "full" channel opening occurs when the lipid moves out of the NPs and clearly stated that extra tension will be required (Lines 545-546). We establish that this is the first and very important step for the initiation of the gating process. We demonstrate that the lipid which moves first and away from the allosteric site NP, upon tension application, results in destabilisation of the channel closed state and dramatically increases the open probability of a substantial (~1/3 of full opening diameter) sub-conducting state. In particular, this first lipid NP removal reduces by more than 60% (~30-60mmHg, instead of 150-160 mmHg are needed) the total energy tension required for full opening. Therefore, only an "extra" 30% is now required for the channel to transit from this state (~9 Å, 270pS) to its fully open state (~25 Å, 2.5-3nS). Elucidating the final step or steps of the gating process (from sub to full opening), will be the focus of our (and also other groups) future studies, but it is clearly out of the scope of the current study.

Similar sub-conducting states have been previously observed and characterised in EcMscL (*Sukharev et al., 1999, J Gen Phys; Sukharev et al., 2001, Nature*). These were 1/10 of full opening signal height (~200-300pS) and usually appeared prior to full channel opening events under applied tension. However, the physical meaning and importance of these states in channel's gating mechanism have remained elusive.

We hereby establish that this conducting state is the first step in MscL's gating process and is the direct consequence of the lipid, which "moves first" away from the allosteric NP site. To do so, the lipid "consumes" 60-70% of the total energy required for full channel opening, which is sufficient to promote a ~9 Å pore opening diameter state, that is approximately 1/3 of the fully open expected diameter (10% of total conductance and 30% of total pore diameter). These lipids may not all or simultaneously move away from the 5 available allosteric sites (pentameric MscL), which may indicate that the participation of the five identical subunits is asymmetric, consistent with previous studies (*Birkner et al., 2012, PNAS*) and also with the effect of modification efficiency, e.g. leading to asymmetric lipid NP occupancy, we have observed.

We have now made amendments in the text to better reflect these points. We have also created a new Figure 5 (Fig S12 in the MS) to summarise these observations and highlight the relation to MscL's overall gating NP lipid removal has, in order facilitate efficient communication with the reader.

Fig 5. Summary of experimental observations on the effect that NP lipid removal has to MscL's overall gating. Lipids moves first out of the NP resulting in a structural (PELDOR) and functional (electrophysiology) response of the channel leading to a sub-conducting state which possess 10% and 30% of channel's total conductance and pore diameter respectively. To implement this step, e.g. lipid to be removed, 60-70% of the total energy required for MscL full opening has to be consumed. A further 30% of the total energy is then required to fully open the channel (2.5nS, 25 Å), which could be implemented through membrane tension application in a single or a number of intermediate steps.

We write:

Lines (586-606): "Following channel activation by NP lipid removal (sub-conducting state) within the membrane, only an "extra" ~30% of the total energy (~30-50mmHg out of 150-160 mmHg) is required, e.g. manifested by the dramatic decrease in pressure threshold, to promote a transition from the sub-conducting (~9 Å and 1/3 of the fully open pore) to the fully open state (~25 Å).

We demonstrate that the "lipid has to move first" and out of the site of allosteric NP. We establish this event constitutes the first and essential step for mechanical sensing and the subsequent first structural response to occur. Lipid chain removal from the NP could be achieved through modification at the entrance of the site of allosteric NP in the absence of bilayer compression or null tension or upon tension increase for the case of WT, non-NP modified channels, similar to the natural process. Independent to the method in order for facilitating NP lipid chain removal the consequence is the destabilization of the closed state of the MS channel and the dramatic increase of the open probability of the first substantial (~1/3 of full opening diameter) conducting state of the channel. We show that by reaching this step the energy required for full MscL opening has been reduced by more than 60% (30-50mmHg, instead of 150-160 mmHg). Namely, MscL needs to consume 60-70% of the total energy required to achieve its full opening in order to remove the lipid chains from its NPs and transit to a sub-conducting state (as a structural response of the lipid NP removal) which possess an opening ~30% (9 Å) of its expected total pore diameter (25 Å) and ~10% (250-270pS) of its total pore conductance (2.5-3nS). Only an "extra" 30% of energy (or membrane tension) will be finally required by MscL in order to transit from its current sub-conducting state (9 Å, 250-270pS) to its fully open state (25 Å, 2.5-3nS)."

and supports 'allostery' remains unclear. The rest of the data represent a lot of work but does not create a cohesive and convincing story.

MscL is a mechanosensitive (not ligand-gated) ion channel and is activated by increased membrane tension. Membrane lipids are able to modulate MS channels without directly accessing their pore, thus achieving gating through an allosteric mechanism. Previous successful attempts to modulate MscL's pore were implemented through application of triggers, which acted directly from within channel's pore (Lines 118-119) (i.e. modified pore residues inaccessible to lipids), not being allosteric and thus not biologically relevant. Therefore, although these studies were highly valuable for controlling MS channel

pores, they did not provide an explanation on how tension is transmitted from the lipids to the channel gate in order to promote gating through the incorporation of lipids. According to the “lipid moves first” model, the lipid is a negative allosteric modulator, which acts distal to the pore. Channel opening is proposed to occur due to lipid moving out from the NPs, leaving the latter unoccupied as a consequence of increased tension, and resulting to a decrease of channel activation free energy. The lipid, following a membrane tension decrease, moves back and occupies this pocket in order to increase the free activation energy and keep the channel closed, otherwise a “leaky” channel would be lethal to the cell. Therefore, the lipid, in the case of MS channels, does not act as a “normal” ligand, but instead as a negative allosteric modulator and conformational changes are not driven by specific ligand-protein interactions. In order to elucidate this naturally occurring allosteric mechanism, it was essential to:

- a) **Identify the location of site of allostery.** We have therefore used rigorous and multiple site directed modification with multiple sites spanning all channel domains including both TM1 and TM2 helices, the S helix and cytosolic bundle
- b) **Demonstrate any structural changes occur only at the allosteric site.** We have thus performed PELDOR measurements, a method which offers subÅ resolution and followed transitions to identify the allosteric site both in detergent solution where membrane compression is not present and two distinct lipid/native environments of different lipid composition and curvature. We further identified optimal lipid types by performing two independent lipid qualitative and quantitative analyses (i.e. NMR and ES -mass spectrometry)
- c) **Demonstrate structural changes initiated at the allosteric site lead to global changes.** We introduced a modification (W) at the entrance of identified allosteric site and recorded large (>7 Å and >12 Å respectively for two distal to the NPs residues) conformational rearrangements >22 Å away from the site of allostery (NP). We further now show that these changes are transmitted to the channel pore lining helix and correlate with the degree of lipid occupancy at the allosteric site. Finally, W modification could act as a further control for excluding any spin label orientation artefacts on distance modelling and measurements.
- d) **Link the occurring structural changes to equivalent functional changes.** We performed patch clamp electrophysiology and showed that two distinct modifications at the entrance of the site of allostery (NP) with different modifications had a dramatic effect on pressure activation threshold, in complete agreement with our model. Moreover, both modified mutants promoted a novel state, due to a spontaneous MS channel structural response, as a direct consequence of disrupting the entrance of the site of allostery. Importantly, we demonstrated that these structural changes are equivalent to the occurring functional changes and thus mechanical responses.
- e) **Demonstrate that any occurring allostery is reversible, as any naturally occurring process should be.** We showed that the structural change which occurs in the absence of membrane (detergent solution) is restored in the presence of membrane compression for two lipid systems of a different lipid composition and curvature (nanodiscs and liposomes). In parallel, we showed that the severe effect on channel function is restored, when the modification responsible for this change, is removed (i.e. incubation of L89R1 with DTT reducing agent).
- f) **To demonstrate that the negative allosteric modulators (NP-lipids) should directly interact with the modification at the molecular level.** We set up and performed atomistic MD simulations of the modified channel within lipid bilayers and quantified the effect of the modification has, to the lipid chain occupancy within the allosteric site (NPs). We showed that the modification a) significantly reduces lipid chain occupancy within the NPs, thus reducing the interaction between the lipid membrane and the allosteric site and also b) has the structural “capacity” to efficiently disrupt the contact between the lipid bilayer and the channel by restricting entrance of lipid chains into the NPs. Further, we now show that there is a correlation between lipid chain NP occupancy and structural perturbations of TM1 pore lining helix. Finally, we provided experimental evidence (i.e. ESEEM, LT CW-EPR and PELDOR combined with charged cluster neutralisation), on which part of the allosteric modulator (i.e. the lipid) is responsible for channel gating. We demonstrated that is the lipid “acyl-chain” occupancy within the site of allostery, rather than the specific lipid “headgroup” binding to the charged cluster responsible for MscL’s gating response.

g) Implement and provide appropriate experimental and computational controls and where available data our controls should be consistent with existing literature. We expressed, purified and modified over 20 mutants, all resulting in well folded proteins and spanning sites proximal and distal to the allosteric site. We obtained high quality PELDOR spectra for 13 of these sites. For the ones which data deviated from the closed state x-ray structure model we performed additional PELDOR in lipids and demonstrated that only the identified mutant located at the entrance of the allosteric site distances reverts back. We used 88W which is immediately adjacent to 89 but not at the entrance of the NPs, paired with other spin labelled residues and observed no structural effect with PELDOR, in contrast to L89W. For our electrophysiology experiments we initially performed WT channel measurements of multiple and independent patches and our data were in complete agreement with literature, ruling out any measurement artefacts. For our MD simulations we performed measurements in detergent along with multiple WT channel experiments to control and elucidate the interplay between the modification and the allosteric site.

Our story is full and completed, cohesive, allowing the interconnection of data generated by multiple independent state-of-the-art experiments leading to a convincing outcome. Importantly, none of our generated data, which “represent a significant amount of cutting edge methodologies and work”, contradict our model, but in their full majority agree with it.

Reviewer #2 (Remarks to the Author):

After reading the initial version of the paper “Allosteric activation...” by Pliotas and coworkers and two subsequent revisions, I remain unconvinced that the main hypotheses are correct. The authors insist that the lipid-accessible pockets on the cytoplasmic side of the transmembrane barrel have to be constantly occupied by lipid chains to stabilize the resting state, and the chains need to be retracted from the pockets in order to initiate the opening. They claim that the sidechain in position 89, when changed to cysteine and modified with MTSL (L89R1), hinders the lipid chain access to the pocket and thus prompts barrel expansion. All provided pieces of evidence are circumstantial.

We thank the Reviewer for their comment. We have now performed multiple additional experiments and further computational analysis, which we include within the new substantially revised version of the MS. We have added new figures and text in both the main manuscript and supplementary material (highlighted in yellow). These newly obtained data are fully supportive of the “lipid moves first” hypothesis. In particular:

- a) We obtained new electrophysiology data from $n = 5$ independent patches on L89W, which show that the pressure activation threshold is reduced dramatically compared to WT channel. A sub-conducting state of ~ 250 pS under no applied tension could also be resolved, suggesting a similar functional behaviour to L89R1. Importantly, we observed minimal threshold variability among multiple L89W patches, consistent with our model for an 100% modified mutant.
- b) We performed computational analysis of the RMSD of the pore lining TM1 helix, in respect to lipid chain contacts with the NPs. New data suggest that there is a correlation between lipid chain availability within the NPs and structural perturbations of the pore lining helix TM1, which controls ion conductance.
- c) We performed total energy ratio (pairwise forces) calculations between TM1 pore lining helix and L89 residue in the absence (WT) and presence of the modification (L89R1). These data suggest that MTSSL's presence causes minimal distortion to the channel gate and is 609x smaller than the effect this has on lipid occupancy reduction, within the site of allostery (NP).
- d) We calculated the expected pore opening diameter of the measured sub-conducting state (now present in both modified variants L89W and L89R1) in the absence of applied tension (250-270pS). Our calculations indicate this sub-conducting state which is present and resolvable in $n=11$ patches with L89W and $n=9$ patches with L89R1, but not WT channel, is consistent with ~ 8.8 - 9.2 Å pore opening diameter. Finally, we independently calculated the structural change observed by PELDOR (between lipids and detergent), only for L89R1 and out of 13 sites tested. This change is consistent with a pore diameter of ~ 9 Å providing an “extra” link between our functional (electrophysiology) and structural (PELDOR) observations.

In order to adequately support these two separate inferences, the authors should have to add three specific parts to the paper.

1. It should be demonstrated that MTSL and a few other MTS reagents (preferably bulky and hydrophobic) shift the activation curve of the spheroplast-expressed L89C TbMscL population to the left. Complete labeling is not required, even partial labeling may produce a shoulder in the activation curve. MscS expressed in the same spheroplasts should be used as an internal tension reference. Measurements in spheroplasts must be done, they should provide a reliable population response.

We respectfully disagree that sulfhydryl modifiers (including MTSL, other MTS-reagents and proxyl spin labels as maleimido-proxyl) are able to react with inner-membrane *E coli* proteins and thus MscL, in order to measure the effect on its conductance and pressure activation threshold. In particular, previous studies provided direct experimental evidence and suggest otherwise. For instance, *Joseph et al., 2015, Angew Chem; Joseph et al., 2016, JACS*) and others have demonstrated that the MTS- could pass through the outer membranes and is immediately being reduced in the periplasm. We here cite a part of from their JACS article:

“The outer membrane is permeable to molecules below 600 Da and the MTSL could easily reach periplasm...”

“However, we have never observed an EPR signal following “MTSL” labeling of E. coli cells expressing “cys-less” BtuB (E coli outer membrane) or other cysteine mutations located in periplasm. Attempts to label cysteines at the periplasmic interface with maleimido-proxyl also did not give a signal, ruling out

any possible interference from the disulfide bond formation (Dsb) system. These observations suggest that the MTSL is reduced following entry into the periplasm. We demonstrated previously that a spin-labeled CNCb1, which binds tightly to BtuB, is not reduced by cells confirming that the reduction must happen only after entry into the periplasm.

“As we reported earlier, no signal could be detected for the WT cell pellet (Joseph et al., 2015, Angew Chem) or for cysteine mutations located in periplasm (8C, 9C).”

Furthermore, the new “in-cell EPR” field had to take the inevitable reduction and cleavage into account and indeed new spin labels and alternative linking strategies have emerged to address these challenges (Martorana et al., 2014, JACS; Qi et al., 2014, JACS).

Formed spheroplasts suitable for patch clamp electrophysiology experiments retain both membranes. We copy from Martinac et al., 2013, *Methods Mol Biol*: “The best spheroplasts for obtaining a giga-ohm seal are the shiny ones. These spheroplasts have two membranes (outer and inner one, since *E coli* is a Gram-negative bacterium).”

Since, upon entering the periplasm any MTS sulfhydryl modifier will be reduced and it will thus not be able to label TbMscL L89C or any other *E coli* inner-membrane protein e.g. whether the protein site is exposed or not, thus it is not technically possible to use MTS reagents and perform the suggested experiments.

However, we have now performed new electrophysiology measurements for L89W and included in the revised MS (Fig 3b,c & S6c). We show that other than MTSSL modifications (i.e. W that will not be reduced nor reductively cleaved) located at the same site (L89W) also cause a dramatic effect on MscL’s pressure activation threshold and therefore function (for more details regarding new data on L89W, please see response below).

The single-channel results presented on Fig. 3b are not reliable without patch imaging. There could be an accidental slippage of liposome patches leading to higher sensitivity.

Sorry, but we strongly disagree with the Reviewer’s comment; we strongly believe that there is no evidence of accidental slippage. The latter may occur:

a) over time and after 5 min following excision from the GUVs (Nomura et al., 2012, *PNAS*). To avoid this in all our patches (>40) we applied tension immediately after (few seconds) and up to 1 min maximum following excision. We apologized we have not included this information in the experimental methods of our previous versions, but we now include in the revised manuscript.

Lines (814-815): “In all cases tension was applied within few seconds (1 min maximum) and immediately after excision, to avoid any accidental patch slippage.”

b) and accounts only for ~20mmHg change in the applied tension. The changes we observed between the WT and L89W exceeded ~100mmHg and measurements were narrowly distributed for both WT and L89W channels. This dramatic difference in threshold could definitively not be a consequence of accidental slippage.

c) resulting to WT channel recording variability and inconsistency with reported values. Namely any slippage would introduce significant “randomness” within measurements among different labs and also patches, resulting in large variability in WT pressure thresholds. All WT channel independent patches gave rise to activation thresholds of ~150 mmHg, which are in complete agreement with previous studies (Mukherjee, et al., 2014, *FASEB J*), while L89W activated at ~ 60mmHg with very narrowly distributed thresholds. We only observed limited variability with L89R1, which can be attributed to the ~60% labelling (Table S1 & S3), while the L89R1 distribution shifted to WT levels, when L89R1 was incubated with DTT for MTSL removal. Patch slippage cannot account for this change in sensitivity.

Moreover, L89W which possesses the equivalent of 100% modification efficiency and exhibited a large decrease in gating threshold pressure, also shows very little variability across the $n = 5$ patches. Finally, L89R values range between, and not beyond, the equivalent WT and L89W data while L89R1 shifted to WT levels when L89R1 was incubated with DTT, further suggesting there is no accidental slippage within our patches.

The statistics is unacceptably small.

There is no evidence to suggest that the data are not reproducible. We used independent numbers of patches that either equals or exceeds the n -values reported in multiple well-established studies in the field of MS channels (MscS and MscL) under applied tension. In particular, this number varies from $n=4$ to $n=5-8$ and for a single case $n=9-15$ (Nomura *et al.*, 2012, PNAS) and $n=3-4$ (Mukherjee *et al.*, FASEB J, 2014). In general, with ion channel gating and permeation parameters so accurately measured by patch clamp electrophysiology, there is little variability between the behaviour of identical ion channel proteins, thus n -values with single digits are the norm. In our study we obtained highly reproducible single channel recordings with both WT and L89W TbMscL, with properties of the former in complete agreement with reported conductance and activation threshold with the same TbMscL construct in GUVs (Mukherjee, *et al.*, 2014, FASEB J). For the rest of our modified variants, we obtained independent patches from $n=9$ for L89R1, $n=7$ for sub-conducting L89R1, $n=9$ for L89R1 with DTT reduction to L89C, and now $n= 5$ for L89W and $n=11$ for sub-conducting L89W, which lies at the upper limit or exceeds the n -values reported in many previous electrophysiology studies. For these reasons, we respectfully disagree with the Reviewer that “statistics” are unacceptably small, they are the same or more than many previously published reports on similar systems. Furthermore, the data enabled nonparametric analysis, which revealed differences between groups with $p<0.001$ (Kruskal-Wallis), which would not be possible if n -values were too small, relative to the measured effect.

2. If the authors want to demonstrate the effect of a bulky sidechain near the entrance, the same measurements should be done with the L89W mutant, for the same reason.

We have now performed L89W patch clamp electrophysiology in GUVs under applied tension. We observe in patches to which we applied tension an activation at dramatically lower activation thresholds compared to WT channel (Fig 1). Further, for ($n=11$) we resolved a sub-conducting state of ~ 250 pS, similar to the ~ 270 pS we previously identified for L89R1, in the absence of applied tension. Finally, we observed very low threshold variability within multiple tested L89W patches, consistent with a 100% “labelled/modified” mutant.

Fig 1 (see also Fig 3b,c & S6c in the MS). a. Representative patch clamp recordings at 20 mV from patches excised from GUVs containing L89W (modified) TbMscL protein. The pressure applied during the recording is indicated below each trace. The current levels representing the closed (“C”) and fully

opened (“O”, approx. 60 pA) channels are indicated. b. Distribution of threshold pressures, applied at 10 mmHg intervals, at which pressure-activated MscL full channel openings were first observed. Data points represent individual excised patches, whiskers show the full range of values, the boxes represent the 25 to 75 percentile range, and the horizontal line is the median. Samples are significantly different ($p < 0.001$, Kruskal-Wallis ANOVA), with 7/9 L89R1 and 5/5 L89W values below the median ($p < 0.005$, Moody Median test). c. Representative recording at +20 mV from a patch excised from GUV containing L89W TbMscL, with no externally-applied pressure. The brief downward deflections (C), approximately 5 pA, are closures from an otherwise constitutively active sub-conductance state (S).

We have now included new Fig 3b,c and Fig S6c and added legend and main text accordingly.

We write:

Lines (68-69): “...of modifications on this site...”

Line 261 (legend): “c. Distribution of threshold pressures, applied at 10 mmHg intervals, at which pressure-activated MscL full channel openings were first observed. Data points represent individual excised patches, whiskers show the full range of values, the boxes represent the 25 to 75 percentile range, and the horizontal line is the median. Samples are significantly different ($p < 0.001$, Kruskal-Wallis ANOVA), with 7/9 L89R1 and 5/5 L89W values below the median ($p < 0.005$, Moody Median test).”

Line (284-290): “Patches excised from GUVs containing L89W TbMscL protein, like with L89R1 modified protein, exhibited a high level of background current in the absence of applied pressure. Occasionally, unitary closures were recorded, which enabled the calculation of a mean unitary conductance of 0.248 ± 0.022 nS ($n=11$) (Fig S6c). In some patches, applying negative pressure evoked opening of a channel with mean conductance of 2.43 ± 0.27 nS ($n=5$), with pressure thresholds ranging -60 to -80 mmHg (Fig 3B).”

Line 452 (legend): “(or equally L89W)”

Lines (475-478): “Importantly, when L89 was mutated to W, we recorded substantial conformational changes occurring distal to the site of allostery NP, in regions which were previously structurally unchanged in the absence of NP modifications (Fig 4).”

Lines (485-486): “...and test the effect on channel structure by introducing other than MTSSL modifying molecules at the allosteric site,...”

Lines (505-508): “~270 pS and ~250 pS state evident in L89R1 and L89W respectively (but not WT) (Fig 3b & S6b,c) in the absence of externally applied tension is consistent with a ~9.2 Å and ~8.8 Å pore diameter estimated opening for L89R1 and L89W respectively(79)”

Lines (561-565): “This may also account for the increased variability of the pressure activation threshold for L89R1, which is on average lower than WT, but then approaches WT values after DTT incubation. This is supported by the low activation threshold of L89W, which is equipped with a tryptophan on every equivalent site, and exhibited very limited variability similar to WT (Fig 3b & S6a).”

3. If the authors claim that the gate is destabilized by the absence of interactions with the aliphatic chains occupying the pockets (and not the distortion of structure and rotation of helices by the bulky, partially hydrophilic MTSL chain),

We have now calculated the pairwise forces between L89R1 and WT with TM1 pore lining helix respectively test whether there is a structural “distortion” caused by the presence of MTSSL at the NP entrance. Our analysis shows that the ratio of distortion to TM1 helix the presence of MTSSL induces, is 609x smaller than the effect its presence has to the total energy ratio induced by the lipid chains directly to the NPs. Therefore, although the expected minimal structural distortion the modification seems to have, this is negligible and incapable of promoting global channel changes we measured by PELDOR and electrophysiology, for two types of modification (R1 and W) which although different, promote similar channel changes.

We have extended Table S5 and included new pairwise force calculations between L89R1 and TM1 (Table S5 bottom row).

We also write:

Lines (404-405): “*d)TM1 and residue 89 (L and MTSSL) (Table S5).*”

Lines (411-413): “*and between pore-lining helix TM1 and L89 or L89R1 (Table S5). This is consistent with a model in which the modification a) does not cause a structural distortions to the gate and...*”

Lines (529-532): “*...without those being structural distortions caused by MTSL’s presence (Table S5).*”

they have to carefully analyse the dynamics (RMSD) of the TM1 bundle (gate) in simulations in which the aliphatic chains are either allowed or disallowed to enter the pockets. If the authors want to connect the altered dynamics of the gate with the observed sub-conducting states, then they have to analyse and demonstrate the changes in gate solvation.

We thank the Reviewer for their comment and the opportunity to provide further computational evidence which supports our proposed model. We have now implemented RMSD calculations for the last 20ns of the atomistic MD simulations (80-100ns) of TM1 in correlation to the number of lipid contacts for both the L89R1 and WT channels and include these new data (lines 397-398, 529-532, 791-794) and new Fig S11c).

We a) observe structural perturbations of the pore lining helix TM1 (L89R1) (given we do not intentionally apply artificially large force fields for reasons we explicitly explain in the MS, lines 542-544) and direct correlation between these changes and lipid chain availability within the NPs. These results suggest the TM1 undergoes structural rearrangements in a concerted manner to lipid occupancy variability within the NPs, and therefore conclude that the channel “senses” and directly responds to subtle lipid chain occupancy changes within its NPs (the site of allostery). Further, analysis on the WT channel (control), which presents constant lipid pocket availability resulted in no TM1 structural perturbations, consistent with a closed channel for which the NPs are constantly (or vast time majority) occupied by lipid chains.

Fig 2. TM1 RMSD (grey line) and A18 lipid contacts (red line) over time (final 20ns of the 100ns total atomistic MD simulation) for L89R1 and WT TbMscL respectively, along with the calculated correlation for each case

We write:

Lines (397-398): “*Changes in lipid chain occupancy within the L89R1 NPs correlated with RMSD changes of the TM1 pore-lining helix, not evident in the WT channel (Fig S11c).*”

Lines (529-532): “*...while lipid occupancy changes within the allosteric NP site showed correlation with structural perturbations occurring at the TM1 pore-lining helix, which controls ion pore conductance (Fig S11c), ...*”

Lines (791-794): “Pair wise energy, RMSD and lipid acyl-chain atom contacts with TbMscL amino acids were calculated using Gromacs command *gmx energy*, *gmx rms* and *gmx mindist* respectively.”

The only result that seems to be real is the small expansion of the barrel when the cysteine in one specific position 89 is modified with MTSL.

We thank the reviewer for their comment and are pleased that the reviewer now recognises there is indeed a “structural” and thus mechanical response of the high pressure threshold channel, as a consequence of modifying the entrance (L89) of the allosteric site (NP). Indeed, this is the key message that we wish to communicate to the scientific community. Regarding the magnitude of the structural change we respectfully disagree with the reviewer as our data from multiple state-of-the-art methods (for details see response below) suggest otherwise, e.g. that it is only a small barrel expansion. In particular, we experimentally measured using PELDOR a 5 Å pore diameter change occurring (L89R1) between DDM and lipids (both liposomes and nanodiscs).

We write in the revised MS:

Line (222): “...cytoplasmic leaflet channel domain of 5 Å, reaching a total diameter of ~9 Å...”

Lines (497-499): “The expanded L89R1 TbMscL state forms an open pore of ~ 9 Å in total diameter at the cytoplasmic leaflet site, e.g. 5 Å diameter increase reported by PELDOR, added to the 3-4 Å diameter of the constriction site of the closed TbMscL state (PDB 2OAR),...”

This measured by PELDOR conformational change, along with the 4 Å initial pore diameter of the x-ray closed structure (PDB 2OAR, *Chang et al., 1998, Science*) results in a substantial channel expanded state, with a 9 Å pore diameter opening, sufficient to conduct ions (Fig 3).

Structural change between DDM vs Nanodisc L89R1 (PELDOR)

Liu, Z., Gandhi, C. S., and Rees, D. C., 2009, *Nature*; Li, J., Guo, J., Ou, X., Zhang, M., Li, Y., and Liu, Z., 2015, *Proc Natl Acad Sci U S A*; Chang, G., Spencer, R. H., Lee, A. T., Barclay, M. T., and Rees, D. C., 1998, *Science*; Plotas, 2017, *Methods Enzymol*

Fig 3. Pore inner-diameter change measured by PELDOR in DDM and NDs. This change is added to the closed state (PDB 2 OAR) initial pore diameter resulting in a state of 9 Å total opening diameter, able to conduct ions

To further confirm and characterise this conductive state, we calculated the ion conductance corresponding to an 9 Å channel pore opening diameter according to *Cruickshank et al., 1997, Biophys J*, (Fig 4). Our calculations resulted in 270 pS for 9.2 Å (L89R1) and 250 pS for 8.8 Å (L89W), pore conductance. There is thus an excellent agreement between the “structural” change we observe by PELDOR and the functional sub-conducting state we resolved within n=18 patches (under no externally-applied pressure) of our NP-modified variants, e.g. L89R1 and L89W, but not for WT.

Functional state (electrophysiology)

Cruickshank, C. C., Minchin, R. F., Le Dain, A. C., and Martinac, B. (1997) Estimation of the pore size of the large-conductance mechanosensitive ion channel of *Escherichia coli*. *Biophys J* 73, 1925-1931

Calculation of the MscL pore size using the channel conductance

The model proposed by Hille (1968) relating pore size and resistance to ion flow in the channel was used to estimate the pore diameter of the MscL. The model assumes that the channel is a uniform cylinder of radius r , length l , in a solution of resistivity ρ . Then the resistance through the channel, R_C , is given by:

$$R_C = \left(l + \frac{\pi r}{2} \right) \frac{\rho}{\pi r^2} \quad (4)$$

and since the conductance, g , is the inverse of R_C , Eq. 4 can be rearranged to give the channel diameter, d , from:

$$d = \frac{\rho g}{\pi} \left(\frac{\pi}{2} + \sqrt{\frac{\pi^2}{4} + \frac{4\pi l}{\rho g}} \right) \quad (5)$$

Calculated total channel pore diameter: **d = 9.2 Å**

Fig 4. Calculation of MscL pore diameter corresponding to a channel with a 270pS conductance (i.e. sub-conducting state we measured in the absence or null tension for L89R1 and L89W). The total diameter is 9.2 Å (270 pS, L89R1, Fig 3b and S6b) and 8.8 Å (250 pS, L89W, Fig S6c), consistent with structural changes measured by PELDOR.

We write:

Lines (505-508): "... ~270 pS and ~250 pS state evident in L89R1 and L89W respectively (but not WT) (Fig 3b & S6b,c) in the absence of externally applied tension is consistent with a ~9.2 Å and ~8.8 Å pore diameter estimated opening for L89R1 and L89W respectively(79)."

This state therefore represents a significant conformational change, which leads to a substantial pore increase and not just an incremental structural distortion. Importantly, it is the direct consequence of the mechanical response to occurred changes within MscL's allosteric site. A pore diameter size of 9 Å does not represent a negligibly expanded state in any membrane channel pore in nature and importantly it is able to readily conduct ions and allow the flow through of molecules. For instance, MscS' "fully" open pore is 14 and eukaryotic pores are only few Å wide. Given TbMscL is the channel with the highest pressure activation threshold known in nature, including EcMscL, a mechanical response of such magnitude demonstrates the vital importance of allosteric sites in mechanical gating.

How this observation relates to the gating mechanism

We thank the reviewer for their comment and the opportunity to clarify and expand on this crucial point. We do not claim that "full" channel opening occurs when the lipid moves out of the NPs and clearly stated that extra tension will be required (Lines 585-586 and Fig S12). We establish that this is the first and very important step for the initiation of the gating process. We demonstrate that the lipid which moves first and away from the allosteric site NP, results in destabilisation of the channel closed state and dramatically increases the open probability of a substantial (~1/3 of full opening diameter) sub-conducting state. In particular, this first lipid NP removal reduces by more than 60% (~30-60mmHg, instead of 150-160 mmHg are needed) the total applied tension required for full opening. Therefore, only an "extra" 40% is now required for the channel to transit from this state (~9 Å, 270pS) to its fully open state (~25 Å, 2.5-3nS). Elucidating the final step or steps of the gating process (from sub to full opening), will be the focus of our (and also other groups) future studies, but it is clearly out of the scope of the current study.

Similar sub-conducting states have been previously observed and characterised in EcMscL (*Sukharev et al., 1999, J Gen Phys; Sukharev et al., 2001, Nature*). These were 1/10 of full opening signal height (~200-300pS) and usually appeared prior to full channel opening events under applied tension. However, the physical meaning and importance of these states in channel's gating mechanism have remained elusive.

We hereby establish that this conducting state is the first step in MscL's gating process and is the direct consequence of the lipid, which "moves first" away from the allosteric NP site. To do so, the lipid "consumes" 60-70% of the total energy required for full channel opening, which is sufficient to promote a ~9 Å pore opening diameter state, that is approximately 1/3 of the fully open expected diameter (10% of total conductance and 30% of total pore diameter). These lipids may not all or simultaneously move away from the 5 available allosteric sites (pentameric MscL), which may indicate that the participation of the five identical subunits is assymmetric, consistent with previous studies (*Birkner et al., 2012, PNAS*) and also with the effect of modification efficiency, e.g. leading to asymmetric lipid NP occupancy, we have observed.

We have now made amendments in the text to better reflect these points. We have also created a new Figure 5 (Fig S12 in the MS) to summarise these observations and highlight the relation to MscL's overall gating NP lipid removal has, in order facilitate efficient communication with the reader.

Fig 5. Fig S12. Summary of experimental observations of the effect that NP lipid removal has to MscL's overall gating. Lipids moves first out of the NP resulting in a structural (PELDOR) and functional (electrophysiology) response of the channel leading to a sub-conducting state which possess 10% and 30% of channel's total conductance and pore diameter respectively. To implement this step, e.g. lipid to be removed, 60-70% of the total tension required for MscL full opening has to be applied. A further 30-40% of the initial total tension is then only required to fully open the channel (2.5nS, 30 Å).

We write:

Lines (584-605): *“Following channel activation by NP lipid removal (sub-conducting state) within the membrane, an “extra” ~40% of the total (full opening) applied tension (60mmHg out of 150mmHg) is required, to promote channel transition from the sub-conducting (~9 Å and 1/3 of the fully open pore) to the fully open state (~25 Å) (Fig S12).*

We demonstrate that the “lipid has to move first” and out of the site of allostery (NP). We establish this event constitutes the first and essential step for MscL's mechanical response to occur. This is also consistent with the presence of two and one lipids within the NPs of MscS (larger NPs than MscL), in the closed (22) and open (15) state respectively, suggesting that a single lipid movement, in or out of the NPs, could lead to structural rearrangements, as the lipid moves first model would predict.

Lipid chain removal from the NP could be achieved a) through modification at the entrance of the site of allostery NP either in the absence of bilayer compression or under null (lipid-glass adhesion) tension for NP-entrance modified channels or b) upon tension application for WT or non-NP-entrance modified channels. Independent to the method used for facilitating NP lipid chain removal, the consequence is the destabilization of the closed state of MscL and the dramatic increase of the open probability of the first substantial conducting state of the channel. We demonstrate that after this first step is reached, the applied tension required for full opening of MscL's pore accounts only for ~40% of initial tension required (~60mmHg, instead of ~150mmHg). Therefore, this tension is mostly required to remove the lipid chain(s) from the NP and allow MscL to transit from the closed to the sub-conducting state (structural

response upon NP lipid removal). The latter has an opening ~30% (9 Å) of the expected total pore diameter (25 Å) and ~10% (250-270pS) of the total pore conductance (2.5-3nS) (Fig S12).”

and supports ‘allostery’ remains unclear. The rest of the data represent a lot of work but does not create a cohesive and convincing story.

MscL is a mechanosensitive (not ligand-gated) ion channel and is activated by increased membrane tension. Membrane lipids are able to modulate MS channels without directly accessing their pore, thus achieving gating through an allosteric mechanism. Previous successful attempts to modulate MscL’s pore were implemented through application of triggers, which acted directly from within channel’s pore (Lines 118-119) (i.e. modified pore residues inaccessible to lipids), not being allosteric and thus not biologically relevant. Therefore, although these studies were highly valuable for controlling MS channel pores, they did not provide an explanation on how tension is transmitted from the lipids to the channel gate in order to promote gating through the incorporation of lipids. According to the “lipid moves first” model, the lipid is a negative allosteric modulator, which acts distal to the pore. Channel opening is proposed to occur due to lipid moving out from the NPs, leaving the latter unoccupied as a consequence of increased tension, and resulting to a decrease of channel activation free energy. The lipid, following a membrane tension decrease, moves back and occupies this pocket in order to increase the free activation energy and keep the channel closed, otherwise a “leaky” channel would be lethal to the cell. Therefore, the lipid, in the case of MS channels, does not act as a “normal” ligand, but instead as a negative allosteric modulator and conformational changes are not driven by specific ligand-protein interactions. In order to elucidate this naturally occurring allosteric mechanism, it was essential to:

- a) **Identify the location of site of allostery.** We have therefore used rigorous and multiple site directed modification with multiple sites spanning all channel domains including both TM1 and TM2 helices, the S helix and cytosolic bundle
- b) **Demonstrate any structural changes occur only at the allosteric site.** We have thus performed PELDOR measurements, a method which offers subÅ resolution and followed transitions to identify the allosteric site both in detergent solution where membrane compression is not present and two distinct lipid/native environments of different lipid composition and curvature. We further identified optimal lipid types by performing two independent lipid qualitative and quantitative analyses (i.e. NMR and ES -mass spectrometry)
- c) **Demonstrate structural changes initiated at the allosteric site lead to global changes.** We introduced a modification (W) at the entrance of identified allosteric site and recorded large (>7 Å and >12 Å respectively for two distal to the NPs residues) conformational rearrangements >22 Å away from the site of allostery (NP). We further now show that these changes are transmitted to the channel pore lining helix and correlate with the degree of lipid occupancy at the allosteric site. Finally, W modification could act as a further control for excluding any spin label orientation artefacts on distance modelling and measurements.
- d) **Link the occurring structural changes to equivalent functional changes.** We performed patch clamp electrophysiology and showed that two distinct modifications at the entrance of the site of allostery (NP) with different modifications had a dramatic effect on pressure activation threshold, in complete agreement with our model. Moreover, both modified mutants promoted a novel state, due to a spontaneous MS channel structural response, as a direct consequence of disrupting the entrance of the site of allostery. Importantly, we demonstrated that these structural changes are equivalent to the occurring functional changes and thus mechanical responses.
- e) **Demonstrate that any occurring allostery is reversible, as any naturally occurring process should be.** We showed that the structural change which occurs in the absence of membrane (detergent solution) is restored in the presence of membrane compression for two lipid systems of a different lipid composition and curvature (nanodiscs and liposomes). In parallel, we showed that the severe effect on channel function is restored, when the modification responsible for this change, is removed (i.e. incubation of L89R1 with DTT reducing agent).
- f) **To demonstrate that the negative allosteric modulators (NP-lipids) should directly interact with the modification at the molecular level.** We set up and performed atomistic MD simulations of the modified channel within lipid bilayers and quantified the effect of the modification has, to the lipid chain occupancy within the allosteric site (NPs). We showed that the modification

a) significantly reduces lipid chain occupancy within the NPs, thus reducing the interaction between the lipid membrane and the allosteric site and also b) has the structural “capacity” to efficiently disrupt the contact between the lipid bilayer and the channel by restricting entrance of lipid chains into the NPs. Further, we now show that there is a correlation between lipid chain NP occupancy and structural perturbations of TM1 pore lining helix. Finally, we provided experimental evidence (i.e. ESEEM, LT CW-EPR and PELDOR combined with charged cluster neutralisation), on which part of the allosteric modulator (i.e. the lipid) is responsible for channel gating. We demonstrated that is the lipid “acyl-chain” occupancy within the site of allostery, rather than the specific lipid “headgroup” binding to the charged cluster responsible for MscL’s gating response.

g) Implement and provide appropriate experimental and computational controls and where available data our controls should be consistent with existing literature. We expressed, purified and modified over 20 mutants, all resulting in well folded proteins and spanning sites proximal and distal to the allosteric site. We obtained high quality PELDOR spectra for 13 of these sites. For the ones which data deviated from the closed state x-ray structure model we performed additional PELDOR in lipids and demonstrated that only the identified mutant located at the entrance of the allosteric site distances reverts back. We used 88W which is immediately adjacent to 89 but not at the entrance of the NPs, paired with other spin labelled residues and observed no structural effect with PELDOR, in contrast to L89W. For our electrophysiology experiments we initially performed WT channel measurements of multiple and independent patches and our data were in complete agreement with literature, ruling out any measurement artefacts. For our MD simulations we performed measurements in detergent along with multiple WT channel experiments to control and elucidate the interplay between the modification and the allosteric site.

Further, a recent CryoEM structure of MscS (*Rasmussen et al., Jul 2019*), resolved **two** lipids within MscS’ NPs and the channel was found to be in the closed state. The open x-ray MscS structure (*Pliotas et al., 2015*) contained **one** lipid within its NPs. According to the “*lipid moves first*” model, one lipid has to move in and out of the NPs to close and open the channel respectively. The effect of NP lipid removal on mechanosensitive ion channel structure and function, is key to the fundamental understanding of the molecular basis of mechanosensation. Clearly this “static” structural finding provides an important and supportive experimental evidence of the validity of the lipid moves first model and is in complete agreement with our findings and initial hypothesis for MscL.

We now discuss these recent findings and include this reference within our revised manuscript. We write:

(Lines 99-100): “*and a recent cryoEM MscS structure in nanodiscs favored the “lipid moves first” model and provided crucial structural evidence (22), suggesting...*”

(Lines 132-133): “*MscS NPs are filled with lipid chains(15, 22)*”

(Lines 590-593): “This is also consistent with the presence of two and one lipids within the NPs of MscS (larger NPs than MscL), in the closed (22) and open (15) state respectively, suggesting that a single lipid movement, in or out of the NPs, could lead to structural rearrangements, as the lipid moves first model would predict.”

Our story is full and completed, cohesive, allowing the interconnection of data generated by multiple independent state-of-the-art experiments leading to a convincing outcome. Importantly, none of our generated data, which “represent a significant amount of cutting edge methodologies and work”, contradict our model, but in their full majority agree with it.